# A Survey on Causal Discovery Methods for I.I.D. and Time Series Data

**Uzma Hasan**  *uzmahasan@umbc.edu*
*Causal AI Lab*
*Department of Information Systems*
*University of Maryland, Baltimore County*
*Baltimore, MD, USA*

**Emam Hossain**  *emamh1@umbc.edu*
*Causal AI Lab*
*Department of Information Systems*
*University of Maryland, Baltimore County*
*Baltimore, MD, USA*

**Md Osman Gani**  *mogani@umbc.edu*
*Causal AI Lab*
*Department of Information Systems*
*University of Maryland, Baltimore County*
*Baltimore, MD, USA*

**Reviewed on OpenReview:** *https://openreview.net/forum?id=YdMrdhGx9y*

## Abstract

The ability to understand causality from data is one of the major milestones of human-level intelligence. Causal Discovery (CD) algorithms can identify the cause-effect relationships among the variables of a system from related observational data with certain assumptions. Over the years, several methods have been developed primarily based on the statistical properties of data to uncover the underlying causal mechanism. In this study, we present an extensive discussion on the methods designed to perform causal discovery from both independent and identically distributed (I.I.D.) data and time series data. For this purpose, we first introduce the common terminologies used in causal discovery literature and then provide a comprehensive discussion of the algorithms designed to identify causal relations in different settings. We further discuss some of the benchmark datasets available for evaluating the algorithmic performance, off-the-shelf tools or software packages to perform causal discovery readily, and the common metrics used to evaluate these methods. We also evaluate some widely used causal discovery algorithms on multiple benchmark datasets and compare their performances. Finally, we conclude by discussing the research challenges and the applications of causal discovery algorithms in multiple areas of interest.

## 1 Introduction

The identification of the cause-effect relationships among the variables of a system from the corresponding data is called Causal Discovery (CD). A major part of the causal analysis involves unfolding the *cause and effect relationships* among the entities in complex systems that can help us build better solutions in health care, earth science, politics, business, education, and many other diverse areas (Peyrot (1996), Nogueira et al. (2021)). The *causal explanations* precisely the causal factors obtained from a causal analysis play an important role in decision-making and policy formulation as well as in foreseeing the consequences of interventions without actually doing them. Causal discovery algorithms enable the *discovery of the underlying causal structure* given a set of observations. The underlying causal structure also known as a causal graph

(CG) is a representation of the cause-effect relationships between the variables in the data (Pearl (2009)). Causal graphs represent the causal relationships with directed arrows from the cause to the effect. Discovering the causal relations, and thereby, the estimation of their effects would enable us to understand the underlying *data generating mechanism* (DGM) better, and take necessary interventional actions. However, traditional Artificial Intelligence (AI) applications rely solely on predictive models and often ignore causal knowledge. Systems without the knowledge of causal relationships often cannot make rational and informed decisions (Marwala (2015)). The result may be devastating when correlations are mistaken for causation. Because two variables can be highly correlated, and yet not have any causal influence on each other. There may be a third variable often called a latent confounder or hidden factor that may be causing both of them (see Figure 2 (a)). Thus, *embedding the knowledge of causal relationships* in black-box AI systems is important to improve their explainability and reliability (Dubois & Prade (2020), Ganguly et al. (2023)). In multiple fields such as healthcare, politics, economics, climate science, business, and education, the ability to understand causal relations can facilitate the formulation of better policies with a greater understanding of the data.

The standard approach to discover the cause-effect relationships is to perform randomized control trials (RCTs) (Sibbald & Roland (1998)). However, RCTs are often infeasible to conduct due to high costs and some ethical reasons (Resnik (2008)). As a result, over the last few decades, researchers have developed a variety of methods to unravel causal relations from purely observational data (Glymour et al. (2019), Vowels et al. (2021)). These methods are often based on some assumptions about the data and the underlying mechanism. The *outcome* of any causal discovery method is a causal graph or a causal adjacency matrix where the cause and effect relations among the entities or variables are represented. The structure of a causal graph is often similar to a *directed acyclic graph (DAG)* where directed edges from one variable to another represent the cause-effect relationship between them. Figure 2 (b) represents a causal graph showing the factors that are responsible for causing Cancer. This type of structural representation of the underlying data-generating mechanism is beneficial for understanding how the system entities interact with each other.

There exists a wide range of approaches for performing causal discovery under different settings or assumptions. Some approaches are designed particularly for *independent and identically distributed (I.I.D.) data* (Spirtes et al. (2000b), Chickering (2002)) i.e. non-temporal data while others are focused on *time series data* (Runge et al. (2019), Hyvärinen et al. (2010)) or temporal data. Since in real-world settings, both types of data are available in different problem domains, it is essential to have approaches to perform causal structure recovery from both of these. Recently, there has been a growing body of research that considers *prior knowledge incorporation* for recovering the causal relationships (Mooij et al. (2020), Hasan & Gani (2022), Hasan & Gani (2023)). Although there exist some surveys (see Table 1) on causal discovery approaches (Heinze-Deml et al. (2018), Glymour et al. (2019), Guo et al. (2020), Vowels et al. (2021), Assaad et al. (2022b)), none of these present a comprehensive review of the different approaches designed for structure recovery from both I.I.D. and time series data. Also, these surveys do not discuss the approaches that perform causal discovery in the presence of background knowledge. Hence, the goal of this survey is to provide an overview of the wide range of existing approaches for performing causal discovery from I.I.D. as well as time series data under different settings. Existing surveys lack a combined overview of the approaches present for both I.I.D. and time series data. So in this survey, we want to introduce the readers to the methods available in both domains. We discuss prominent methods based on the different approaches such as conditional independence (CI) testing, score function usage, functional causal models (FCMs), continuous optimization strategy, prior knowledge infusion, and miscellaneous ones. These methods primarily differ from each other based on the primary strategy they follow. Apart from introducing the different causal discovery approaches

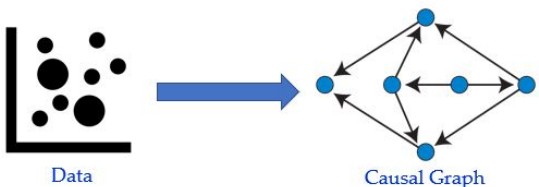

Figure 1: Causal Discovery: Identification of a causal graph from data.

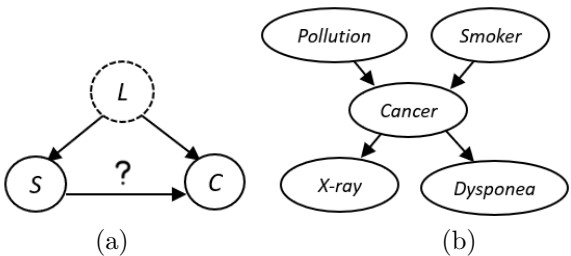

(a)                  (b)

Figure 2: (a) Latent confounder $L$ causes both variables $S$ and $C$, and the association between $S$ and $C$ is denoted by **?** which can be mistaken as causation. The graph in (b) is a causal graph depicting the causes and effects of cancer (Korb & Nicholson (2010)).

and algorithms for I.I.D. and time series data, we also discuss the different tools, metrics, and benchmark datasets used for performing CD and the challenges and applications of CD in a wide range of areas.

Table 1: Comparison among the existing surveys for causal discovery approaches. A discussion on the different approaches can be found in section 3 and section 4.

| Survey | Focused Approaches | I.I.D. Data | Time Series Data |
|---|---|:---:|:---:|
| Heinze-Deml et al. (2018) | Constraint, Score, Hybrid & FCM-based approaches. | ✓ | × |
| Glymour et al. (2019) | Traditional Constraint-based, Score-based, & FCM-based approaches. | ✓ | × |
| Guo et al. (2020) | Constraint-based, Score-based, & FCM-based approaches. | ✓ | × |
| Vowels et al. (2021) | Continuous Optimization-based. | ✓ | × |
| Assaad et al. (2022b) | Constraint-based, Score-based, FCM-based, etc. approaches for time series data. | × | ✓ |
| This study | Constraint-based, Score-based, FCM-based, Hybrid-based, Continuous-Optimization-based, Prior-Knowledge-based, and Miscellaneous. | ✓ | ✓ |

To summarize, the structure of this paper is as follows: *First*, we provide a brief introduction to the common terminologies in the field of causal discovery (section 2). *Second*, we discuss the wide range of causal discovery approaches that exist for both I.I.D. (section 3) and time-series data (section 4). *Third*, we briefly overview the common evaluation metrics (section 5) and datasets (section 6) used for evaluating the causal discovery approaches, and report the performance comparison of some causal discovery approaches in section 7. *Fourth*, we list the different technologies and open-source software (section 8) available for performing causal discovery. *Fifth*, we discuss the challenges (section 9.1) and applications (section 9.2) of causal discovery in multiple areas such as healthcare, business, social science, economics, and so on. *Lastly*, we conclude by discussing the scopes of improvement in future causal discovery research, and the importance of causality in improving the existing predictive AI systems which can thereby impact informed and reliable decision-making in different areas of interest (section 10).

## 2 Preliminaries of Causal Discovery

In this section, we briefly discuss the important terminologies and concepts that are widely used in causal discovery. Some common notations used to explain the terminologies are presented in Table 2.

Table 2: Common notations.

| Notation | Description |
|---|---|
| $G$ | A graph or DAG or ground-truth graph |
| $G'$ | An estimated graph |
| $X, Y, Z, W$ | Observational variables |
| $X - Y$ | An unoriented or undirected edge between $X$ and $Y$ |
| $X \rightarrow Y$ | A directed edge from $X$ to $Y$ where $X$ is the cause and Y is the effect |
| $X \not\rightarrow Y$ | Absence of an edge or causal link between $X$ and $Y$ |
| $X \rightarrow Z \leftarrow Y$ | V-structure or Collider where $Z$ is the common child of $X$ and $Y$ |
| $\perp\!\!\!\perp$ | Independence or d-separation |
| $X \perp\!\!\!\perp Y \mid Z$ | $X$ is d-separated from $Y$ given $Z$ |

## 2.1 Graphical Models

A graph $G = (V, E)$ consists of a set of vertices (nodes) $V$ and a set of edges $E$ where the edges represent the relationships among the vertices. Figure 3 (a) represents a graph $G$ with vertices $V = [X, Y, Z]$ and edges $E = [(X, Y), (X, Z), (Z, Y)]$. There can be different types of edges in a graph such as directed edges ($\rightarrow$), undirected edges (-), bi-directed edges ($\leftrightarrow$), etc. (Colombo et al. (2012)). A graph that consists of only undirected edges (-) between the nodes which represent their adjacencies is called a **skeleton graph** $S_G$. This type of graph is also known as an **undirected graph** (Figure 3 (b)). A graph that has a mixture of different types of edges is known as a **mixed graph** $M_G$ (Figure 3 (c)). A **path** $p$ between two nodes $X$ and $Y$ is a sequence of edges beginning from $X$ and ending at $Y$. A **cycle** $c$ is a path that begins and ends at the same vertex. A graph with no cycle $c$ is called an **acyclic graph**. And, a directed graph in which the edges have directions ($\rightarrow$) and has no cycle is called a **directed acyclic graph** (DAG). In a DAG $G$, a directed path from $X$ to $Y$ implies that $X$ is an ancestor of $Y$, and $Y$ is a descendant of $X$. The graph $G$ in Figure 3 (a) is a DAG as it is acyclic, and consists of directed edges.

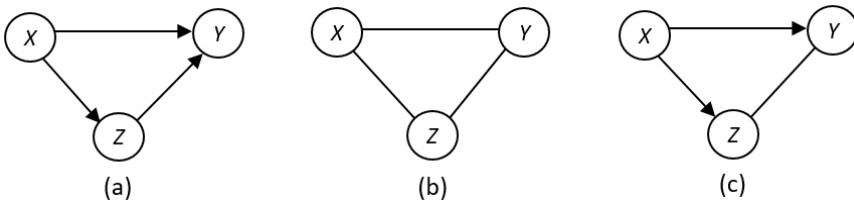

Figure 3: (a) A graph $G$, (b) its *skeleton* graph $S_G$, (c) a *mixed graph* $M_G$ with directed & undirected edges.

There can be different kinds of DAGs based on the type of edges they contain. A class of DAG known as **partially directed acyclic graph** (PDAG) contains both directed ($\rightarrow$) and undirected (-) edges. The mixed graph of Figure 3 (c) is also a PDAG. A **completed PDAG** (CPDAG) consists of directed ($\rightarrow$) edges that exist in every DAG $G$ having the same conditional dependencies, and undirected (-) edges that are reversible in $G$. An extension of DAGs that retain many of the significant properties that are associated with DAGs is known as **ancestral graphs** (AGs). Two different DAGs may lead to the same ancestral graph (Richardson & Spirtes (2002a)). Often there are hidden confounders and selection biases in real-world data. Ancestral graphs can represent the data-generating mechanisms that may involve latent confounders and/or selection bias, without explicitly modeling the unobserved variables. There exist different types of ancestral graphs. A **maximal ancestral graph** (MAG) is a mixed graph that can have both directed ($\rightarrow$) and bidirectional ($\leftrightarrow$) edges (Richardson & Spirtes (2002b)). A **partial ancestral graph** (PAG) can have four types of edges such as directed ($\rightarrow$), bi-directed ($\leftrightarrow$), partially directed (o$\rightarrow$), and undirected ($-$) (Spirtes (2001)). That is, edges in a PAG can have three kinds of endpoints: $-$, o, or $>$. An ancestral graph without bi-directed edges ($\leftrightarrow$) is a DAG (Triantafillou & Tsamardinos (2016)).

## 2.2 Causal Graphical Models

A **causal graphical model** (CGM) or **causal graph** (CG) is a DAG $G$ that represents a joint probability distribution $P$ over a set of random variables $X = (X_1, X_2, \ldots, X_d)$ where $P$ is Markovian with respect to $G$. In a CGM, the nodes represent variables $X$, and the arrows represent causal relationships between them. The joint distribution $P$ can be factorized as follows where $pa(x_i, G)$ denotes the parents of $x_i$ in $G$.

$$P(x_1, \ldots, x_d) = \prod_{i=1}^{d} P(x_i | pa(x_i, G)) \tag{1}$$

Causal graphs are often used to study the underlying data-generating mechanism in real-world problems. For any dataset $D$ with variables $X$, causal graphs can encode the cause-effect relationships among the variables using directed edges ($\rightarrow$) from cause to the effect. Most of the time causal graphs take the form of a DAG. In Figure 3 (a), $X$ is the cause that effects both $Y$ and $Z$ (i.e. $Y \leftarrow X \rightarrow Z$). Also, $Z$ is a cause of $Y$ (i.e. $Z \rightarrow Y$). The mechanism that enables the estimation of a causal graph $G$ from a dataset $D$ is called **causal discovery (CD)** (Figure 1). The outcome of any causal discovery algorithm is a causal graph $G$ where the directed edges ($\rightarrow$) represent the cause-and-effect relationship between the variables $X$ in $D$. However, some approaches have different forms of graphs (PDAGs, CPDAGs, ancestral graphs, etc.) as the output causal graph. Table 3 lists the output causal graphs of some common approaches which are discussed in section 3.

Table 3: List of some CD algorithms with their output causal graphs. A detailed discussion of the algorithms is in section 3. The cells with ✓ represent the type of graph produced by the corresponding algorithm.

| Algorithms | DAG | PDAG | CPDAG | MAG | PAG |
|:---:|:---:|:---:|:---:|:---:|:---:|
| PC | | | ✓ | | |
| FCI | | | | | ✓ |
| RFCI | | | | | ✓ |
| GES | | | ✓ | | |
| GIES | | ✓ | | | |
| MMHC | ✓ | | | | |
| LiNGAM | ✓ | | | | |
| NOTEARS | ✓ | | | | |
| GSMAG | | | | ✓ | |

### 2.2.1 Key Structures in Causal Graphs

There are three fundamental **building blocks** (key structures) commonly observed in the graphical models or causal graphs, namely, **Chain, Fork,** and **Collider.** Any graphical model consisting of at least three variables is composed of these key structures. We discuss these basic building blocks and their implications in dependency relationships below.

**Definition 1 (Chain)** *A chain $X \rightarrow Y \rightarrow Z$ is a graphical structure or a configuration of three variables $X$, $Y$, and $Z$ in graph $G$ where $X$ has a directed edge to $Y$ and $Y$ has a directed edge to $Z$ (see Figure 4 (a)). Here, $X$ causes $Y$ and $Y$ causes $Z$, and $Y$ is called a mediator.*

**Definition 2 (Fork)** *A fork $Y \leftarrow X \rightarrow Z$ is a triple of variables $X$, $Y$, and $Z$ where one variable is the common parent of the other two variables. In Figure 4 (b), the triple $(X, Y, Z)$ is a fork where $X$ is a common parent of $Y$ and $Z$.*

**Definition 3 (Collider/V-structure)** *A v-structure or collider $X \rightarrow Z \leftarrow Y$ is a triple of variables $X$, $Y$, and $Z$ where one variable is a common child of the other two variables which are non-adjacent. In Figure 4 (c), the triple $(X, Y, Z)$ is a v-structure where $Z$ is a common child of $X$ and $Y$, but $X$ and $Y$ are non-adjacent in the graph. Figure 4 (d) is also a collider with a descendant $W$.*

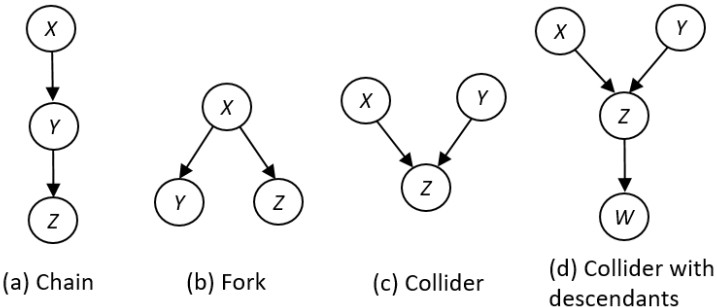

Figure 4: Fundamental building blocks in causal graphical models.

### 2.2.2 Conditional Independence in Causal Graphs

Testing for **conditional independence** (CI) between the variables is one of the most important techniques to find the causal relationships among the variables. Conditional independence between two variables $X$ and $Y$ results when they are independent of each other given a third variable $Z$ (i.e. $X \perp\!\!\!\perp Y \mid Z$). In the case of causal discovery, CI testing allows deciding if any two variables are causally connected or disconnected. An important criterion for CI testing is the *d-separation* criterion which is formally defined below.

**Definition 4 (d-separation)** *(Pearl (1988)) A path p in G is blocked by a set of nodes N if either*

  i. *p contains a chain of nodes $X \to Y \to Z$ or a fork $X \leftarrow Y \to Z$ such that the middle node $Y$ is in $N$,*

 ii. *p contains a collider $X \to Y \leftarrow Z$ such that the collision node $Y$ is not in $N$, and no descendant of $Y$ is in $N$.*

*If $N$ blocks every path between two nodes, then they are d-separated, conditional on $N$, and thus are independent conditional on $N$.*

In *d-separation*, *d* stands for *directional*. The d-separation criterion provides a set of rules to check if two variables are independent when conditioned on a set of variables. The conditioning variable can be a single variable or a set of variables. However, two variables with a directed edge ($\to$) between them are always dependent. The set of testable implications provided by *d-separation* can be benchmarked with the available data $D$. If a graph $G$ might have been generated from a dataset $D$, then *d-separation* tells us which variables in $G$ must be independent conditional on other variables. If every *d-separation* condition matches a conditional independence in data, then no further test can refute the model (Pearl (1988)). If there is at least one path between two variables that is unblocked, then they are *d-connected*. If two variables are *d-connected*, then they are most likely dependent (except intransitive cases) (Pearl (1988)). The d-separation or conditional independence between the variables in the **key structures** (Figure 4) or building blocks of causal graphs follow some rules which are discussed below:

  i. *Conditional Independence in Chains:* If there is only one unidirectional path between variables $X$ and $Z$ (Figure 4 (a)), and $Y$ is any variable or set of variables that intercept that path, then $X$ and $Z$ are conditionally independent given $Y$, i.e. $X \perp\!\!\!\perp Z \mid Y$.

 ii. *Conditional Independence in Forks:* If a variable $X$ is a common cause of variables $Y$ and $Z$, and there is only one path between $Y$ and $Z$, then $Y$ and $Z$ are independent conditional on $X$ (i.e. $Y \perp\!\!\!\perp Z \mid X$) (Figure 4(b)).

iii. *Conditional Independence in Colliders:* If a variable $Z$ is the collision node between two variables $X$ and $Y$ (Figure 4(c)), and there is only one path between $X$ and $Y$, then $X$ and $Y$ are unconditionally independent (i.e. $X \perp\!\!\!\perp Y$). But, they become dependent when conditioned on $Z$ or any descendants of $Z$ (Figure 4(d)).

### 2.2.3   Markov Equivalence in Causal Graphs

A set of causal graphs having the same set of conditional independencies is known as a ***Markov equivalence class*** (MEC). Two DAGs that are Markov equivalent have the *(i) same skeleton* (the underlying undirected graph) and (ii) *same v-structures (colliders)* (Verma & Pearl (2022)). That is, all DAGs in a MEC share the same edges, regardless of the direction of those edges, and the same colliders whose parents are not adjacent. *Chain* and *Fork* share the same independencies, hence, they belong to the same MEC (Figure 5).

**Definition 5 (Markov Blanket)** *For any variable X, its Markov blanket (MB) is the set of variables such that X is independent of all other variables given MB. The **members** in the Markov blanket of any variable will include all of its **parents, children, and spouses**.*

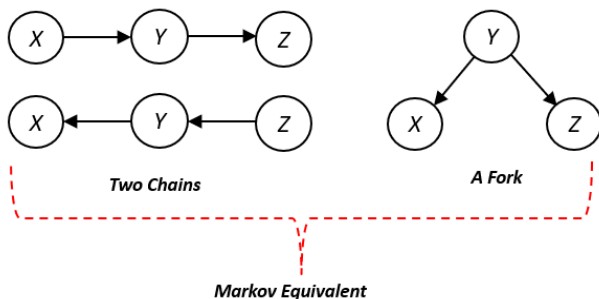

Figure 5: Markov Equivalence in Chains and Fork.

Markov equivalence in different types of DAGs may vary. A *partial* DAG (PDAG) a.k.a an essential graph (Perković et al. (2017)) can represent an equivalence class of DAGs. Each equivalent class of DAGs can be uniquely represented by a PDAG. A *completed* PDAG or CPDAG represents the union (over the set of edges) of Markov equivalent DAGs, and can uniquely represent an MEC (Malinsky & Spirtes (2016b)). More specifically, in a CPDAG, an undirected edge between any two nodes $X$ and $Y$ indicates that some DAG in the equivalence class contains the edge $X \rightarrow Y$ and some DAG may contain $Y \rightarrow X$. Figure 6 shows a CPDAG and the DAGs ($G$ and $H$) belonging to an equivalence class.

Markov equivalence in the case of ancestral graphs works as follows. A *maximal ancestral graph* (MAG) represents a DAG where all hidden variables are marginalized out and preserves all conditional independence relations among the variables that are true in the underlying DAG. That is, MAGs can model causality and conditional independencies in causally insufficient systems (Triantafillou & Tsamardinos (2016)). *Partial ancestral graphs* (PAGs) represent an equivalence class of MAGs where all common edge marks shared by all members in the class are displayed, and also, circles for those marks that are uncommon are presented. PAGs represent all of the observed d-separation relations in a DAG. Different PAGs that represent distinct equivalence classes of MAGs involve different sets of conditional independence constraints. An MEC of MAGs can be represented by a PAG (Malinsky & Spirtes (2016b)).

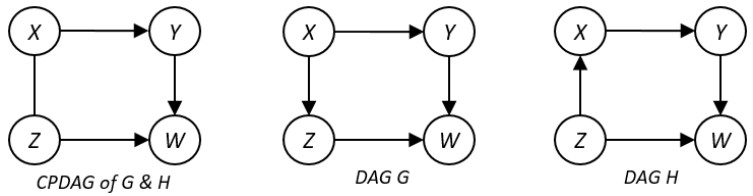

Figure 6: DAGs $G$ and $H$ belong to the same MEC. The leftmost graph is a CPDAG of $G$ and $H$ with an undirected edge ($-$) between $X$ and $Z$, and the rest of the edges same as in $G$ and $H$.

### 2.3 Structural Causal Models

Pearl (2009) defined a *class of models* for formalizing structural knowledge about the *data-generating process* known as the *structural causal models (SCMs)*. The SCMs are valuable tools for reasoning and decision-making in causal analysis since they are capable of representing the underlying causal story of data (Kaddour et al. (2022)).

**Definition 6 (Structural Causal Model)** *Pearl (2009); A structural causal model is a 4-tuple $M = \langle U, V, F, P(u) \rangle$, where*

  i. *$U$ is a set of background variables (also called exogenous) that are determined by factors outside the model.*

 ii. *$V$ is a set $\{V_1, V_2, \ldots, V_n\}$ of endogenous variables that are determined by variables in the model, viz. variables in $U \cup V$.*

iii. *$F$ is a set of functions $\{f_1, f_2, \ldots, f_n\}$ such that each $f_i$ is a mapping from the respective domains of $U_i \cup PA_i$ to $V_i$ and the entire set $F$ forms a mapping from $U$ to $V$. In other words, each $f_i$ assigns a value to the corresponding $V_i \in V$, $v_i \leftarrow f_i(pa_i, u_i)$, for $i = 1, 2, \ldots n$.*

 iv. *$P(u)$ is a probability function defined over the domain of $U$.*

Each SCM $M$ is associated with a *causal graphical model $G$* that is a DAG, and a set of functions $f_i$. *Causation in SCMs* can be interpreted as follows: a variable $Y$ is directly caused by $X$ if $X$ is in the function $f$ of $Y$. In other words, each $f_i$ assigns a value to the corresponding $V_i \in V$, $v_i \leftarrow f_i(pa_i, u_i)$, for $i = 1, 2, \ldots n$. In the SCM of Figure 7, $X$ is a direct cause of $Y$ as $X$ appears in the function that assigns $Y$'s value. That is, if a variable $Y$ is the child of another variable $X$, then $X$ is a direct cause of $Y$. In Figure 7, $U_X$, $U_Y$ and $U_Z$ are the exogenous variables; $X$, $Y$ and $Z$ are the endogenous variables, and $f_X$, $f_Y$ & $f_Z$ are the functions that assign values to the variables in the system. Any variable is *an exogenous variable* if ($i$) it is an unobserved or unmeasured variable and ($ii$) it cannot be a descendant of any other variables. Every *endogenous variable* is a descendant of at least one exogenous variable.

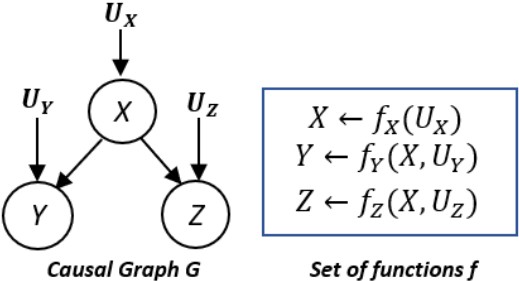

Figure 7: A Structural Causal Model (SCM) with causal graph $G$ and functions $f_X$, $f_Y$ and $f_Z$ which denotes how the variables $X$, $Y$, and $Z$ are generated respectively.

### 2.4 Causal Assumptions

Often, the available data provide only partial information about the underlying causal story. Hence, it is essential to make some assumptions about the world for performing causal discovery (Lee & Honavar (2020)). Following are the common assumptions usually made by causal discovery algorithms.

  i. ***Causal Markov Condition (CMC):*** The causal Markov assumption states that a variable $X$ is independent of every other variable (except its descendants) conditional on all of its direct causes (Scheines (1997)). That is, the CMC requires that every variable in the causal graph is independent of its non-descendants conditional on its parents (Malinsky & Spirtes (2016a)). In Figure 8, $W$ is the only descendant of $X$. As per the CMC, $X$ is independent of $Z$ conditioned on its parent $Y$ ($X \perp\!\!\!\perp Z \mid Y$).

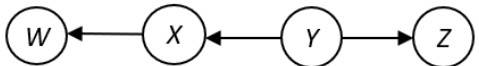

Figure 8: Illustration of the causal Markov condition (CMC) among four variables.

ii. ***Causal Faithfulness Condition (CFC):*** The faithfulness assumption states that except for the variables that are d-separated in a DAG, all other variables are dependent. More specifically, for a set of variables $V$ whose causal structure is represented by a DAG $G$, no conditional independence holds unless entailed by the causal Markov condition (Ramsey et al. (2012)). That is, the CFC a.k.a the Stability condition is a converse principle of the CMC. CFC can be also explained in terms of d-separation as follows: For every three disjoint sets of variables $X$, $Y$, and $Z$, if $X$ and $Y$ are not d-separated by $Z$ in the causal DAG, then $X$ and $Y$ are not independent conditioned on $Z$ (Ramsey et al. (2012)). The faithfulness assumption may fail in certain scenarios. For example, it fails whenever there exist two paths with equal and opposite effects between variables. It also fails in systems with deterministic relationships among variables, and also, when there is a failure of transitivity along a single path (Weinberger (2018)).

iii. ***Causal Sufficiency:*** The causal sufficiency assumption states that there exist no latent/hidden/unobserved confounders, and all the common causes are measured. Thus, the assumption of causal sufficiency is satisfied only when all the common causes of the measured variables are measured. This is a strong assumption as it restricts the search space of all possible DAGs that may be inferred. However, real-world datasets may have hidden confounders which might frequently cause the assumption to be violated in such scenarios. Algorithms that violate the causal sufficiency assumption may observe degradation in their performance. The causal insufficiency in real-world datasets may be overcome by leveraging domain knowledge in the discovery pipeline. The CMC tends to fail for a causally insufficient set of variables.

iv. ***Acyclicity:*** It is the most common assumption which states that *there are no cycles in a causal graph.* That is, a graph needs to be acyclic in order to be a causal graph. As per the acyclicity condition, there can be no directed paths starting from a node and ending back to itself. This resembles the structure of a directed acyclic graph (DAG). A recent approach (Zheng et al. (2018)) has formulated a new function (Equation 2) to enforce the acyclicity constraint during causal discovery in continuous optimization settings. The weighted adjacency matrix $W$ in Equation 2 is a DAG if it satisfies the following condition where $\circ$ is the Hadamard product, $e^{W \circ W}$ is the matrix exponential of $W \circ W$, and $d$ is the total number of vertices.

$$h(W) = tr(e^{W \circ W}) - d = 0 \tag{2}$$

v. ***Data Assumptions:*** There can be different types of assumptions about the data. Data may have linear or nonlinear dependencies and can be continuously valued or discrete valued in nature. Data can be independent and identically distributed (I.I.D.) or the data distribution may shift with time (e.g. time-series data). Also, the data may belong to different noise distributions such as Gaussian, Gumbel, or Exponential noise. Occasionally, some other data assumptions such as the existence of selection bias, missing variables, hidden confounders, etc. are found. However, in this survey, we do not focus much on the methods with these assumptions.

## 3  Causal Discovery Algorithms for I.I.D. Data

Causal graphs are essential as they represent the underlying causal story embedded in the data. There are two very common approaches to recovering the causal structure from observational data, *i) Constraint-based* (Spirtes et al. (2000b), Spirtes (2001), Colombo et al. (2012)) and *ii) Score-based* (Chickering (2002)). Among the other types of approaches, *functional causal models (FCMs)-based* (Shimizu et al. (2006), Hoyer et al. (2008)) approaches and *hybrid* approaches (Tsamardinos et al. (2006)) are noteworthy. Recently, some *gradient-based* approaches have been proposed based on neural networks (Abiodun et al. (2018)) and

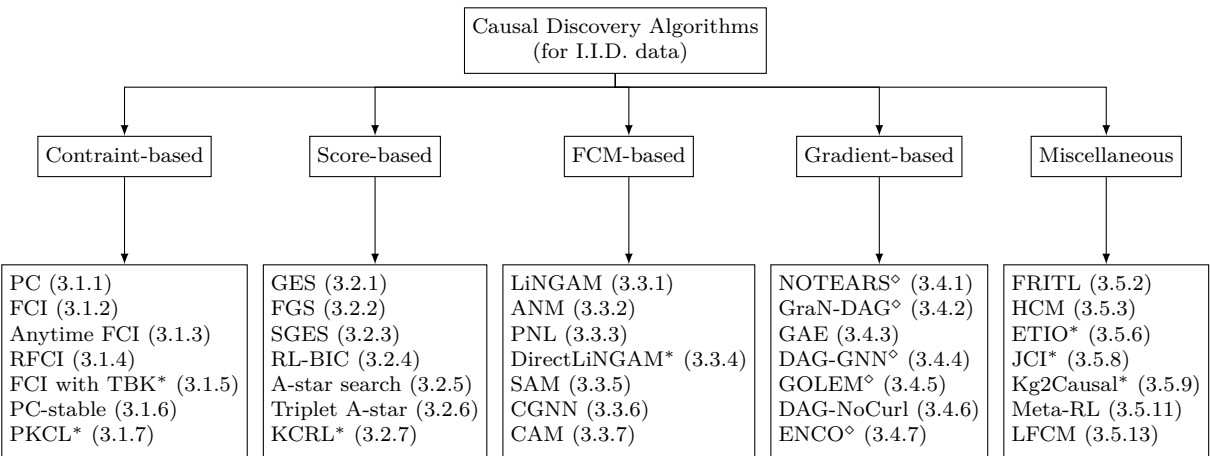

Figure 9: Taxonomy of some causal discovery approaches for I.I.D. data. The approaches are classified based on their core contribution or the primary strategy they adopt for causal structure recovery. The approaches that leverage prior knowledge are marked by an $*$ symbol. Some of the gradient-based optimization approaches that use a score function are indicated by a $\diamond$ symbol. They are primarily classified as gradient-based methods because of the use of gradient descent for optimization. However, they can be a score-based method too as they compute data likelihood scores on the way.

a modified definition (Equation 2) of the acyclicity constraint (Zheng et al. (2018), Yu et al. (2019)). Other approaches include the ones that prioritize the use of *background knowledge* and provides ways to incorporate prior knowledge and experts' opinion into the search process (Wang et al. (2020); Sinha & Ramsey (2021)). In this section, we provide an overview of the causal discovery algorithms for I.I.D. data based on the different types of approaches mentioned above. The algorithms primarily distinguish from each other based on the core approach they follow to perform causal discovery. We further discuss noteworthy similar approaches specialized for non-I.I.D. or time series data in section 4.

## 3.1 Constraint-based

Testing for conditional independence (CI) is a core objective of constraint-based causal discovery approaches. Conditional independence tests can be used to recover the causal skeleton if the probability distribution of the observed data is faithful to the underlying causal graph (Marx & Vreeken (2019)). Thus, constraint-based approaches conduct CI tests between the variables to check for the presence or absence of edges. These approaches infer the conditional independencies within the data using the *d-separation criterion* to search for a DAG that entails these independencies, and detect which variables are d-separated and which are d-connected (Triantafillou & Tsamardinos (2016)). $X$ is conditionally independent of $Z$ given $Y$ i.e. $X \perp\!\!\!\perp Z \mid Y$ in Figure 10 (a) and in Figure 10 (b), $X$ and $Z$ are independent, but are not conditionally independent given $Y$. Table 4 lists different types of CI tests used by constraint-based causal discovery approaches.

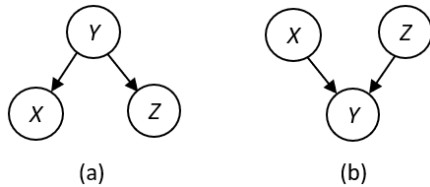

Figure 10: (a) $X \perp\!\!\!\perp Z \mid Y$ and (b) $X$ and $Z$ are not conditionally independent given $Y$.

Table 4: Types of conditional independence (CI) tests. Please refer to the study Runge (2018) for a detailed discussion on CI tests.

| | Conditional Independence Test | Ref. |
|---|---|---|
| 1. | Conditional Distance Correlation (CDC) test | Wang et al. (2015) |
| 2. | Momentary Conditional Independence (MCI) | Runge et al. (2019) |
| 3. | Kernel-based CI test (KCIT) | Zhang et al. (2012) |
| 4. | Randomized Conditional Correlation Test (RCoT) | Strobl et al. (2019) |
| 5. | Generative Conditional Independence Test (GCIT) | Bellot & van der Schaar (2019) |
| 6. | Model-Powered CI test | Sen et al. (2017) |
| 7. | Randomized Conditional Independence Test (RCIT) | Strobl et al. (2019) |
| 8. | Kernel Conditional Independence Permutation Test | Doran et al. (2014) |
| 9. | Gaussian Processes and Distance Correlation-based (GPDC) | Rasmussen et al. (2006) |
| 10. | Conditional mutual information estimated with a k-nearest neighbor estimator (CMIKnn) | Runge (2018) |

### 3.1.1 PC

The Peter-Clark (PC) algorithm (Spirtes et al. (2000b)) is one of the oldest constraint-based algorithms for causal discovery. To learn the underlying causal structure, this approach depends largely on conditional independence (CI) tests. This is because it is based on the concept that two statistically independent variables are not causally linked. The outcome of a PC algorithm is a CPDAG. It learns the CPDAG of the underlying DAG in three steps: *Step 1 - Skeleton identification, Step 2 - V-structures determination, and Step 3 - Edge orientations.* It starts with a fully connected undirected graph using every variable in the dataset, then eliminates the unconditionally and conditionally independent edges (skeleton detection), then it finds and orients the v-structures or colliders (i.e. X → Y ← Z) based on the d-separation set of node pairs, and finally orients the remaining edges based on two aspects: i) availability of no new v-structures, and ii) not allowing any cycle formation. The assumptions made by the PC algorithm include acyclicity, causal faithfulness, and causal sufficiency. It is computationally more feasible for sparse graphs. An implementation of this algorithm can be found in the CDT repository (`https://github.com/ElementAI/causal_discovery_toolbox`) and also, in the gCastle toolbox (Zhang et al. (2021a)). A number of the constraint-based approaches namely FCI, RFCI, PCMCI, PC-stable, etc. use the PC algorithm as a backbone to perform the CI tests.

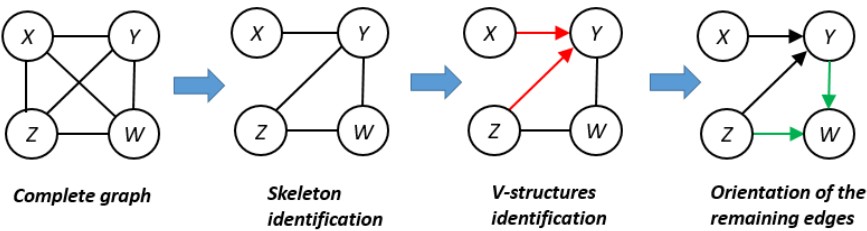

Figure 11: Step-by-step workflow of the PC (Spirtes et al. (2000b)) algorithm.

### 3.1.2 FCI

The Fast Causal Inference (FCI) algorithm (Spirtes et al. (2000a)) is a variant of the PC algorithm that can infer conditional independencies and learn causal relations in the presence of many arbitrary latent and selection variables. As a result, it is accurate in the large sample limit with a high probability even when there exists *hidden variables*, and *selection bias* (Berk (1983)). The first step of the FCI algorithm is similar to the PC algorithm where it starts with a complete undirected graph to perform the skeleton determination. After that, it requires additional tests to learn the correct skeleton and has additional orientation rules. In the worst case, the number of conditional independence tests performed by the algorithm grows exponentially with the number of variables in the dataset. This can affect both the speed and the accuracy of the algorithm in the

case of small data samples. To improve the algorithm, particularly in terms of speed, there exist different variants such as the RFCI (Colombo et al. (2012)) and the Anytime FCI (Spirtes (2001)) algorithms.

### 3.1.3   Anytime FCI

Anytime FCI (Spirtes (2001)) is a modified and faster version of the FCI (Spirtes et al. (2000a)) algorithm. The number of CI tests required by FCI makes it infeasible if the model has a large number of variables. Moreover, when the FCI requires independence tests conditional on a large set of variables, the accuracy decreases for a small sample size. The outer loop of the FCI algorithm performs independence tests conditional on the increasing size of variables. In the anytime FCI algorithm, the authors showed that this outer loop can be stopped anytime during the execution for any smaller variable size. As the number of variables in the conditional set reduces, anytime FCI becomes much faster for the large sample size. More importantly, it is also more reliable on limited samples since the statistical tests with the lowest power are discarded. To support the claim, the authors provided proof for the change in FCI that guarantees good results despite the interruption. The result of the interrupted anytime FCI algorithm is still valid, but as it cannot provide answers to most questions, the results could be less informative compared to the situation if it was allowed to run uninterrupted.

### 3.1.4   RFCI

Really Fast Causal Inference (RFCI) (Colombo et al. (2012)) is a much faster variant of the traditional FCI for learning PAGs that uses fewer CI tests than FCI. Unlike FCI, RFCI assumes that causal sufficiency holds. To ensure soundness, RFCI performs some additional tests before orienting v-structures and discriminating paths. It conditions only on subsets of the adjacency sets and unlike FCI, avoids the CI tests given subsets of possible d-separation sets which can become very large even for sparse graphs. As a result, the number of these additional tests and the size of their conditioning sets are small for sparse graphs which makes RFCI much faster and computationally feasible than FCI for high-dimensional sparse graphs. Also, the lower computational complexity of RFCI leads to high-dimensional consistency results under weaker conditions than FCI.

### 3.1.5   FCI with Tiered Background Knowledge

Andrews et al. (2020) show that the Fast Causal Inference (FCI) algorithm (Spirtes et al. (2000a)) is sound and complete with tiered background knowledge (TBK). By *tiered background knowledge*, it means any knowledge where the variables may be partitioned into two or more mutually exclusive and exhaustive subsets among which there is a known causal order. Tiered background knowledge may arise in many different situations, including but not limited to instrumental variables, data from multiple contexts and interventions, and temporal data with contemporaneous confounding. The proof that FCI is complete with TBK suggests that the algorithm is able to find all of the causal relationships that are identifiable from tiered background knowledge and observational data under the typical assumptions.

### 3.1.6   PC-stable

The independence tests in the original PC method are prone to errors in the presence of a few samples. Additionally, because the graph is updated dynamically, maintaining or deleting an edge incorrectly will affect the neighboring sets of other nodes. As a result, the sequence in which the CI tests are run will affect the output graph. Despite the fact that this order dependency is not a significant issue in low-dimensional situations, it is a severe problem in high-dimensional settings. To solve this problem, Colombo et al. (2014) suggested changing the original PC technique to produce a stable output skeleton that is independent of the input dataset's variable ordering. This approach, known as the stable-PC algorithm, queries and maintains the neighbor (adjacent) sets of every node at each distinct level. Since the conditioning sets of the other nodes are unaffected by an edge deletion at one level, the outcome is independent of the variable ordering. They demonstrated that this updated version greatly outperforms the original algorithm in high-dimensional settings while maintaining the original algorithms' low-dimensional settings performance. However, this

modification lengthens the algorithm's runtime even more by requiring additional CI checks to be done at each level. The R-package pcalg contains the source code for PC-stable.

### 3.1.7 PKCL

Wang et al. (2020) proposed an algorithm, **P**rior-**K**nowledge-driven Local **C**ausal Structure **L**earning (PKCL), to discover the underlying causal mechanism between *bone mineral density* (BMD) and its factors from clinical data. It first discovers the neighbors of the target variables and then detects the MaskingPCs to eliminate their effect. After that, it finds the spouse of target variables utilizing the neighbors set. This way the skeleton of the causal network is constructed. In the global stage, PKCL leverages the *Markov blanket (MB)* sets learned in the local stage to learn the global causal structure in which prior knowledge is incorporated to guide the global learning phase. Specifically, it learns the causal direction between feature variables and target variables by combining the constraint-based and score-based structure search methods. Also, in the learning phase, it automatically adds casual direction according to the available prior knowledge.

## 3.2 Score-based

Score-based causal discovery algorithms search over the space of all possible DAGs to find the graph that best explains the data. Typically, any score-based approach has two main components: *(i) a search strategy* to explore the possible search states or space of candidate graphs $G'$, *and (ii) a score function* to assess the candidate causal graphs. The search strategy along with a score function helps to optimize the search over the space of all possible DAGs. More specifically, a score function $S(G', D)$ maps causal graphs $G'$ to a numerical score, based on how well $G'$ fits a given dataset $D$. A commonly used score function to select causal models is the *Bayesian Information Criterion (BIC)* (Schwarz (1978a)) which is defined below:

$$\mathcal{S}(G', D) = -2\log \mathcal{L}\{G', D\} + k\log n, \tag{3}$$

where $n$ is the number of samples in $D$, $k$ is the dimension of $G'$ and $\mathcal{L}$ is the maximum-likelihood function associated with the candidate graph $G'$. The lower the BIC score, the better the model. BDeu, BGe, MDL, etc. (listed in Table 5) are some of the other commonly used score functions. These objective functions are optimized through a heuristic search for model selection. After evaluating the quality of the candidate causal graphs using the score function, the score-based methods output one or more causal graphs that achieve the highest score (Huang et al. (2018b)). We discuss some of the well-known approaches in this category below.

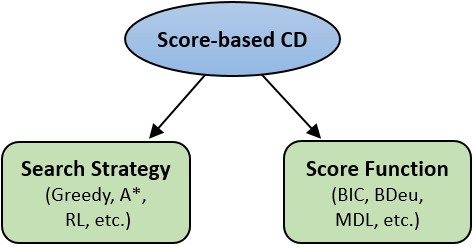

Figure 12: General components of a score-based causal discovery approach.

### 3.2.1 GES

Greedy Equivalence Search (GES) (Chickering (2002)) is one of the oldest score-based causal discovery algorithms that perform a greedy search over the space of equivalence classes of DAGs. Each search state is represented by a CPDAG where some insert and delete operators allow for single-edge additions and deletions respectively. Primarily GES works in two phases: i) Forwards Equivalence Search (FES), and ii) Backward Equivalence Search (BES). In the first phase, FES starts with an empty CPDAG (no-edge model), and greedily adds edges by taking into account every single-edge addition that could be performed to every DAG in the current equivalence class. After an edge modification is done to the current CPDAG, a score function is used to score the model. If the new score is better than the current score, only then the

modification is allowed. When the forward phase reaches a local maximum, the second phase, BES starts where at each step, it takes into account all single-edge deletions that might be allowed for all DAGs in the current equivalence class. The algorithm terminates once the local maximum is found in the second phase. Implementation of GES is available at the following Python packages: Causal Discovery Toolbox or CDT (Kalainathan & Goudet (2019)) and gCastle (Zhang et al. (2021a)). GES assumes that the score function is decomposable and can be expressed as a sum of the scores of individual nodes and their parents. A summary workflow of GES is shown in Figure 13.

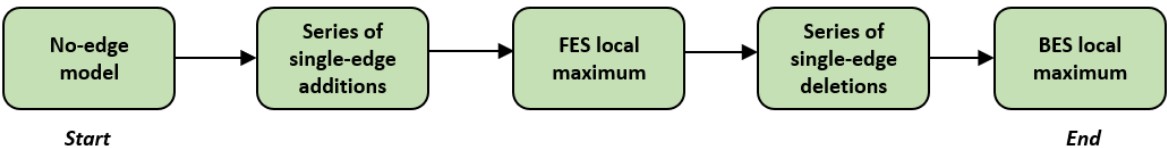

Figure 13: Different stages in the GES algorithm.

### 3.2.2 FGS

Fast Greedy Search (FGS) (Ramsey (2015)) is another score-based method that is an optimized version of the GES algorithm (Chickering (2002)). This optimized algorithm is based on the faithfulness assumption and uses an alternative method to reduce scoring redundancy. An ascending list $L$ is introduced which stores the score difference of arrows. After making a thorough search, the first edge e.g. $X \to Y$ is inserted into the graph and the graph pattern is reverted. For variables that are adjacent to $X$ or $Y$ with positive score differences, new edges are added to $L$. This process in the forward phase repeats until the $L$ becomes empty. Then the reverse phase starts, filling the list $L$ and continuing until $L$ is empty. This study considered the experiment where GES was able to search over 1000 samples with 50,000 variables in 13 minutes using a 4-core processor and 16GB RAM computer. Following the new scoring method, FGS was able to complete the task with 1000 samples on 1,000,000 variables for sparse models in 18 hours using a supercomputer having 40 processors and 384GB RAM at the Pittsburgh Supercomputing Center. The code for FGS is available on GitHub as a part of the Tetrad project: `https://github.com/cmu-phil/tetrad`.

### 3.2.3 SGES

Selective Greedy Equivalence Search (SGES) (Chickering & Meek (2015)) is another score-based causal discovery algorithm that is a restrictive variant of the GES algorithm (Chickering (2002)). By assuming perfect generative distribution, SGES provides a polynomial performance guarantee yet maintains the asymptotic accuracy of GES. While doing this, it is possible to keep the algorithm's large sample guarantees by ignoring all but a small fraction of the backward search operators that GES considered. In the forward phase, SGES uses a polynomial number of insert operation calls to the score function. In the backward phase, it consists of only a subset of delete operators of GES which include, consistent operators to preserve GES's consistency over large samples. The authors demonstrated that, for a given set of graph-theoretic complexity features,

Table 5: Some commonly used score functions for causal discovery. Please refer to the study Huang et al. (2018a) for a detailed discussion of the score functions.

| Score Function/Criterion | Ref. |
|---|---|
| Minimum description length (MDL) | Schwarz (1978b) |
| Bayesian information criterion (BIC) | Schwarz (1978a) |
| Akaike information criterion (AIC) | Akaike (1998) |
| Bayesian Dirichlet equivalence score (BDeU) | Buntine (1991) |
| Bayesian metric for Gaussian networks (BGe) | Geiger & Heckerman (1994) |
| Factorized normalized maximum likelihood (fNML) | Silander et al. (2008) |

such as maximum-clique size, the maximum number of parents, and v-width, the number of score assessments by SGES can be polynomial in the number of nodes and exponential in these complexity measurements.

### 3.2.4 RL-BIC

RL-BIC is a score-based approach that uses *Reinforcement Learning (RL)* and a BIC score to search for the DAG with the best reward (Zhu et al. (2019)). For data-to-graph conversion, it uses an *encoder-decoder architecture* that takes observational data as input and generates graph adjacency matrices that are used to compute rewards. The reward incorporates a BIC score function and two penalty terms for enforcing acyclicity. The *actor-critic RL algorithm* is used as a *search strategy* and the final output is the causal graph that achieves the best reward among all the generated graphs. The approach is applicable to small and medium graphs of up to 30 nodes. However, dealing with large and very large graphs is still a challenge for it. This study mentions that their future work involves developing a more efficient and effective score function since computing scores is much more time-consuming than training NNs. The original implementation of the approach is available at: `https://github.com/huawei-noah/trustworthyAI`.

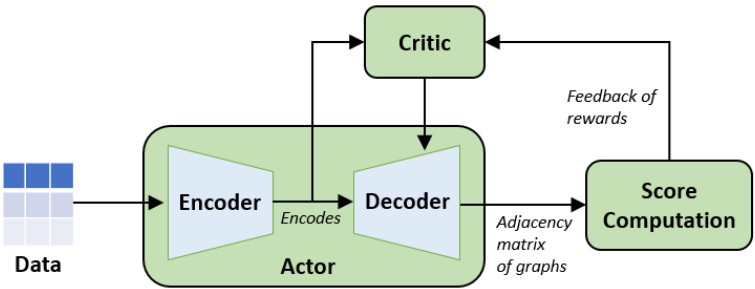

Figure 14: Components of the RL-BIC (Zhu et al. (2019)) approach.

### 3.2.5 A* search

Xiang & Kim (2013) proposed a one-stage method for learning sparse network structures with continuous variables using the A* search algorithm with lasso in its scoring system. This method increased the computational effectiveness of popular exact methods based on dynamic programming. The study demonstrated how the proposed approach achieved comparable or better accuracy with significantly faster computation time when compared to two-stage approaches, including L1MB and SBN. Along with that, a heuristic approach was added that increased A* lasso's effectiveness while maintaining the accuracy of the outcomes. In high-dimensional spaces, this is a promising approach for learning sparse Bayesian networks.

### 3.2.6 Triplet A*

Lu et al. (2021) uses the *A* exhaustive search* (Yuan & Malone (2013)) combined with an optimal BIC score that requires milder assumptions on data than conventional CD approaches to guarantee its asymptotic correctness. The optimal BIC score combined with the exhaustive search finds the MEC of the true DAG if and only if the true DAG satisfies the optimal BIC Condition. To gain scalability, they also developed an approximation algorithm for complex large systems based on the A* method. This extended approach is named Triplet A* which can scale up to more than 60 variables. This extended method is rather general and can be used to scale up other exhaustive search approaches as well. Triplet A* can particularly handle linear Gaussian and non-Gaussian networks. It works in the following way. Initially, it makes a guess about the parents and children of each variable. Then for each variable $X$ and its neighbors $(Y, Z)$, it forms a cluster consisting of $X, Y, Z$ with their direct neighbors and runs an exhaustive search on each cluster. Lastly, it combines the results from all clusters. The study shows that empirically Triplet A* outperforms GES for large dense networks.

### 3.2.7 KCRL

Prior **K**nowledge-based **C**ausal Discovery Framework with **R**einforcement **L**earning a.k.a. KCRL (Hasan & Gani (2022)) is a framework for causal discovery that utilizes prior knowledge as constraints and penalizes the search process for violation of these constraints. This utilization of background knowledge significantly improves performance by reducing the search space, and also, enabling a faster convergence to the optimal causal structure. KCRL leverages reinforcement learning (RL) as the search strategy where the RL agent is penalized each time for the violation of any imposed knowledge constraints. In the KCRL framework (Figure 15), at first, the observational data is fed to an RL agent. Here, data-to-adjacency matrix conversion is done using an encoder-decoder architecture which is a part of the RL agent. At every iteration, the agent produces an equivalent adjacency matrix of the causal graph. A comparator compares the generated adjacency matrix with the true causal edges in the prior knowledge matrix $P_m$, and thereby, computes a penalty $p$ for the violation of any ground truth edges in the produced graph. Each generated graph is also scored using a standard scoring function such as BIC. A reward $R$ is estimated as a sum of the BIC score $S_{BIC}$, the penalty for acyclicity $h(W)$, and $\beta$ weighted prior knowledge penalty $\beta p$. Finally, the entire process halts when the stopping criterion $S_c$ is reached, and the best-rewarded graph is the final output causal graph. Although originally KCRL was designed for the healthcare domain, it can be used in any other domain for causal discovery where some prior knowledge is available. Code for KCRL is available at `https://github.com/UzmaHasan/KCRL`.

$$R = S_{BIC} + \beta p + h(W) \tag{4}$$

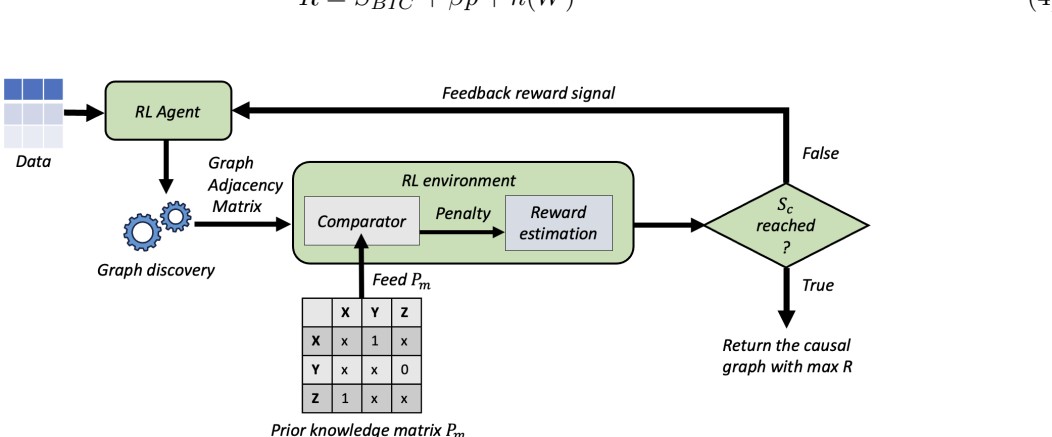

Figure 15: The KCRL (Hasan & Gani (2022)) framework.

Another recent method called **KGS** (Hasan & Gani (2023)) leverages prior causal information such as the presence or absence of a causal edge to guide a greedy score-based causal discovery process towards a more restricted and accurate search space. It demonstrates how the search space as well as scoring candidate graphs can be reduced when different edge constraints are leveraged during a search over equivalence classes of causal networks. It concludes that any type of edge information is useful to improve the accuracy of the graph discovery as well as the run time.

### 3.2.8 ILP-based structure learning

Bartlett & Cussens (2017) looked into the application of integer linear programming (ILP) to the structure learning problem. To boost the effectiveness of ILP-based Bayesian network learning, they suggested adding auxiliary implied constraints. Experiments were conducted to determine the effect of each constraint on the optimization process. It was discovered that the most effective configuration of these constraints could significantly boost the effectiveness and speed of ILP-based Bayesian network learning. The study made a significant contribution to the field of structure learning and showed how well ILP can perform under non-essential constraints.

### 3.3 Functional Causal Model-based

Functional Causal Model (FCM) based approaches describe the causal relationship between variables in a specific functional form. FCMs represent variables as a function of their parents (direct causes) together with an independent noise term $E$ (see Equation 5) (Zhang et al. (2015)). FCM-based methods can distinguish among different DAGs in the same equivalence class by imposing additional assumptions on the data distributions and/or function classes (Zhang et al. (2021b)). Some of the noteworthy FCM-based causal discovery approaches are listed below.

$$X = f(PA_X) + E \tag{5}$$

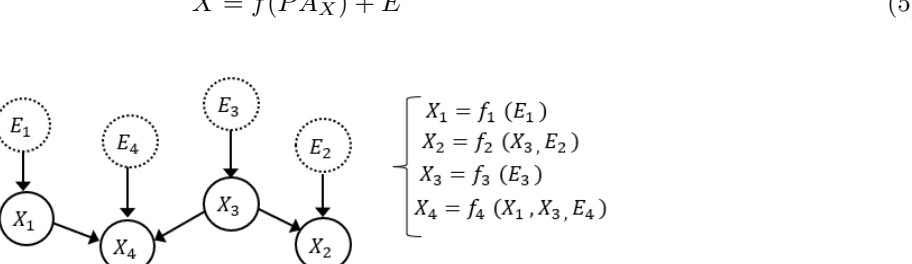

Figure 16: A functional causal model (FCM) with four variables.

#### 3.3.1 L*i*NGAM

Linear Non-Gaussian Acyclic Model (LiNGAM) aims to discover the causal structure from observational data under the assumptions that the data generating process is linear, there are no unobserved confounders, and noises have non-Gaussian distributions with non-zero variances (Shimizu et al. (2006)). It uses the statistical method known as independent component analysis (ICA) (Comon (1994)), and states that when the assumption of **non-Gaussianity** is valid, the complete causal structure can be estimated. That is, the causal direction is identifiable if the variables have a linear relation, and the noise ($\varepsilon$) distribution is non-Gaussian in nature. Figure 17 depicts three scenarios where when $X$ and $\varepsilon$ are Gaussian (case 1), the predictor and regression residuals are independent of each other. For the other two cases, $X$ and $\varepsilon$ are non-Gaussian, and we see that for the regression in the anti-causal or backward direction ($X$ given $Y$), the regression residual and the predictor are not independent as earlier. That is, for the non-Gaussian cases, independence between regression residual and predictor occurs only for the correct causal direction. There are 3 properties of a LiNGAM. *First*, the variables $x_i = x_1, x_2, ..., x_n$ are arranged in a causal order $k(i)$ such that the cause always preceedes the effect. *Second*, each variable $x_i$ is assigned a value as per the Equation 6 where $e_i$ is the noise/disturbance term and $b_{ij}$ denotes the causal strength between $x_i$ and $x_j$. *Third*, the exogenous noise $e_i$ follows a non-Gaussian distribution, with zero mean and non-zero variance, and are independent of each other which implies that there is no hidden confounder. Python implementation of the LiNGAM algorithm is available at `https://github.com/cdt15/lingam` as well as in the gCastle package (Zhang et al. (2021b)). Any standard ICA algorithm which can estimate independent components of many different distributions can be used in LiNGAM. However, the original implementation uses the FastICA (Hyvarinen (1999)) algorithm.

$$x_i = \sum_{k(j)<k(i)} b_{ij}x_j + e_i \tag{6}$$

#### 3.3.2 ANM

Hoyer et al. (2008) performs causal discovery with additive noise models (ANMs) and provides a generalization of the linear non-Gaussian causal discovery framework to deal with nonlinear functional dependencies where the variables have an additive noise. It mentions that nonlinear causal relationships typically help to break the symmetry between the observed variables and help in the identification of causal directions. ANM assumes that the data generating process of the observed variables is as per the Equation 7 where a

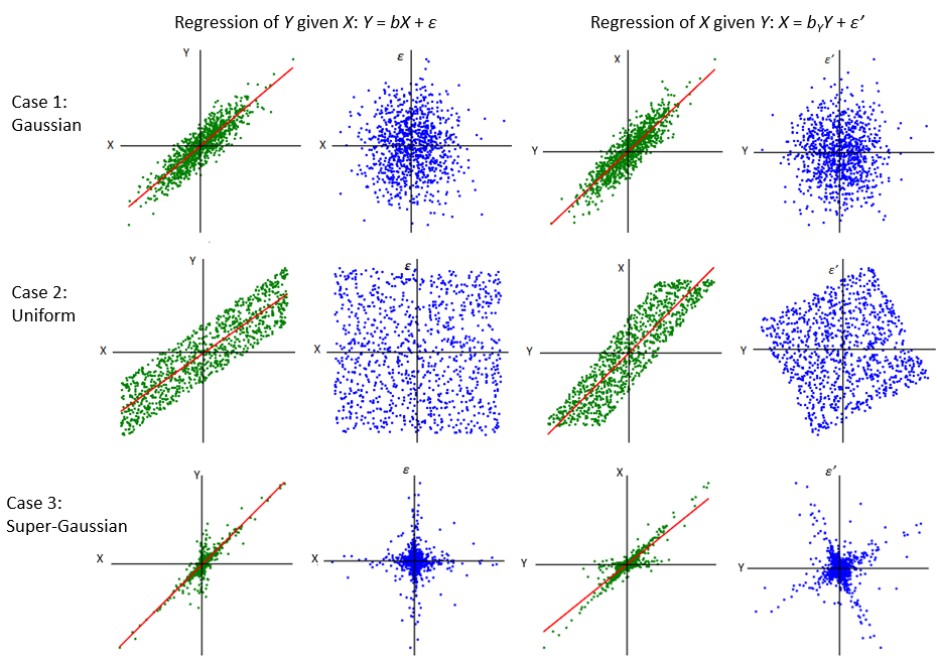

Figure 17: Causal asymmetry between two variables having a linear relation (Glymour et al. (2019)). Here, the causal direction is from $X$ to $Y$. A total of three scenarios are depicted where both $X$ and $\varepsilon$ follow the i) Gaussian, ii) Uniform, or iii) Super-Gaussian distribution for each of the scenarios.

variable $x_i$ is a function of its parents and the noise term $e_i$ which is an independent additive noise. An implementation of ANM is available in the gCastle package (Zhang et al. (2021a)).

$$x_i = f(PA_{x_i}) + e_i \tag{7}$$

### 3.3.3 PNL

Post-nonlinear (PNL) acyclic causal model with additive noise (Zhang & Hyvärinen (2010)) is a highly realistic model where each observed continuous variable is made up of additive noise-filled nonlinear functions of its parents, followed by a nonlinear distortion. The influence of sensor distortions, which are frequently seen in practice, is taken into account by the second stage's nonlinearity. A two-step strategy is proposed to separate the cause from the effect in a two-variable situation, consisting of restricted nonlinear ICA followed by statistical independence tests. The PNL model was able to effectively separate causes from effects when applied to solve the "CauseEffectPairs" task proposed by Mooij & Janzing (2010) in the Pot-luck challenge. That is, it successfully distinguished the cause from the effect, even if the nonlinear function of the cause is not invertible.

### 3.3.4 Direct-L$i$NGAM

Shimizu et al. (2011) proposed DirectLiNGAM, a direct method for learning a linear non-Gaussian structural equation model (SEM) which is a direct method to estimate causal ordering and connection strengths based on non-Gaussianity. This approach estimates a causal order of variables by successively reducing each independent component from given data in the model which is completed in steps equal to the number of the variables in the model. Once the causal order of variables is identified, their connection strengths are estimated using conventional covariance-based methods such as least squares and maximum likelihood approaches. If the data strictly follows the model i.e. if all the model assumptions are met and the sample size is infinite, it converges to the right solution within a small number of steps. If some prior knowledge on a part of the structure is available, it suggests using those for more efficient learning. Doing so will reduce

the number of causal orders and connection strengths to be estimated. Its implementation can be found at: `https://github.com/huawei-noah/trustworthyAI/tree/master/gcastle`.

### 3.3.5 SAM

Kalainathan et al. (2018) proposed the algorithm known as *Structural Agnostic Modeling* (SAM) that uses an *adversarial learning* approach to find the causal graphs. Particularly, it searches for an FCM using *Generative Adversarial Neural-networks (GANs)* and enforces the discovery of sparse causal graphs through adequate regularization terms. A learning criterion that combines distribution estimation, sparsity, and acyclicity constraints is used to enforce the end-to-end optimization of the graph structure and parameters through stochastic gradient descent. SAM leverages both conditional independencies and distributional asymmetries in the data to find the underlying causal mechanism. It aims to achieve an optimal complexity/fit trade-off while modeling the causal mechanisms. SAM enforces the acyclicity constraint of a DAG using the function in Equation 8 where, $A$ is the adjacency matrix of the ground-truth graph $G$, and $d$ denotes the total number of nodes in $G$. The latest implementation of SAM is available in the CDT package (Kalainathan & Goudet (2019)). Also, an older version of SAM is available at `https://github.com/Diviyan-Kalainathan/SAM`.

$$\sum_{i=1}^{d} = \frac{\text{tr}(A^i)}{i!} = 0 \tag{8}$$

### 3.3.6 CGNN

**C**ausal **G**enerative **N**eural **N**etworks (CGNN) is an FCM-based framework that uses *neural networks (NNs)* to learn the joint distribution of the observed variables (Goudet et al. (2018)). Particularly, it uses a generative model that minimizes the *maximum mean discrepancy* (MMD) between the generated and observed data. CGNN has a high computational cost. However, it proposes an approximate learning criterion to scale the computational cost to linear complexity in the number of observations. This framework can also be used to simulate interventions on multiple variables in the dataset. An implementation of CGNN in Pytorch is available at `https://github.com/FenTechSolutions/CausalDiscoveryToolbox`.

### 3.3.7 CAM

**C**ausal **A**dditive **M**odel (CAM) is a method for estimating high-dimensional additive structural equation models which are logical extensions of linear structural equation models (Bühlmann et al. (2014)). In order to address the difficulties of computation and statistical accuracy in the absence of prior knowledge about underlying structure, the authors established consistency of the maximum likelihood estimator and developed an effective computational algorithm. The technique was demonstrated using both simulated and actual data and made use of tools in sparse regression techniques. The authors also discussed identifiability problems and the enormous size of the space of potential models, which presents significant computational and statistical accuracy challenges.

### 3.3.8 CAREFL

Causal Autoregressive Flows (CAREFL) uses *autoregressive flow models* (Huang et al. (2018c)) for causal discovery by interpreting the ordering of variables in an autoregressive flow based on structural equation models (SEMs) (Khemakhem et al. (2021)). In general, SEMs define a generative model for data based on causal relationships. CAREFL shows that particularly *affine flows* define a new class of causal models where the noise is modulated by the cause. For such models, it proves a new causal identifiability result that generalizes additive noise models. To learn the causal structure efficiently, it selects the ordering with the highest test log-likelihood and reports a measure of causal direction based on the likelihood ratio for non-linear SEMs. Autoregressive flow models also enable CAREFL to evaluate interventional queries by fixing the interventional variable while sampling from the flow. Moreover, the invertible property of autoregressive flows facilitates counterfactual queries as well. Code implementation of CAREFL is available at `https://github.com/piomonti/carefl`.

### 3.4 Gradient-based

Some of the recent studies in causal discovery formulate the structure learning problem as a continuous optimization task using the least squares objective and an algebraic characterization of DAGs (Zheng et al. (2018), Ng et al. (2020)). Specifically, the combinatorial structure learning problem has been transformed into a continuous one and solved using gradient-based optimization methods (Ng et al. (2019)). These methods leverage gradients of an objective function with respect to a parametrization of a DAG matrix. Apart from the usage of well-studied gradient-based solvers, they also leverage GPU acceleration which has changed the nature of the task (Ng et al. (2020)). Furthermore, to accelerate the task they often employ deep learning models that are capable of capturing complex nonlinear mappings (Yu et al. (2019)). As a result, they usually have a faster training time as deep learning is known to be highly parallelizable on GPU, which gives a promising direction for causal discovery with gradient-based methods (Ng et al. (2019)). In general, these methods are more global than other approximate greedy methods. This is because they update all edges at each step based on the gradient of the score and as well as based on the acyclicity constraint.

#### 3.4.1 NOTEARS

DAGs with NO TEARS (Zheng et al. (2018)) is a recent breakthrough in the field of causal discovery that formulates the structure learning problem as a purely continuous constrained optimization task. It leverages an algebraic characterization of DAGs and provides a novel characterization of acyclicity that allows for a smooth global search, in contrast to a combinatorial local search. The full form of the acronym NOTEARS is **N**on-combinatorial **O**ptimization via **T**race **E**xponential and **A**ugmented lag**R**angian for **S**tructure learning which particularly handles linear DAGs. It assumes a linear dependence between random variables and thus models data $D$ as a structural equation model. To discover the causal structure, it imposes the proposed acyclicity function (Equation 10) as a constraint combined with a weighted adjacency matrix $W$ with least squares loss. The algorithm aims to convert the traditional combinatorial optimization problem into a continuous constrained optimization task by leveraging an algebraic characterization of DAGs via the trace exponential acyclicity function as follows:

$$\min_{\substack{W \in \mathbb{R}^{d \times d} \\ \text{subject to } G(W) \in DAGs}} F(W) \quad \Longleftrightarrow \quad \min_{\substack{W \in \mathbb{R}^{d \times d} \\ \text{subject to } h(W)=0}} F(W) \ , \tag{9}$$

where $G(W)$ is a graph with $d$ nodes induced by the weighted adjacency matrix $W$, $F : \mathbb{R}^{d \times d} \to \mathbb{R}$ is a regularized score function with a least-square loss $\ell$, and $h : \mathbb{R}^{d \times d} \to \mathbb{R}$ is a smooth function over real matrices that enforces acyclicity. Overall, the approach is simple and can be executed in about 50 lines of Python code. Its implementation in Python is publicly available at `https://github.com/xunzheng/notears`. The acyclicity function proposed in NOTEARS is as follows where $\circ$ is the Hadamard product and $e^{W \circ W}$ is the matrix exponential of $W \circ W$.

$$h(W) = \text{tr}(e^{W \circ W}) - d = 0 \tag{10}$$

#### 3.4.2 GraN-DAG

Gradient-based Neural DAG Learning (GraN-DAG) is a causal structure learning approach that uses *neural networks (NNs)* to deal with non-linear causal relationships (Lachapelle et al. (2019)). It uses a stochastic gradient method to train the NNs to improve scalability and allow implicit regularization. It formulates a *novel characterization of acyclicity* for NNs based on NOTEARS (Zheng et al. (2018)). To ensure acyclicity in non-linear models, it uses an argument similar to NOTEARS and applies it first at the level of neural network paths and then at the graph paths level. For regularization, GraN-DAG uses a procedure called *preliminary neighbors selection* (PNS) to select a set of potential parents for each variable. It uses a final pruning step to remove the false edges. The algorithm works well mostly in the case of non-linear Gaussian additive noise models. An implementation of GraN-DAG can be found at `https://github.com/kurowasan/GraN-DAG`.

### 3.4.3 GAE

**G**raph **A**uto**e**ncoder (GAE) approach is a gradient-based approach to causal structure learning that uses a *graph autoencoder framework* to handle nonlinear structural equation models (Ng et al. (2019)). GAE is a *special case* of the causal additive model (CAM) that provides an alternative generalization of NOTEARS for handling nonlinear causal relationships. GAE is easily applicable to vector-valued variables. The architecture of GAE consists of a variable-wise encoder and decoder which are basically multi-layer perceptrons (MLPs) with shared weights across all variables $X_i$. The encoder-decoder framework allows the reconstruction of each variable $X_i$ to handle the nonlinear relations. The final goal is to optimize the reconstruction error of the GAE with $l_1$ penalty where the optimization problem is solved using the augmented Lagrangian method (Nemirovsky (1999)). The approach is competitive in terms of scalability as it has a near-linear training time when scaling up the graph size to 100 nodes. Also, in terms of time efficiency, GAE performs well with an average training time of fewer than 2 minutes even for graphs of 100 nodes. Its implementation can be found at the gCastle (Zhang et al. (2021a)) repository.

### 3.4.4 DAG-GNN

DAG Structure Learning with Graph Neural Networks (DAG-GNN) is a graph-based deep generative model that tries to capture the sampling distribution faithful to the ground-truth DAG (Yu et al. (2019)). It leverages variational inference and a parameterized pair of *encoder-decoders* with specially designed *graph neural networks (GNN)*. Particularly, it uses *Variational Autoencoders (VAEs)* to capture complex data distributions and sample from them. The weighted adjacency matrix $W$ of the ground-truth DAG is a learnable parameter with other neural network parameters. The VAE model naturally handles various data types both continuous and discrete in nature. In this study, the authors also propose a *variant of the acyclicity function* (Equation 11) which is more suitable and practically convenient for implementation with the existing deep learning methods. In the acyclicity function, $d =$ the number of nodes, $\alpha$ is a hyperparameter, and $I$ is an identity matrix. An implementation of the DAG-GNN algorithm is available at `https://github.com/fishmoon1234/DAG-GNN`.

$$\text{tr}[(I + \alpha W \circ W)^d] - d = 0 \tag{11}$$

### 3.4.5 GOLEM

Gradient-based Optimization of DAG-penalized Likelihood for learning linear DAG Models (GOLEM) is a *likelihood-based* causal structure learning approach with *continuous unconstrained optimization* (Ng et al. (2020)). It studies the asymptotic role of the sparsity and DAG constraints for learning DAGs in both linear Gaussian and non-Gaussian cases. It shows that when the optimization problem is formulated using a likelihood-based objective instead of least squares (used by NOTEARS), then instead of a hard DAG constraint, applying only soft sparsity and DAG constraints is enough for learning the true DAG under mild assumptions. Particularly, GOLEM tries to optimize the score function in Equation 12 w.r.t. the weighted adjacency matrix $B$ representing a directed graph. Here, $L(B;x)$ is the maximum likelihood estimator, $R_{sparse}(B)$ is a penalty to encourage sparsity (i.e. fewer edges), and $R_{DAG}(B)$ is the penalty that enforces DAGness on $B$.

$$S(B;x) = L(B;x) + R_{sparse}(B) + R_{DAG}(B) \tag{12}$$

In terms of denser graphs, GOLEM seems to outperform NOTEARS since it can reduce the number of optimization iterations which makes it robust in terms of scalability. With gradient-based optimization and GPU acceleration, it can easily handle thousands of nodes while retaining high accuracy. An implementation of GOLEM can be found at the gCastle (Zhang et al. (2021a)) repository.

### 3.4.6 DAG-NoCurl

DAG-NoCurl also known as DAGs with No Curl uses a two-step procedure for the causal DAG search (Yu et al. (2021)). At first, it finds an initial cyclic solution to the optimization problem and then employs the

*Hodge decomposition* (Bhatia et al. (2012)) of graphs to learn an acyclic graph by projecting the cyclic graph to the gradient of a potential function. The goal of this study is to investigate how the causal structure can be learned without any explicit DAG constraints by directly optimizing the DAG space. To do so, it proposes the method DAG-NoCurl based on the graph Hodge theory that implicitly enforces the acyclicity of the learned graph. As per the Hodge theory on graphs (Lim (2020)), a DAG is a sum of three components: *a curl-free, a divergence-free,* and *a harmonic component.* The curl-free component is an acyclic graph that motivates the naming of this approach. An implementation of the method can be found at the link `https://github.com/fishmoon1234/DAG-NoCurl`.

### 3.4.7   ENCO

Efficient Neural Causal Discovery without Acyclicity Constraints (ENCO) *uses both observational and interventional data* by modeling a probability for every possible directed edge between pairs of variables (Lippe et al. (2021)). It formulates the graph search as an optimization of independent edge likelihoods, with the edge orientation being modeled as a separate parameter. This approach guarantees convergence when interventions on all variables are available and do not require explicitly constraining the score function with respect to acyclicity. However, the algorithm works on partial intervention sets as well. Experimental results suggest that ENCO is robust in terms of scalability, and *is able to detect latent confounders.* When applied to large networks having 1000 nodes, it is capable of recovering the underlying structure due to the benefit of its low-variance gradient estimators. The source code of ENCO is available at this site: `https://github.com/phlippe/ENCO`.

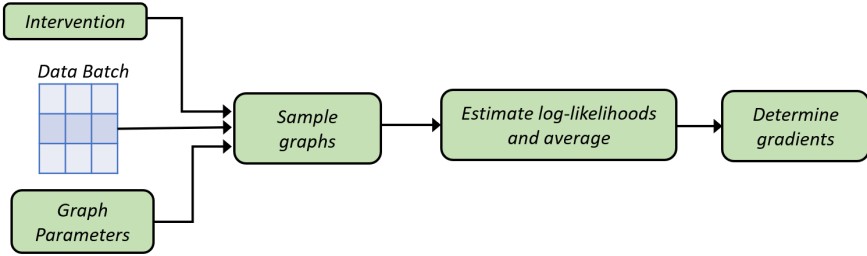

Figure 18: Graph optimization mechanism of ENCO.

### 3.4.8   MCSL

**M**asked Gradient-based **C**ausal **S**tructure **L**earning (MCSL) (Ng et al. (2022)) utilizes a reformulated structural equation model (SEM) for causal discovery using gradient-based optimization that leverages the *Gumbel-Softmax approach* (Jang et al. (2016)). This approach is used to approximate a binary adjacency matrix and is often used to approximate samples from a categorical distribution. MCSL reformulates the SEM with additive noises in a form parameterized by the binary graph adjacency matrix. It states that, if the original SEM is identifiable, then the adjacency matrix can be identified up to super-graphs of the true causal graph under some mild conditions. For experimentation, MCSL uses multi-layer perceptrons (MLPs), particularly having 4-layers as the model function which is denoted as MCSL-MLP. An implementation of the approach can be found in the gCastle (Zhang et al. (2021a)) package.

### 3.4.9   DAGs with No Fears

Wei et al. (2020) provides an in-depth analysis of the NOTEARS framework for causal structure learning. The study proposed a local search post-processing algorithm that significantly increased the precision of NOTEARS and other algorithms and deduced Karush-Kuhn-Tucker (KKT) optimality conditions for an equivalent reformulation of the NOTEARS problem. Additionally, the authors compared the effectiveness of NOTEARS and Abs-KKTS on various graph types and discovered that Abs-KKTS performed better than NOTEARS in terms of accuracy and computational efficiency. The authors concluded that this work improved the understanding of optimization-based causal structure learning and may result in

further advancements in precision and computational effectiveness. The code implementation is available at `https://github.com/skypea/DAG_No_Fear`.

### 3.5 Miscellaneous Approaches

Apart from the types of approaches mentioned so far, there are some other causal discovery approaches that use some specialized or unique techniques to search for the graph that best describes the data. There also exists some methods that are specialized to handle latent or unobserved confounders. Also, there are some approaches that are hybrid in nature, i.e. they are based on the combination of constraint-based, score-based, FCM-based, gradient-based, etc. causal discovery approaches. For example, some approaches integrate conditional independence testing along with score functions to design a hybrid approach for causal discovery. A detailed discussion can be found below.

#### 3.5.1 MMHC

**M**ax-**M**in **H**ill **C**limbing (MMHC) is a hybrid causal discovery technique that incorporates the concepts from both score-based and constraint-based algorithms (Tsamardinos et al. (2006)). A challenge in causal discovery is the identification of causal relationships within a reasonable time in the presence of thousands of variables. MMHC can reliably learn the causal structure in terms of time and quality for high-dimensional settings. MMHC is a two-phase algorithm that assumes faithfulness. In the first phase, MMHC uses Max-Min Parents and Children (MMPC) (Tsamardinos et al. (2003)) to initially learn the skeleton of the network. In the second phase, using a greedy Bayesian hill-climbing search, the skeleton is oriented. In the sample limit, MMHC's skeleton identification phase is reliable, but the orientation phase offers no theoretical assurances. From the results of the experiments performed, MMHC outperformed PC (Spirtes et al. (2000b)), Sparse Candidate (Friedman et al. (2013)), Optimal Reinsertion (Moore & Wong (2003)), and GES (Chickering (2002)) in terms of computational efficiency. Considering the quality of reconstruction, MMHC performs better than all the above-mentioned algorithms except for GES when the sample size is 1000. The authors also proved the correctness of the results. The implementation of MMHC is available at `http://www.dsl-lab.org/supplements/mmhcpaper/mmhcindex.html` as part of Causal Explorer 1.3, a library of Bayesian network learning and local causal discovery methods.

#### 3.5.2 FRITL

To discover causal relationships in linear and non-Gaussian models, Chen et al. (2021) proposed a hybrid model named FRITL. FRITL works in the *presence or absence of latent confounders* by incorporating independent noise-based techniques and constraint-based techniques. FRITL makes causal Markov assumption, causal faithfulness assumption, linear acyclic non-Gaussianity assumption, and one latent confounder assumption. In the *first phase* of FRITL, the FCI algorithm is used to generate asymptotically accurate results. Unfortunately, relatively few unconfounded direct causal relations are normally determined by the FCI since it always reveals the presence of confounding factors. In the *second phase*, FRITL identifies the unconfounded causal edges between observable variables within just those neighboring pairings that have been influenced by the FCI results. The *third stage* can identify confounders and the relationships that cause them to affect other variables by using the Triad condition (Cai et al. (2019)). If further causal relationships remain, *Independent Component Analysis* (ICA) is finally applied to a notably reduced group of graphs. The authors also theoretically proved that the results obtained from FRITL are efficient and accurate. FRITL produces results that are in close accord with neuropsychological opinion and in exact agreement with a causal link that is known from the experimental design when applied to real functional magnetic MRI data and the SACHS (Sachs et al. (2005)) dataset.

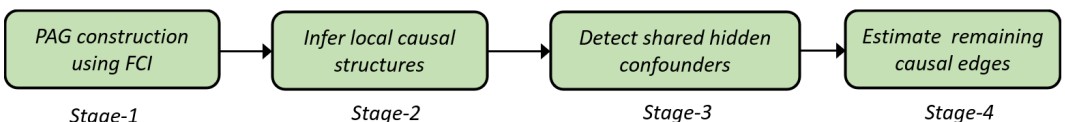

Figure 19: Stages of the FRITL model.

### 3.5.3 HCM

Most of the causal discovery algorithms are applicable only to either discrete or continuous data. However, in reality, we often have to work with mixed-type data (e.g., shopping behavior of people) which don't receive enough attention in causal discovery. Li et al. (2022) proposed the approach **H**ybrid **C**ausal Discovery on **M**ixed-type Data (HCM) to identify causal relationships with mixed variables. HCM works under the causal faithfulness and causal Markov assumption. HCM has three phases where in the *first phase*, the skeleton graph is learned in order to limit the search space. To do this, they used the PC-stable approach along with their proposed Mixed-type Randomized Causal Independence Test (MRCIT) which can handle mixed-type data. They also introduced a generalized score function called Cross-Validation based Mixed Information Criterion (CVMIC). In the *second phase*, starting with an empty DAG, they add edges to the DAG based on the highest CVMIC score. In order to reduce false positives, the learned causal structure is pruned using MRCIT once again in the *final phase* with a slightly bigger conditional set. They compared their approach with other causal discovery approaches for mixed data and showed HCM's superiority. However, they didn't consider any unobserved confounders in the dataset which allows for further improvement. They made the code available on the following GitHub site: `https://github.com/DAMO-DI-ML/AAAI2022-HCM`.

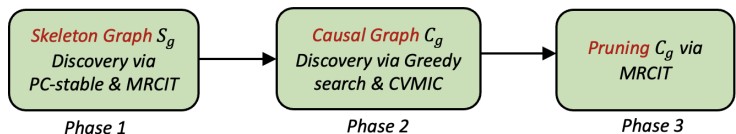

Figure 20: Different phases of the method HCM.

### 3.5.4 SADA

One of the biggest limitations of the traditional causal discovery methods is that these models cannot identify causal relations when the problem domain is large or there is a small number of samples available. To solve this problem, Cai et al. (2013) proposed a *Split-and-Merge* causal discovery method named SADA which assumes causal faithfulness. Even in situations when the sample size is substantially less than the total number of variables, SADA can reliably identify the causal factors. SADA divides the main problem into two subproblems and works in three phases. Initially, SADA separates the variables of the causal model into two sets $V_1$ and $V_2$ using a causal cut set $C$ where all paths between $V_1$ and $V_2$ are blocked by $C$. This partitioning is continued until the variables in each subproblem are less than some threshold. In the next phase, any arbitrary causal algorithm is applied to both subproblems and the causal graphs are generated. Here, they used LiNGAM as the causal algorithm. Then these graphs are merged in the final step. But to handle the conflicts while merging, they only kept the most significant edge and eliminated the others whenever there existed multiple causal paths between two variables in the opposite direction. They compared the performance of SADA against baseline LiNGAM (without splitting and merging), and the results showed that SADA achieved better performance in terms of the metrics precision, recall, and F1 score.

### 3.5.5 CORL

Ordering-based Causal Discovery with Reinforcement Learning (CORL) formulates the ordering search problem as a *multi-step Markov decision process* (MDP) to learn the causal graph (Wang et al. (2021)). It implements the ordering generating process with an *encoder-decoder architecture* and finally uses RL to optimize the proposed model based on the reward mechanisms designed for each order. A generated ordering is then processed using variable selection to obtain the final causal graph. According to the empirical results, CORL performs better than existing RL-based causal discovery approaches. This could happen because CORL does not require computing the matrix exponential term with $O(d^3)$ cost because of using ordering search. CORL is also good in terms of scalability and has been applied to graphs with up to 100 nodes. The gCastle package contains an implementation of CORL.

### 3.5.6   ETIO

ETIO is a versatile *logic-based* causal discovery algorithm specialized for business applications (Borboudakis & Tsamardinos (2016)). Its features include i) the ability to utilize prior causal knowledge, ii) addressing selection bias, hidden confounders, and missing values in data, and iii) analyzing data from pre and post-interventional distribution. ETIO follows a *query-based approach*, where the user queries the algorithm about the causal relations of interest. In the first step, ETIO performs several CI tests on the input dataset. Particularly, it performs non-Bayesian tests that return p-values of the null hypothesis of conditional independencies. Then it employs an empirical Bayesian method that converts the p-values of dependencies and interdependencies into probabilities. Later, it selects a consistent subset of dependence, and prior knowledge constraints to resolve conflicts which are ranked in order of confidence. Particularly, ETIO imposes an m-separation constraint if a given independence is more probable than the corresponding dependence. These imposed constraints are the ones that correspond to test results, in order of probability, while removing conflicting test results. Finally, it identifies all invariant features based on input queries using the well-known declarative programming language, answer set programming (Gelfond & Lifschitz (1988)).

### 3.5.7   $b$QCD

Discovering causal relationships from observational data has been a challenging task, especially for the bivariate cases as it is difficult to determine whether there actually exists a cause-effect relationship or whether it is the effect of a *hidden confounder*. Tagasovska et al. (2020) proposed the approach **b**ivariate **Q**uantile **C**ausal **D**iscovery (bQCD) to determine causal relationships in bivariate settings. Although they made no assumptions on the class of causal mechanisms, they did assume that there exists no confounder, feedback, or selection bias. They utilized *quantile scoring* in place of Kolmogorov complexity (Kolmogorov (1963)), and used conditional quantiles, pinball loss instead of conditional mean, and squared loss. The approach bQCD performs almost similarly to the state-of-the-art techniques but it is much more computationally inexpensive. Also, the usage of quantile conditioning instead of mean conditioning makes bQCD more robust to heavy tails as the mean is more susceptible to outliers than the quantile. Moreover, not making any assumptions about the parametric class allows bQCD to be applied to a variety of processes where baseline methods perform significantly poorly when the assumptions do not hold. The source code of bQCD written in R is available on this site: `https://github.com/tagas/bQCD`.

### 3.5.8   JCI

**J**oint **C**ausal **I**nference (JCI) leverages prior knowledge by combining data from multiple datasets from different contexts (Mooij et al. (2020)). Particularly, JCI is a *causal modeling framework* rather than a specific algorithm, and it can be implemented using any causal discovery algorithm that can take into account some background knowledge. The main idea of JCI is to first, consider auxiliary context variables that describe the context of each data set, then, pool all the data from different contexts, including the values of the context variables, into a single data set, and finally apply standard causal discovery methods to the pooled data, incorporating appropriate background knowledge on the causal relationships involving the context variables. The framework is simple and easily applicable as it deals with latent confounders, cycles (if the causal discovery method supports this), and various types of interventions in a unified way. The JCI framework also facilitates analysis of data from almost arbitrary experimental designs which allow researchers to trade off the number and complexity of experiments to be done with the reliability of the analysis for the purpose of causal discovery.

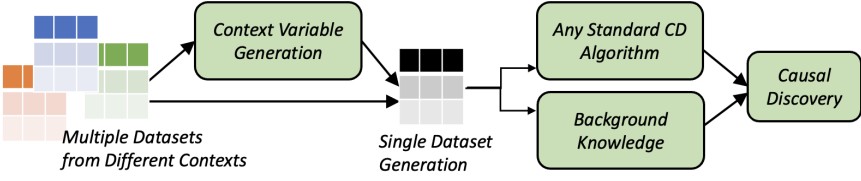

Figure 21: Workflow of the JCI framework.

### 3.5.9 Kg2Causal

Kg2Causal (Sinha & Ramsey (2021)) uses a large-scale general-purpose biomedical knowledge graph as a prior for data-driven causal discovery. With a set of observed nodes in a dataset and some relationship edges between the nodes derived from a knowledge graph, Kg2Causal uses the knowledge graph-derived edges to guide the data-driven discovery of a causal graph. The main ideas of this approach are first, mapping each variable in the dataset to a node in the knowledge graph, and querying relationships between them; next, extracting a subgraph containing the connected variables with edges between them; and then this edge set is used as prior knowledge to guide an optimizing scoring step for inferring the causal graph. An implementation of Kg2Causal is available at `https://github.com/meghasin/Kg2Causal` in R language.

### 3.5.10 C-MCMC

**C**onstrained MCMC (C-MCMC) introduces *prior knowledge* into the *Markov chain Monte Carlo (MCMC)* algorithm for structure learning (Xu et al. (2015)). C-MCMC uses the following *three types of prior knowledge*: the existence of parent nodes, absence of parent nodes, and distribution knowledge including the conditional probability distribution (CPD) of edges and the probability distribution (PD) of nodes. All prior knowledge should be given by domain experts. Existence knowledge means that for any node $X_i$, a node-set $pa(X_i)$ includes all parent nodes of $X_i$. The absence of knowledge means that for a node $X_i$, a node-set $pa(X_i)$ does not include any parent node of $X_i$. PD/CPD knowledge means that the PD of a node and the CPD of an edge are known. Considering that the prior knowledge may not be consistent and reliable, a confidence lambda is assigned by domain experts on each of the prior knowledge that ranges from 0 to 1. This denotes the certainty level of prior knowledge. A *lambda* value of 1 indicates very high confidence in this knowledge.

### 3.5.11 M*eta*-RL

Meta-RL is a *meta-learning algorithm* in a Reinforcement Learning (RL) setting where the agent learns to *perform interventions* to construct a causal graph (Sauter et al. (2022)). The goal is to be able to use previous learning experiences during training to generalize in unseen environments. This approach has some strong assumptions such as i) each environment is defined by an acyclic SCM, ii) every observable variable can be intervened on, iii) for each environment in the training set, the underlying SCM is given, and iv) intervention can be performed on at most one variable at a time. Meta-RL has two phases: i) Training, and ii) Application. The training phase starts by randomly choosing an SCM from a set of environments. There are mainly two sets of actions that an agent performs: *a) interventional actions*, and *b) structure actions.* In each step, any one action can be performed on the set of variables to generate a PDAG. The *agent policy is updated* via the *interventional actions* in each step. However, in case of the structural actions (e.g. add, delete, or reverse), the agent policy only gets updated at the end of the training procedure where a reward is sent to the agent. The reward is computed by comparing the hamming distance of the generated PDAG to the true causal structure when the training is completed. A *recurrent LSTM layer* enables the policy to remember samples from the post-interventional distributions in the earlier steps. This should help to better identify causal relations since the results of sequential interventions can be used to estimate the distribution. Once trained, Meta-RL can then be applied to environments that have a structure unseen during training. For training, 24 SCMs with 3 observable variables, and 542 SCMs with 4 observable variables were created. Code to reproduce experiments or run Meta-RL is available at `https://github.com/sa-and/interventional_RL`. One limitation of this approach is that it needs modification in terms of scalability. Also, in real-world scenarios, every variable might not be accessible for intervention.

### 3.5.12 Tabu search for SEM

Marcoulides et al. (1998) presents an approach to structural equation modeling (SEM) specification search that makes use of Tabu search, a heuristic optimization algorithm. Using a neighborhood of the current solution as its focus, the tabu search technique avoids local optimality by examining the area around the current solution. To prevent cycling, it assigns recently involved attributes a *tabu status.* A number of definitions and parameters, such as the neighborhood definition and the model selection criterion, are necessary

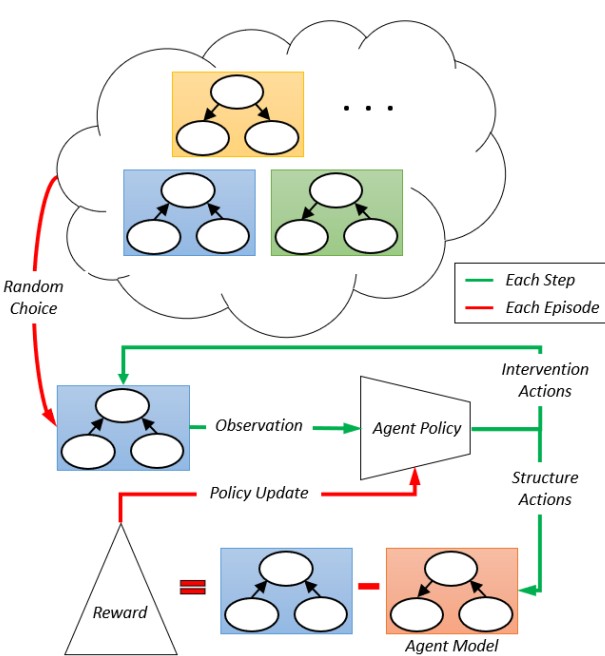

Figure 22: Training phase of the Meta-RL algorithm (Sauter et al. (2022)).

to implement the Tabu search procedure for SEM specification search. The authors conclude that Tabu search is a promising strategy for SEM specification search after demonstrating its efficacy in a number of example analyses.

### 3.5.13   LFCM

**L**atent **F**actor **C**ausal **M**odels (LFCMs) (Squires et al. (2022)) perform causal discovery in the *presence of latent variables*. These models are motivated by gene regulatory networks. LFCMs work in three stages where they discover: (i) clusters of observed nodes, (ii) a partial ordering over clusters, and (iii) finally, the entire structure over both observed and latent nodes. A graph $G$ is called a latent factor causal model (LFCM) if it satisfies the following conditions: (a) Unique cluster assumption: Each observed node has exactly one latent parent, (b) Bipartite assumption: There are no edges between pairs of observed nodes or between pairs of latent nodes, (c) Triple-child assumption: Each latent node has at least 3 observed children and (d) Double-parent assumption. The other assumption of LFCMs is that it allows non-exogenous latent variables. For cluster formation, LFCMs rely on t-separation (Sullivant et al. (2010)). When two ordered pairs of variables [e.g. $(X_i, X_j)$ and $(X_u, X_v)$] are t-separated, then they belong to the same cluster. LFCMs are a biologically motivated class of causal models with latent variables. The limitations of LFCMs include their applicability to only a linear Gaussian SEM, some major structural restrictions, and that it can fail when the true graph violates the double parent assumption.

There are some other noteworthy methods that are specialized to handle latent variables or unobserved confounders. Kocaoglu et al. (2017) presented a non-parametric algorithm for learning a causal graph in the presence of hidden variables. The study took a stage-by-stage approach, first to learn the induced graph between observational variables, and then use it to discover the existence and location of the latent variables. The authors further proposed an algorithm to discover the latent structure between variables depending on the adjacency. To identify ancestral relationships and transitive closure of the causal graph, the algorithm employed a pairwise independence test under interventions. Then Kocaoglu et al. (2019) addressed causal discovery by linking conditional independencies from observed data to graphical constraints using the d-separation criterion. It broadened the application of this strategy to scenarios involving numerous experimental and observational distributions. The authors proposed that CIs and d-separation constraints are just a subset of broader constraints obtained from comparing various distributions, which is especially useful

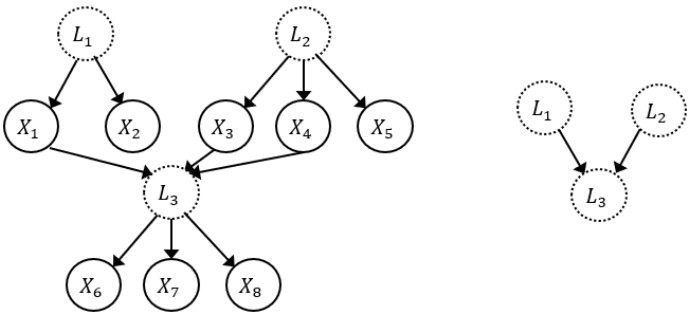

Figure 23: The graph $G$ on the left is a latent factor causal model (LFCM), and the graph on the right is the latent graph $L(G)$ for $G$ (Squires et al. (2022)).

in the context of do-calculus for soft interventions. They introduced the notion *interventional equivalence class of causal graphs with latent variables*, which linked graphical structures to groups of interventional distributions that adhered to do-calculus. Two causal graphs are interventionally equivalent if they produce identical interventional distributions that can not be distinguished by invariances.

Sometimes complex systems require the knowledge of both observations and experiments for recovering the underlying causal relationships. Utilizing both observational and interventional data from various domains, the authors in Li et al. proposed a novel approach for identifying causal structures in semi-Markovian systems with latent confounders. They made a link between learning from interventional data within a single domain and learning from observational data across domains. They introduced the idea of S-Markov, a property connecting multi-domain distributions to pairs of causal graphs and interventional targets, to navigate the complexities of observational and experimental data. A new causal discovery algorithm called S-FCI was introduced that builds on the S-Markov property and is capable of effectively learning from a mixture of observational and interventional data from various domains.

Jaber et al. (2020) integrates soft experimental and observational data to find the structure in non-Markovian systems with latent variables. They introduced the idea of $\Psi$-Markov in this context when the intervention targets were unidentified. This idea links a causal graph $G$ and a list of interventional targets $I$ to the causal invariances found in both observational and interventional data distributions. They also introduced a graphical method for evaluating equivalence between causal graphs with various interventional targets and an algorithm for learning the equivalence class.

Recently, Bellot et al. (2022) studied structure learning in discrete models with arbitrary latent dependencies, and proposed a new score based on the asymptotic expansion of the marginal likelihood to capture both equality and inequality constraints in observational data. Furthermore, it claims to be the first score-based method to learn causal models with latent variables.

The different methods discussed in this section so far use a variety of strategies to perform causal discovery under diverse settings and assumptions. Therefore, we present a comparative analysis of some of the common methods in Table 6 based on their assumptions, output causal graph, techniques used, advantages, and disadvantages. This comparative analysis will help readers to find the similar and dissimilar methods, and also help in deciding which method could be appropriate for performing causal discovery given the data and its assumptions.

Table 6: Comparison among some causal discovery algorithms for I.I.D. data.

| Methods | Assumptions | Outcome | Technique used | Advantages | Disadvantages |
|---|---|---|---|---|---|
| PC | Faithfulness, Sufficiency | CPDAG | Conditional Independence (CI) Tests | Computationally more feasible for sparse graphs. | Lacks scalability as less feasible for denser graphs |
| FCI | Causal Markov condition, faithfulness | PAG | CI tests, Skeleton finding step same as PC | Handles latent and selection variables. | In the worst case, the no. of CI tests performed grows exponentially with the no. of variables. |
| RFCI | CMC, faithfulness | PAG | Conditional Independence tests | Faster variant of FCI, uses fewer CI tests, computationally feasible for high-dimensional sparse graphs | Performs some additional tests before orienting v-structures and discriminating paths |
| GES | Decomposable score function | CPDAG | Score-based greedy search | Run time faster compared to the constraint-based methods | Search space can grow exponentially with the growing no. of variables |
| FGS | Weak Faithfulness | DAG | Score-based greedy search | Faster variant of GES, reduces scoring redundancy, enables parallelization | Require high power computing to run |
| RL-BIC | Decomposable score function, acyclicity | DAG | BIC score-based reinforcement learning search | Model gets feedback to update /correct its search strategy | Scalable only up to a few (around 30) variables |
| Triplet A* | Acyclicity | DAG | A* search combined with BIC score | Can handle both linear Gaussian & non-Gaussian networks, scales up to more than 60 variables | Has complexity issues |
| KCRL | Decomposable score-function, unbiased prior knowledge | DAG | Score-based RL search strategy | Considers prior knowledge constraints | Lacks scalability |
| LiNGAM | Linear DGP, no unobserved confounders, non-Gaussian noises | DAG | FCM-based, uses independent component analysis | Determines the direction of every causal arrow, does not require the faithfulness assumption. | Estimated results may vary in case of mixed data (categorical values) |
| SAM | Acyclicity | DAG | Searches for an FCM using GANs | Good for sparse causal graphs | Not suitable for dense graphs |
| Tabu search | Finite search space, well-defined objective function | DAG | Iteratively modifies the model and evaluates its fit to the data by examining a neighborhood of the current solution | Considers avoiding local optima and also, cycle avoidance | Quality of the results depends on the quality of the initial solution and the choice of parameters |
| CAM | Sufficiency, Acyclicity | DAG | FCM-based, uses sparse regression techniques and decouples order search among the variables | Establishes consistency of the maximum likelihood estimator for low and high-dimensional cases | Faces performance and computational challenges as the number of variables increases |
| NOTEARS | Acyclicity, linear dependence between variables | DAG | Models data as a SEM, uses a regularized score-function with a least-square loss | Simple method, easy to implement | Works well mostly for continuous data |
| GAE | Structure learning under additive noise models | DAG | Gradient-based approach that uses a graph autoencoder framework | Can scale up to 100 nodes, Lower run time, easily applicable to vector-valued variables, & handles non-linear relations well | May not work well for linear causal relations |
| DAG-GNN | Faithfulness, sufficiency | DAG | Gradient-based, uses blackbox stochastic optimization solvers to solve the sub problem of maximizing the ELBO | Can handle both discrete and vector-valued variables, and is capable of capturing complex nonlinear mappings | Assumes acyclic causal relationships, which may not always be the case in real-world scenarios |
| MMHC | Faithfulness, sufficiency; decomposable score-function | DAG | Hybrid: uses both score and constraint-based techniques such as MMPC to initially learn the skeleton of the network, then uses greedy Bayesian hill-climbing search | Good computational efficiency, scalability, and applicable for high-dimensional settings | Scales up better only with large number of samples |
| FRITL | Faithfulness, non-Gaussianity, latent confounder assumption | PAG | Hybrid, uses FCI, Triad condition and ICA | Works in presence or absence of latent confounders | Not generalizable in non-linear gaussian cases |
| Kg2Causal | Presence of a knowledge graph | DAG | Prior knowledge constraints added in a score-based method | Leverages information from existing literature or domain | Requires a knowledge graph |

# 4 Causal Discovery Algorithms for Time Series Data

Time series data arise when observations are collected over a period of time. So far the methods that we have discussed are specialized for causal discovery from I.I.D. or time-independent data. However, often, real-world data in different domains can be a time series (non-I.I.D. data). For this type of data, there are different specialized causal discovery approaches based on CI testing, SEM/FCMs, Granger causality (Granger (1969)), or deep neural networks. In this section, first, we provide a brief introduction to some of the common terminologies related to time-series data and temporal causal discovery. Then, we discuss the notable causal discovery approaches for time-series data.

**Definition 7 (Time Series Data)** *Time series data is a collection of observations measured over consistent intervals of time. The observation of a time series variable $X^j$ at time $t$ is denoted by $X_t^j$.*

Examples of time series data include retail sales, stock prices, climate data, heart rate of patients, brain activity recordings, temperature readings, etc. Any time series data may have the following ***properties***:

i **Trend***:* When the data show a long-term rise or fall, a trend is present. Such long-term increases or decreases in the data might not be always linear. The trend is also referred to as *changing direction* when it might switch from an upward trend to a downward trend.

ii **Seasonality***:* It refers to the seasonal characteristics of time series data. Seasonality exists when the data regularly fluctuates based on different time spans (e.g. daily/weekly/ monthly/quarterly/yearly). An example is temperature data, where it is mostly observed that the temperature is higher in the summer, and lower in the winter. Any analysis related to time series usually takes advantage of the seasonality in data to develop more robust models.

iii **Autocorrelation***:* Autocorrelation or self-correlation is the degree of similarity between a given time series and a lagged version of itself over successive time intervals. Time series data is usually autocorrelated i.e., the past influences the present and future (Lawton et al. (2001)).

iv **Stationarity & Non-stationarity***:* Stationarity means that the joint probability distribution of the stochastic process does not change when shifted in time. A time series is stationary if it has causal links such that for variables $X^i$ and $X^j$, if $X^i \to X^j$ at any timestamp $t$, then $X^i \to X^j$ also holds for all $t'$ $\neq t$. This condition does not hold for a non-stationary time series where $X^i \to X^j$ at a particular time $t$ need not necessarily be true at any other time stamp $t'$.

Let $X_{1:t}^j = \{X_{1:t}^1, X_{1:t}^2, \ldots, X_{1:t}^n\}$ be a multivariate time series with $n$ variables and $t$ time steps. At any particular timestamp $t$, the state of the $n$ variables can be represented as $X_t^j = \{X_t^1, X_t^2, \ldots, X_t^n\}$. The past of a variable $X_t^j$ is denoted by $X_{1:t-1}^j$. The parent set of a variable includes all the nodes with an edge towards it. The goal of any temporal causal discovery approach is to discover the causal relationships between the time series variables. Any time series causal graph may have the following ***types of causal relationships/edges***: *(i) Instantaneous edges*, and *(ii) Lagged edges*.

**Definition 8 (Instantaneous Causal Effect)** *When the delay between cause and effect is 0 timesteps, i.e. causal effects are of the form $X_t^i \to X_t^j$ or $X_t^i \to X_t^i$ (self-causation), then it is known as an instantaneous or contemporaneous causal relationship/effect (Nauta et al. (2019)).*

**Definition 9 (Lagged Causal Effect)** *When the delay between cause and effect is at least 1 or more timesteps (i.e. causal effects of the form $X_{t-}^i \to X_t^j$ or $X_{t-}^i \to X_t^i$), then it is known as a lagged causal relationship/effect. That is, a lagged causal effect occurs when a variable causes another variable or itself with a time lag = 1 or more.*

In Figure 24, the red-colored edges represent the instantaneous causal effect (relationships among the variables at the same time step), and the blue edges represent the lagged causal effect. The green edges represent a special form of temporal causal relationships known as the ***changing modules*** (CM). The CMs

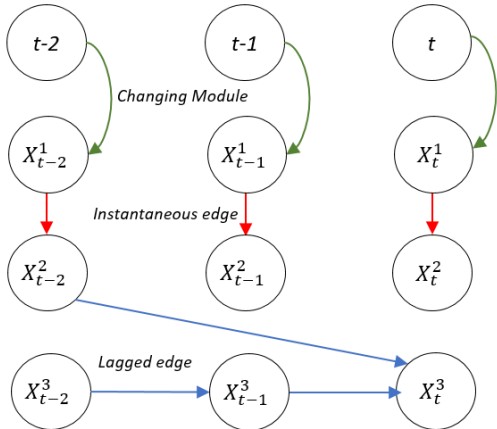

Figure 24: Types of causal relationships: Instantaneous edges (red), Lagged edges (blue), and Changing modules (green).

represent the direct effect of a time stamp on a variable (e.g. $t \rightarrow X_t^1$ in Figure 24). Details on CM are available in Ferdous et al. (2023b).

The causal graphs produced by different temporal causal discovery algorithms vary based on the details of the relationships they represent. Any temporal causal discovery algorithm may produce any of the following two **types of temporal causal graph** as its outcome: a *full-time causal graph* or a *summary causal graph* (see Figure 25).

**Definition 10 (Full-time Causal Graph)** *A full-time causal graph represents both the instantaneous ($X_t^i \rightarrow X_t^j$ or $X_t^i \rightarrow X_t^i$) and time-lagged ($X_{t-}^i \rightarrow X_t^j$ or $X_{t-}^i \rightarrow X_t^i$) causal edges where the lag between a cause and effect is specified in the graph. A full-time casual graph may sometimes present the changing modules as well.*

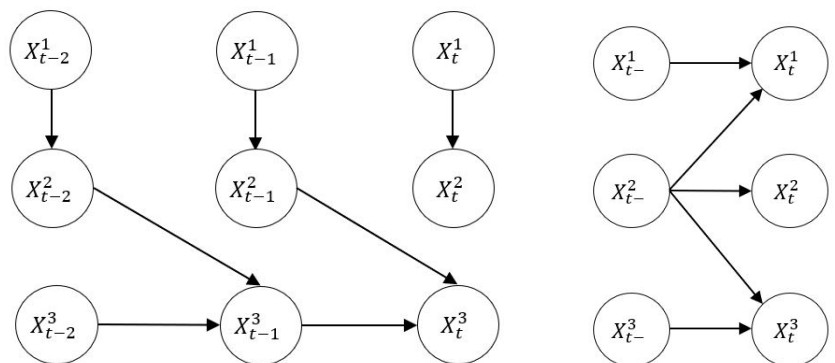

Figure 25: Full-time causal graph (Left) & Summary causal graph (Right)

Figure 25 (left) represents a full-time causal graph where both instantaneous relations (e.g. $X_t^1 \rightarrow X_t^2$, $X_{t-1}^1 \rightarrow X_{t-1}^2$), and lagged relations (e.g. $X_{t-1}^2 \rightarrow X_t^3$, $X_{t-1}^3 \rightarrow X_t^3$) among the variables are depicted.

**Definition 11 (Summary Causal Graph)** *A summary causal graph is a reduced version of a full-time causal graph where each lagged node represents the entire past ($X_{t-}^j$) of its corresponding instantaneous node ($X_t^j$), and the exact time lag between the cause and effect is not specified in the graph.*

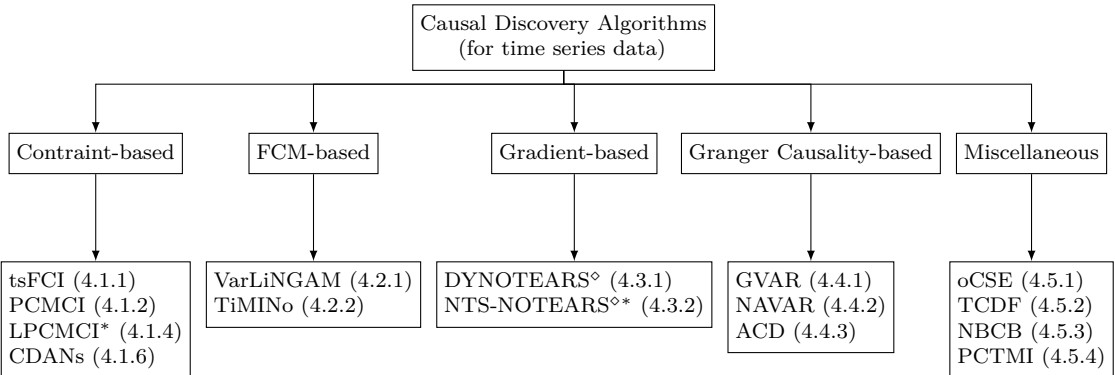

Figure 26: Taxonomy of some of the discussed causal discovery approaches for time series data. The approaches are classified based on their core contribution or the primary strategy they adopt for causal structure recovery. The approaches that can leverage prior knowledge are marked by an $*$ symbol. Some of the gradient-based approaches that use a score function are indicated by a $\diamond$ symbol. They are primarily classified as such as they use gradient descent for optimization. However, they can be a score-based method too as they compute data likelihood scores on the way.

In the following subsections, we describe briefly some of the notable causal discovery algorithms that focus on time series data. Figure 26 presents a taxonomy of some of the discussed approaches.

## 4.1 Constraint-based

### 4.1.1 $ts$FCI

The algorithm *time series* FCI or tsFCI (Entner & Hoyer (2010)) adapts the Fast Causal Inference (Spirtes et al. (2000a)) algorithm (developed for the causal analysis of non-temporal variables) to infer causal relationships from time series data. It works in two phases: (i) an *adjacency phase*, and (ii) an *orientation phase*. It makes use of temporal priority and consistency throughout time to orient edges and restrict conditioning sets. It provides a window causal graph, and an advantage is that it *can detect lagged hidden confounders*. However, a disadvantage is that it cannot model cyclic contemporaneous causation, and also instantaneous relationships. A code package that implements tsFCI is available at `https://sites.google.com/site/dorisentner/publications/tsfci`.

### 4.1.2 PCMCI

A problem with large-scale time series data is that although adding more variables makes causal analysis more interpretable, if the additional variables don't have a significant effect on the causal model, this, in turn, makes the analysis less powerful, and original causal relations may also be overlooked. Moreover, at large dimensions, certain nonlinear tests even lose their ability to limit false positive rates (FPRs). Runge et al. (2019) proposed a two-stage algorithm PCMCI that can overcome this problem. In *Step-1*, the model selects conditions using $PC_1$ (a variant of the skeleton discovery part of the PC algorithm) to remove irrelevant variables which solve the issue of low power in the causal discovery process. In *Step-2*, the momentary conditional independence (MCI) test is used which helps to reduce the FPR even when the data is highly correlated. The MCI test measures if two variables are independent or not given their parent sets (see Equation 13).

$$X_{t-\tau}^i \perp\!\!\!\perp X_t^j | P_A(X_t^j), P_A(X_{t-\tau}^i) \tag{13}$$

PCMCI assumes that the data is stationary, has time-lagged dependencies, and also assumes causal sufficiency. Even when the stationary assumption is violated (probably by obvious confounders), PCMCI still provides a more robust performance than Lasso regression or the PC algorithm. However, for highly predictable

systems where little new information is produced at each time step, PCMCI is not a good fit. Python implementation of PCMCI is available in the *Tigramite* package (`https://github.com/jakobrunge/tigramite`).

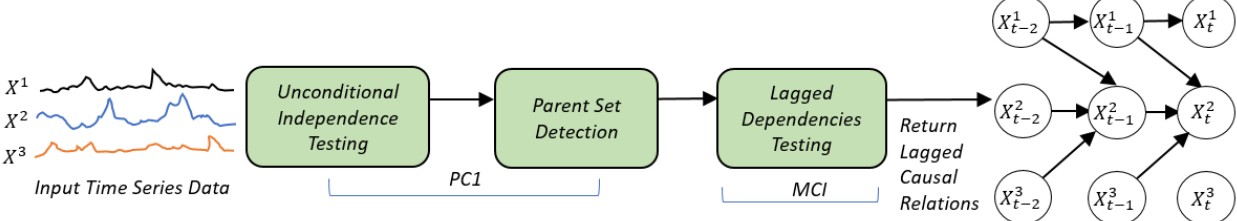

Figure 27: Steps involved in the PCMCI method for time series causal discovery.

### 4.1.3 PCMCI+

PCMCI+ (Runge (2020)) is an extension of the PCMCI algorithm to discover contemporary or instantaneous causal links. PCMCI+ also assumes causal sufficiency like the PCMCI algorithm. It is also a two-stage algorithm where in the first stage, irrelevant edges from the causal model are eliminated. Unlike PCMCI, the edges are removed separately for lagged and contemporary conditioning sets where the contemporary phase employs more CI tests than the lagged phase. In the second stage, PCMCI+ employs the notion of momentary conditional independence (MCI) to improve the selection of conditioning sets for the various CI tests, improving their autocorrelation calibration, and boosting their detection power. The results show that when there is high autocorrelation in the data, PCMCI+ can achieve better performance in terms of higher recall, lower false positives, and faster execution compared to the PC algorithm. For lower autocorrelation, PCMCI+ performs almost similarly to PC. Implementation of PCMCI+ is also available in the Tigramite package (`https://github.com/jakobrunge/tigramite`).

### 4.1.4 LPCMCI

Latent PCMCI (LPCMCI) is a constraint-based causal discovery algorithm to determine causal relationships from large-scale time series data (Gerhardus & Runge (2020)). This is another extension of the PCMCI algorithm as it can discover causal relationships *even in the presence of latent confounders.* Moreover, it gives the flexibility to use the model when the data is linear or nonlinear, and also when the data has lagged or contemporary conditioning sets. The authors identified that when the CI tests have a low effect size, existing techniques like FCI suffer from low recall in the presence of autocorrelation. They demonstrated that this issue can be solved by including causal parents in the conditioning sets. By utilizing the orientation rules, these parents can be identified as early as in the edge removal stage. The results show that the proposed LPCMCI method can achieve higher recall than the baseline model SVAR-FCI. However, LPCMCI cannot differentiate all members of the Markov class, and also, when the faithfulness assumption doesn't hold, LPCMCI might lead to an incorrect conclusion. Along with PCMCI and PCMCI+, the Python code of LPCMCI is also available in the Tigramite GitHub package.

### 4.1.5 CD-NOD

Many existing approaches assume that the causal model is static, and therefore, there will be a fixed joint distribution of the observed data. However, these methods fail when the underlying data changes over time, and causal parameters vary during the period. Huang et al. (2020) proposed a causal discovery method that assumes that the parameter of the causal model can change over time or different datasets, and they named the method CD-NOD, *Constraint-based Causal Discovery from Heterogeneous/Nonstationary Data.* The proposed method can determine causal direction by taking advantage of distribution shifts, and these distribution changes, in the presence of stationary confounders, are helpful for causal discovery. The distribution shifts can be either time or domain indexes and are denoted by a surrogate variable $C$. Broadly, CD-NOD has two phases where in the *first phase* it recovers the causal skeleton $S_G$, and in the *second phase* it orients the edges as per some orientation rules. Given that the causal model offers a concise summary of how the joint

distribution changes, they demonstrated that distribution shift contains important information for causal discovery. Recently, researchers discovered that this idea could help solve machine learning problems of domain adaptation and forecasting in nonstationary situations (Schölkopf et al. (2012); Zhang et al. (2013)). The conducted experiments in this study demonstrate the changes of causal influence between the different states of brain functions, and the empirical results show that CD-NOD has improved precision and F1 score. However, they didn't consider that the causal directions might flip, and the power of conditional independence tests might reduce because of the distribution shifts. The algorithm's source code is available in the following link: `https://github.com/Biwei-Huang/Causal-Discovery-from-Nonstationary-Heterogeneous-Data`.

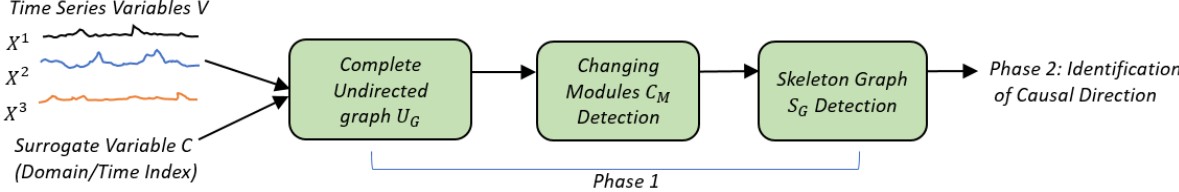

Figure 28: Illustration of CD-NOD's phase-1.

### 4.1.6  CDANs

Ferdous et al. (2023b) introduces a constraint-based causal discovery approach called CDANs for autocorrelated and non-stationary time series data that handles high dimensionality issues. The method identifies both lagged and instantaneous causal edges along with changing modules that vary over time. By optimizing the conditioning sets in a constraint-based search, and also considering lagged parents instead of conditioning on the entire past, it tries to address the high dimensionality problem. CDANs first detect the lagged adjacencies, then identify the changing modules and instantaneous adjacencies, and finally determine the causal direction. The code to implement this method is available at `https://github.com/hferdous/CDANs`. An extended version of this study is presented in Ferdous et al. (2023a), where the method called **eCDANs** is introduced that is capable of detecting lagged and instantaneous causal relationships along with temporal changes. The method eCDANs addresses high dimensionality by optimizing the conditioning sets while conducting CI tests and identifies the changes in causal relations by introducing a proxy variable to represent time dependency.

## 4.2  Functional Causal Model (FCM)-based

### 4.2.1  VarLiNGAM

VarLiNGAM (Hyvärinen et al. (2010)) combines the non-Gaussian instantaneous models with autoregressive models and shows that a non-Gaussian model is identifiable without prior knowledge of network structure. It estimates both instantaneous and lagged causal effects in models that are an example of structural vector autoregressive (SVAR) models. These models are a combination of structural equation models (SEM) and vector autoregressive (VAR) models. VarLiNGAM also shows that taking instantaneous influences into account can change the values of the time-lagged coefficients to a great extent. Thus, neglecting instantaneous influences can lead to misleading interpretations of causal effects. It also assesses the significance of the estimated causal relations. An implementation of this method is available at: `https://lingam.readthedocs.io/en/latest/tutorial/var.html`.

### 4.2.2  T*i*MINo

**Ti**me-series **M**odels with **I**ndependent **No**ise (TiMINo) (Peters et al. (2013)) studies a class of restricted structural equation models (SEMs) for time-series data that include nonlinear and instantaneous effects. It assumes $X_t$ to be a function of all direct causes and some noise variable, the collection of which is supposed to be jointly independent. The algorithm is based on unconditional independence tests and is applicable to multivariate, linear, nonlinear, and instantaneous interactions. If the model assumptions are

not satisfied by the data, TiMINo remains mostly undecided instead of making wrong causal decisions. While methods like Granger causality are built on the asymmetry of time direction, TiMINo additionally takes into account identifiability emerging from restricted SEMs. This leads to a straightforward way of dealing with unknown time delays in different time series. An implementation of TiMINo is available in this repository: `https://github.com/ckassaad/causal_discovery_for_time_series`.

### 4.3 Gradient-based

### 4.3.1 DYNOTEARS

Pamfil et al. (2020) proposed the Dynamic NOTEARS (DYNOTEARS) which is a structure learning approach for dynamic data that simultaneously estimates contemporaneous (intra-slice) and time-lagged (inter-slice) relationships between variables in a time-series. DYNOTEARS revolves around minimizing a penalized loss subject to an acyclicity constraint. The optimization finds the conditional dependencies that are best supported by the data. It leverages insight from the approach NOTEARS (Zheng et al. (2018)) which uses an algebraic characterization of acyclicity in directed graphs for static data. The assumptions made by DYNOTEARS include that the structure of the network is fixed through time, and is identical for all time series in the data. This approach is scalable to high-dimensional datasets. An implementation of this approach is available in the CausalNex library (`https://github.com/quantumblacklabs/causalnex`), and also at `https://github.com/ckassaad/causal_discovery_for_time_series`.

### 4.3.2 NTS-NOTEARS

NTS-NOTEARS (Sun et al. (2021)) is a causal discovery method for time series data that uses 1-D convolutional neural networks (CNNs) to capture linear, nonlinear, lagged, and instantaneous relations among variables in a time series data along with ensuring the acyclicity property of a DAG. It extends the continuous optimization-based approach NOTEARS for learning nonparametric instantaneous DAGs, and adapts the acyclicity constraint from that approach. It assumes that there are no latent confounders in the data, and the underlying data-generating process is fixed and stationary over time. NTS-NOTEARS is faster than other constraint-based methods because of the use of nonlinear conditional independence tests. It incorporates prior knowledge into the learning process to promote the use of optimization constraints on convolutional layers for better casual discovery. Its implementation is available at: `https://github.com/xiangyu-sun-789/NTS-NOTEARS/`.

### 4.4 Granger Causality (GC)-based

Granger (1969) investigated the causal relationships between the variables in a time series data which is known as Granger Causality (GC). It is based on the basic assumption that *causes precede their effects*. The author defines GC as follows: *A time series variable $X^i$ causes $X^j$, if the probability of $X^j$ conditional on its own past, and the past of $X^i$ (besides the set of the available information) does not equal the probability of $X^j$ conditional on its own past alone.* The GC test can't be performed directly on non-stationary data. The non-stationary data needs to be transformed into stationary data by differencing it, either using first-order or second-order differencing. Granger Causality can be used when there are no latent confounders, and also, no instantaneous effects exist, i.e., no variable causes another variable at the same time stamp.

### 4.4.1 GVAR

**G**eneralized **V**ector **A**uto**R**egression (GVAR) (Marcinkevičs & Vogt (2021)) is a framework for inferring multivariate Granger causality under nonlinear dynamics based on autoregressive modeling with self-explaining neural networks. It allows the detection of signs of Granger-causal effects and inspection of their variability over time in addition to relational inference. It focuses on two aspects: first, inferring Granger-causal relationships in multivariate time series under nonlinear dynamics, and second, inferring signs of Granger-causal relationships. A reproducible code of the approach is available at: `https://github.com/i6092467/GVAR`.

### 4.4.2 NAVAR

Bussmann et al. (2021) proposed the approach **N**eural **A**dditive **V**ector **A**uto**R**egression (NAVAR) which is a causal discovery approach for capturing nonlinear relationships using *neural networks*. It is particularly trained using deep neural networks that extract the (additive) Granger causal influences from the time evolution in multivariate time series. NAVAR assumes an additive structure where the predictions depend linearly on independent nonlinear functions of the individual input variables. These nonlinear functions are modeled using neural networks. The additive structure of NAVAR allows scoring and ranking the causal relationships. Currently, NAVAR is implemented with MLPs and LSTMs as the backbone using Python which is available at: `https://github.com/bartbussmann/NAVAR`. However, more complex architectures such as dilated CNNs and transformers can also be used to model NAVAR.

### 4.4.3 ACD

Most causal discovery algorithms applied for time-series analysis find a causal graph for the data, and then refit the model whenever new samples do not fit with the underlying causal graph. But in many cases, samples share connections among them, for example, the brain activity of different regions at different times. When the algorithms fit a new model, this dynamic nature between the samples is lost, and can no longer identify the actual causal relation. To solve this problem, Löwe et al. (2022) proposed the **A**mortized **C**ausal **D**iscovery (ACD) technique which can identify the causal relations when samples are from different causal graphs but share common dynamics. ACD consists of an encoder and a decoder. The encoder predicts the causal graph's edges by learning Granger causal relations, and under the assumed causal model, the decoder simulates the dynamics of the system for the next time-step. Implementation of the model is available at: `https://github.com/loeweX/AmortizedCausalDiscovery`.

## 4.5 Miscellaneous Approaches

### 4.5.1 oCSE

Causal network inference by Optimal Causation Entropy (oCSE) (Sun et al. (2015a)) is based on the *optimal causation entropy principle* which utilizes a two-step process (*aggregative discovery and progressive removal*) to jointly infer the *set of causal parents* of each node. It proposes a theoretical development of *causation entropy*, an information-theoretic statistic designed for causal inference. Particularly, it proves the optimal causation entropy principle for Markov processes which is as follows: *the set of nodes that directly cause a given node is the unique minimal set of nodes that maximizes causation entropy*. This principle transforms the problem of causal inference into the optimization of causation entropy. Causation entropy can be regarded as a type of conditional mutual information designed for causal structure inference which generalizes the traditional, unconditioned version of transfer entropy. Causation entropy when applied to Gaussian variables also generalizes Granger causality and conditional Granger causality. An advantage of the method oCSE is that it often requires a relatively smaller number of samples, and fewer computations to achieve high accuracy. Due to its aggregative nature, the conditioning set encountered in entropy estimation remains relatively low-dimensional for sparse networks. An implementation of the oCSE algorithm is available on this website: `https://github.com/ckassaad/causal_discovery_for_time_series`.

### 4.5.2 TCDF

**T**emporal **C**ausal **D**iscovery **F**ramework (TCDF) (Nauta et al. (2019)) is a *deep learning framework* that discovers the causal relationships in observational time series data. Broadly, TCDF has the following steps: (i) Time series prediction, (ii) Attention interpretation, (iii) (a) Causal validation, (iii) (b) Delay discovery, and (iv) Temporal causal graph construction. TCDF consists of $N$ independent *attention-based convolutional neural networks (CNNs)* all with the same architecture but a different target time series. Each network receives all observed time series as input. The goal of each network is to predict one time series based on the past values of all time series in the dataset. A time series $X_i$ is considered a potential cause of the target time series $X_j$ if the attention score is beyond a certain threshold. By comparing all attention scores, a set of potential causes is formed for each time series. TCDF validates whether a potential cause

(found by the attention mechanism) is an actual cause of the predicted time series by applying a causal validation step. TCDF uses *permutation importance (PI)* as a causal validation method which measures how much an error score increases when the values of a variable are randomly permuted. Finally, all validated causal relationships are included in a temporal causal graph. TCDF learns the time delay between cause and effect by interpreting the network's kernel weights. This framework has experimented with simulated financial market data and FMRI data. It discovered roughly 95–97% of the time delays correctly. However, it performs slightly worse on short time series in FMRI data since a deep learning method has many parameters to fit. An implementation of TCDF can be found at: `https://github.com/M-Nauta/TCDF`.

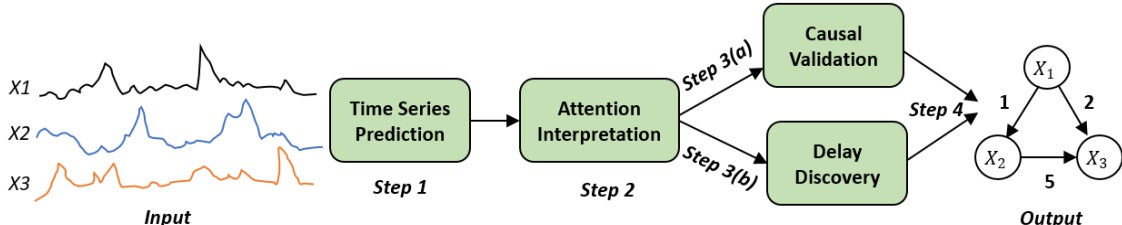

Figure 29: Interpretation of how the TCDF (Nauta et al. (2019)) method works. The numbers on the causal edges denote the time delay between the cause and effect.

### 4.5.3 NBCB

NBCB (Assaad et al. (2021)) or Noise-based/Constraint-based approach is a *hybrid* approach that learns a *summary causal graph* from observational time series data without being restricted to the Markov equivalent class even in the case of instantaneous relations. A *summary causal graph* is one that represents the causal relations between time series without including lags. That is, it only represents the cause-effect relations in a given time series without the time delay between the cause and the effect. To find the summary graph, NBCB uses a hybrid approach which is divided into two steps. First, it uses a noise-based procedure to find the potential causes of each time series under the assumption of additive noise models (ANMs). Then, it uses a constraint-based approach to prune all unnecessary causes and hence ends up with an oriented causal graph. The second step is based on a new temporal causation entropy measure proposed by this study that is an extension of the causation entropy to time series data for handling lags bigger than one time step. Furthermore, this study relies on a lighter version of the faithfulness hypothesis, namely adjacency faithfulness. An implementation of NBCB is available in the site `https://github.com/ckassaad/causal_discovery_for_time_series`.

### 4.5.4 PCTMI

PCTMI (Assaad et al. (2022a)) is an entropy-based approach that discovers the summary causal graph for time series data with potentially different sampling rates. To do so this study proposes a new *temporal mutual information measure* defined on a window-based representation of time series. Then it shows how this measure relates to an entropy reduction principle that can be seen as a special case of the *probabilistic raising principle*. PCTMI combines these two concepts in a PC-like algorithm (Spirtes et al. (2000b)) to construct the summary causal graph. PCTMI focuses particularly on the summary graph, rather than the full-time graph. It has mainly two steps: *(i) Skeleton construction and (ii) Edge orientation*. The skeleton construction as well as the orientation of instantaneous relations is similar to the PC algorithm but adapted for time series data. To orient the lagged relations, it uses the rules of an *entropic reduction* (ER) principle (Michalos (1972)). PCTMI assumes both the causal Markov condition and faithfulness of the data distribution, common assumptions for constraint-based CD approaches. An implementation of PCTMI is available on this website: `https://github.com/ckassaad/causal_discovery_for_time_series`.

The methods discussed above use different strategies to perform causal discovery under a variety of settings and assumptions. Therefore, we present a comparative analysis of some of the common approaches in Table 7 based on their assumptions, output, techniques used, advantages, and disadvantages.

Table 7: Comparison of some causal discovery algorithms for time series data.

| Methods | Assumptions | Technique used | Outcome Graph | Advantages | Disadvantages |
|---|---|---|---|---|---|
| tsFCI | Causal Markov condition, faithfulness | Constraint-based, FCI adaptation for time series data | Partial ancestral graph | Handles hidden confounders | Non-stationarity property is not handled |
| PCMCI | Data stationarity, causal sufficiency | Constraint-based, momentary CI (MCI) test | Full time graph with lagged edges | Handles high-dimensional networks, applicable on linear or nonlinear, and continuous or discrete data | Contemporaneous edges is not identified, some causal links remain unoriented |
| PCMCI+ | Causal sufficiency | Constraint-based, MCI test | Full time graph with lagged and contemporaneous edges | Detects both contemporaneous and lagged causal links | Some causal links remain unoriented |
| LPCMCI | Causal faithfulness, absence of selection bias | Constraint-based | Time series directed maximal ancestral graphs (DMAGs) | Handles large-scale time series data, latent confounders, and discrete/continuous-valued data, incorporates prior knowledge | May lead to wrong conclusions if faithfulness is violated, can't distinguish all members of MEC |
| CD-NOD | Pseudo Causal Sufficiency, faithfulness, no selection bias | Constraint-based | Summary or full time graph incorporating the changing modules | Detects changing modules | Some causal directions may not be identifiable, if the identifiability conditions are not satisfied |
| VarLiNGAM | DGP is linear, there are no unobserved confounders, and non-Gaussian noises with non-zero variances | FCM/SEM-based | Full-time causal DAG (with causal strengths) | Detects both lagged and instantaneous temporal edges | Hidden confounders are not handled, and inapplicable to non-linear data |
| TiMINO | Independent noise, identifiability | FCM/SEM-based | Causal summary time graph | Applicable to multivariate, linear, nonlinear and instantaneous cases | Rigid about the model assumptions, sometimes remain undecided |
| DYNOTEARS | Acyclicity, network structure is fixed through time and is identical for all time series in the data | Score-based approach | Full time causal graph | Scalable to high-dimensional datasets, produces both lagged and instantaneous edges | Undersampling is not handled well, inapplicable to non-linear data |
| NTS-NOTEARS | No latent confounders, DGP is fixed and stationary over time. | Gradient-based, uses 1-D CNNs | Full time causal graph | Incorporates prior knowledge | Inapplicable when there exists latent confounders |
| NAVAR | Additive structure | Granger-causality based | Full time causal graph (with lagged edges only) | Easy to implement, can detect nonlinear relations | Produced score-matrix can be analyzed only by the CausaMe platform |
| oCSE | Faithfulness, Acyclicity | Optimal Causation Entropy principle | Summary causal graph | Often requires a relatively smaller number of samples | Takes very long to run time for dense graphs |
| TCDF | Temporal precedence: the cause precedes its effect | Attention-based CNNs | Full time causal graph | Easy to implement, can detect hidden confounders | Performs slightly worse on short time series, does not ensure acyclicity |
| NBCB | Adjacency faithfulness | Hybrid (constraint and noise-based), causation entropy | Summary causal graph | Not restricted to the MEC | Does not estimate a full-time causal graph |
| PCTMI | Causal Markov condition and faithfulness | Constraint-based, skeleton construction using PC, uses ER principle to orient lagged edges | Summary causal graph | Lower complexity | Does not estimate a full-time causal graph |

# 5 Evaluation Metrics for Causal Discovery

In this section, we discuss the common metrics used to evaluate the performance of causal discovery algorithms. These metrics are common for both I.I.D. and time series causal discovery evaluation.

- **Structural Hamming Distance (SHD):** SHD is the total number of edge additions, deletions, or reversals that are needed to convert the estimated graph $G'$ into its ground-truth graph $G$ (Zheng et al. (2018); Cheng et al. (2022)). It is estimated by determining the missing edges, extra edges, and edges with incorrect direction in the produced graph compared to its true graph. A lower hamming distance means the estimated graph is closer to the true graph, and vice versa. An estimated graph is fully accurate when its SHD = 0. We show the calculation of SHD for the graphs in Figure 30 using the formula in Equation 14 where $A$ = total number of edge additions, $D$ = total number of edge deletions, and $R$ = total number of edge reversals. In Figure 30, we need to *add* the edge $D \rightarrow C$, *delete* the edges $D \rightarrow B$ and $D \rightarrow A$, and *reverse* the edges $C \rightarrow B$ and $C \rightarrow A$ in the generated graph (graph b) to convert it into the true graph (graph a). Therefore, the $SHD = 1 + 2 + 2 = 5$ means a total of 5 actions are required to reach the true graph (graph a).

$$SHD = A + D + R \tag{14}$$

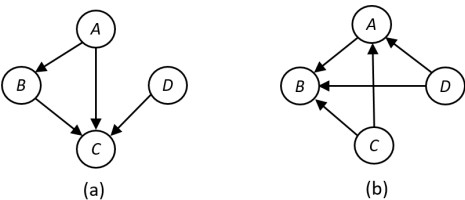

Figure 30: (a) Ground-truth graph $G$, and (b) Estimated graph $G'$.

- **Structural Intervention Distance (SID):** SID is a distance metric for DAGs proposed by Peters & Bühlmann (2015). It measures the closeness between DAGs in terms of their capacities for causal effects. Specifically, it computes the number of falsely inferred intervention distributions (Cheng et al. (2022)) to reflect how false edges in the generated graph can influence the effects obtained.

- **False Discovery Rate (FDR):** FDR is the expected fraction of false discoveries among all the discoveries. In terms of causal discovery, FDR represents the ratio of the extra edges over the sum of the true edges and extra edges. Here, extra edges mean the edges that are present in the estimated graph but not present in the actual graph or the false positives (FP), and true edges mean edges that are present in both the graphs or the true positives (TP). The lower the FDR, the better the performance of causal discovery.

$$FDR = \frac{FP}{TP + FP} \tag{15}$$

- **True Positive Rate (TPR):** TPR denotes the proportion of the positives in the data correctly identified as positives. In terms of causal graphs, TPR is the ratio of the edges in the estimated graph that are also present in the true graph (TP) to the total number of true edges (true positives (TP) and false negatives (FN)). The higher the TPR of an estimated graph, the better the discovery.

$$TPR = \frac{TP}{ActualPositive} = \frac{TP}{TP + FN} \tag{16}$$

- **False Positive Rate (FPR):** In general terms, FPR is the proportion of negatives that are incorrectly identified as positives. In terms of causal graphs, FPR is the ratio of the false edges produced by the estimated

graph that are absent in the true graph (false positives/extra edges) over the sum of true negatives (TN) and false positives (FP). The lower the FPR, the better the causal discovery performance.

$$FPR = \frac{FP}{Actual Negative} = \frac{FP}{TN + FP} \tag{17}$$

- **Precision:** Precision returns the proportion of true positives (TP) among all the values predicted as positive. That is, out of all the positives predicted, what percentage is truly positive. In terms of causal discovery, precision is the fraction of the correct or semi-correct edges over all the produced edges (Shen et al. (2020)).

$$Precision = \frac{TP}{TP + FP} \tag{18}$$

- **Recall:** Recall returns the proportion of the correctly predicted positive values. That is, out of the total positives, what percentage are predicted as positive? In causal discovery, recall is the fraction of edges in the ground-truth graph that are correctly or semi-correctly estimated (Shen et al. (2020)). The recall metric is the same as TPR.

$$Recall = \frac{TP}{TP + FN} \tag{19}$$

- **F1 Score:** The F1 score metric combines the precision and recall metrics into a single metric. It is the harmonic mean of precision and recall and is mostly used in cases of imbalanced data.

$$F1\ score = \frac{2TP}{2TP + FN + FP} \tag{20}$$

- **Matthews Correlation Coefficient (MCC):** MCC is a single-value metric that summarizes the confusion matrix. It takes into account all four entries of the confusion matrix (TP, TN, FP, and FN). The value of MCC is 1 when the discovery of edges is fully accurate (FP = FN = 0), indicating perfect causal discovery. On the contrary, when the algorithm always misidentifies (TP = TN = 0), then the MCC is -1, representing the worst possible discovery. Thus, the MCC value lies between -1 and 1.

$$MCC = \frac{TP \times TN - FP \times FN}{\sqrt{(TN + FN)(FP + TP)(TN + FP)(FN + TP)}} \tag{21}$$

From the above-listed metrics, the SHD metric is more insightful compared to the others as it allows us to know how far the estimated graph is from the ground-truth graph. SHD provides insight into the total number of modifications that need to be done in the estimated graph to transform it into the ground-truth graph. The MCC metric is also good in terms of summarizing the outcome of a CD algorithm as it is the only metric that takes into account all of TP, TN, FP, and FN. The F1 score metric is quite useful too as it combines the precision and recall metrics. One should consider using the SHD metric in order to understand how far the estimated graph is from the ground-truth graph. The TPR metric can be used if one is interested only in knowing the proportion of the true edges discovered. One can use the FDR metric to get an idea about the proportion of the false estimated edges by the algorithm. However, consideration of only a single metric for the performance evaluation of the algorithms can be problematic. A single metric can not fully express the actual performance of the approach.

## 6  Datasets for Causal Discovery

There are a couple of benchmark causal discovery datasets from different domains that are often used for the evaluation of causal discovery approaches. In this section, we discuss briefly some commonly used I.I.D. and time series datasets.

### 6.1 I.I.D. datasets

- **ASIA:** ASIA is a synthetic dataset, also known as the Lung Cancer dataset (Lauritzen & Spiegelhalter (1988)). The associated graph (Figure 31 (a)) is a small toy network that models lung cancer in patients from Asia. Particularly, it is about different lung diseases (tuberculosis, lung cancer, or bronchitis), their relations to smoking, and patients' visits to Asia. This dataset is often used for benchmarking causal graphical models. The ground-truth graph has 8 nodes and 8 edges. Lippe et al. (2021), and Hasan & Gani (2022) have used this dataset for the evaluation of their approaches. It is available here: `https://www.bnlearn.com/bnrepository/discrete-small.html#asia`.

- **LUCAS**: The LUCAS (Lung Cancer Simple Set) is a synthetic dataset that contains toy data generated artificially by causal Bayesian networks with binary variables (Lucas et al. (2004)). Here, the target variable is *Lung Cancer*. The data-generating model of the LUCAS dataset is a Markov process, which means that the state of the children is entirely determined by the state of the parents. The ground-truth graph (Figure 31 (b)) is a small network with 12 variables and 12 edges. Hasan & Gani (2022) used this dataset to evaluate their framework. The dataset and ground truth can be found here: `https://www.causality.inf.ethz.ch/data/LUCAS.html`.

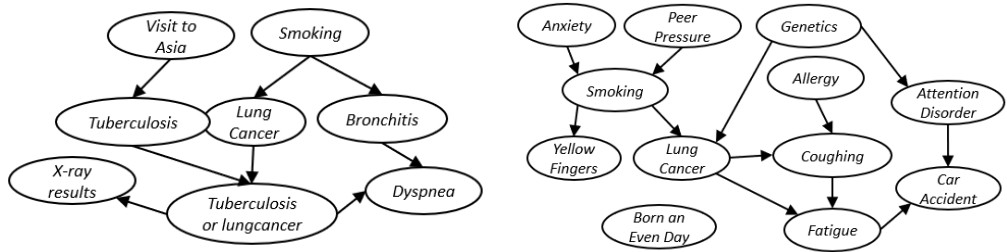

Figure 31: Ground-truth network of the ASIA (left), and LUCAS (right) datasets.

- **SACHS**: SACHS (Sachs et al. (2005)) is a real dataset that measures the expression levels of multiple phosphorylated protein and phospholipid components in human cells. It is the most commonly used dataset for evaluating causal discovery approaches. It has a small network with 11 nodes and 17 edges (Figure 32). The dataset has both observational and interventional samples. Most of the CD approaches use the $n = 853$ observational samples to evaluate their method. This dataset has been used by many approaches such as Zheng et al. (2018), Zhu et al. (2019), Ng et al. (2020), Lachapelle et al. (2019) & Ng et al. (2022), Lippe et al. (2021) for evaluation purposes. Link: `https://www.bnlearn.com/bnrepository/discrete-small.html#sachs`.

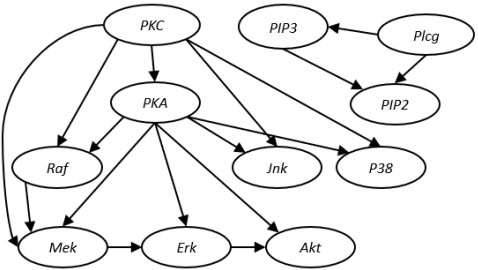

Figure 32: Ground-truth network of the SACHS dataset.

- **CHILD**: The CHILD (Spiegelhalter et al. (1993)) dataset is a medical Bayesian network for diagnosing congenital heart disease in a newborn "blue baby". The ground-truth network is a medium graph that consists of 20 nodes and 25 edges (Figure 34). The dataset includes features such as patient demographics, physiological characteristics, and lab test reports (Chest X-ray, CO2 reports, etc.). This dataset was used by Lippe et al. (2021) in their study, and can be found here: `https://www.bnlearn.com/bnrepository/discrete-medium.html#child`.

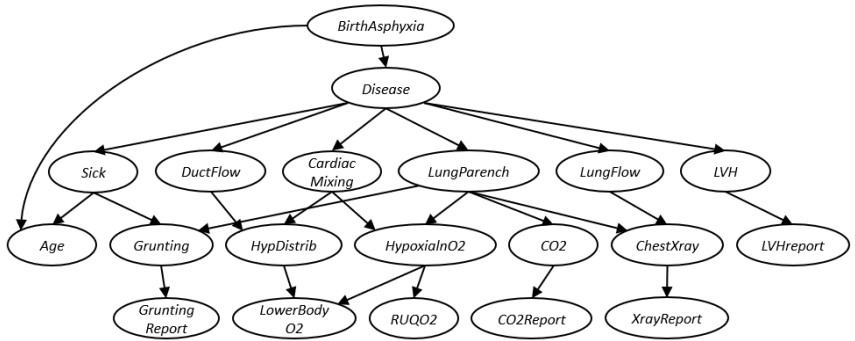

Figure 33: Ground-truth network of the CHILD dataset.

- **ALARM**: A Logical Alarm Reduction Mechanism (ALARM) is a patient monitoring system (Beinlich et al. (1989)) designed to provide an alarm message for patients, and has an associated synthetic dataset. In particular, it implements a cautionary alarm message for patient monitoring. The ground-truth graph is a medium-sized network with 37 nodes and 46 edges. This dataset was used by Yu et al. (2019), and Cai et al. (2013) to evaluate their approaches. The ground-truth network is available in this repository: `https://www.bnlearn.com/bnrepository/`.

- **HEPAR2**: It is a probabilistic causal model for the diagnosis of liver disorders (Onisko (2003)). This causal Bayesian network tries to capture the causal links among different risk factors, diseases, symptoms, and test results. The ground-truth graph is a large network with 70 nodes and 123 edges which is available in the bnlearn (Scutari (2009)) repository: `https://www.bnlearn.com/bnrepository/discrete-large.html#hepar2`.

## 6.2 Time Series datasets

- **fMRI** datasets: Functional Magnetic Resonance Imaging (fMRI) is a popular approach to investigating dynamic brain networks (Cao et al. (2019)). Different types of fMRI data are often used to evaluate time-series causal discovery approaches. Zhang et al. (2017) used the fMRI Hippocampus dataset (Laumann et al. (2015)) that contains signals from six separate brain regions. Nauta et al. (2019) used a simulated blood oxygen level-dependent (BOLD) fMRI dataset that has 28 different underlying networks from 50 brain regions. It measures the neural activity of different brain regions based on the changes in blood flow. Huang et al. (2020) tested their approach using the task fMRI data to learn information flows between brain regions, and how causal influences change across resting state and task states. Some simulated fMRI data is available here: `https://github.com/M-Nauta/TCDF/tree/master/data/fMRI`.

- **CauseMe - Earth Sciences** data: CauseMe (Muñoz-Marí et al. (2020)) is a platform that contains benchmark causal discovery datasets to evaluate, and compare the performance of different CD approaches. It contains datasets generated from both synthetic models mimicking real challenges and real-world data sets from the earth science domain where the ground-truth network is known with high confidence. Bussmann et al. (2021) used different datasets from the CauseMe platform in their study. Specifically, they used the synthetic nonlinear VAR dataset, the hybrid climate and weather dataset, and the real-world river run-off dataset to evaluate their algorithm. It was also used by Runge et al. (2019) in their experiments. The datasets can be found at: `https://causeme.uv.es/`.

- **causaLens** datasets: Lawrence et al. (2021) from causaLens proposed a framework for generating synthetic time series data with a known ground truth causal structure for evaluating time series causal discovery approaches. They have an open-source repository (`https://github.com/causalens/cdml-neurips2020`) that captures the source code and datasets of their proposed framework. Datasets can be generated specifying different assumptions (causal sufficiency, I.I.D., instantaneous effects, etc.) using an example script in the repository. This facilitates the users to generate

data as per their requirements. Located in England, causaLens is a leading software company with a focus on developing intelligent machines based on causal AI.

- **DREAM3 challenge** datasets: DREAM3 (Prill et al. (2010)) is a simulated gene expression dataset often used for evaluating time-series causal discovery algorithms. It has five different datasets of E. coli and yeast gene networks (Ecoli1, Ecoli2, Yeast1, Yeast2, and Yeast3), each consisting of a maximum of 100 variables. Bussmann et al. (2021) used this dataset to evaluate their approach. Every dataset has 46 time series and every time series consists of only 21 timesteps. Some of these datasets can be found here: `https://github.com/bartbussmann/NAVAR`.

- **Stock market** datasets: Stock market datasets contain multiple continuous time series data which are very useful to assess temporal causal discovery algorithms. Huang et al. (2020) used two different stock market datasets downloaded from Yahoo Finance to test their approach. It contains daily returns of stocks from Hong Kong and the United States. Link to a simulated Finance dataset from the study by Nauta et al. (2019): `https://github.com/M-Nauta/TCDF/tree/master/data/Finance`.

## 7 Benchmarking Causal Discovery Algorithms

In this section, we report the performance of some common causal discovery approaches on I.I.D. and time series datasets. We compare the approaches in terms of three common metrics: *SHD*, *TPR*, and *FDR*.

### 7.1 Experiments on I.I.D. data

For causal discovery on the I.I.D. datasets, we choose the following commonly used datasets with available ground-truth graphs: *ASIA* (small network), *CHILD* and *ALARM* (medium networks), and *HEPAR2* (large network). The CSV version of the datasets and their corresponding ground-truths are available in the causal-learn repository: `https://github.com/py-why/causal-learn`. The causal discovery approaches that are benchmarked for the I.I.D. datasets are: *PC, GES, LiNGAM, Direct-LiNGAM, NOTEARS, DAG-GNN, GraN-DAG, GOLEM,* and *MCSL*. The implementations of the algorithms have been adopted from the gCastle (Zhang et al. (2021a)) package.

Table 8: The benchmarking of some common causal discovery algorithms for I.I.D. datasets. The best results w.r.t each metric (SHD, TPR, and FDR) are boldfaced. Lower SHD and FDR are better, while a higher TPR signifies a better performance.

| | ASIA | | | CHILD | | | ALARM | | | HEPAR2 | | |
|---|---|---|---|---|---|---|---|---|---|---|---|---|
| **Methods** | SHD | TPR | FDR | SHD | TPR | FDR | SHD | TPR | FDR | SHD | TPR | FDR |
| PC | 5 | 0.6 | 0.3 | 43 | 0.24 | 0.86 | 55 | 0.67 | 0.6 | 172 | 0.35 | 0.75 |
| GES | **4** | **0.63** | 0.38 | 34 | 0.38 | 0.89 | 56 | **0.74** | 0.61 | **70** | 0.5 | 0.23 |
| LiNGAM | 7 | 0.25 | 0.6 | 23 | 0.28 | 0.63 | 43 | 0.43 | 0.55 | 111 | 0.1 | 0.32 |
| Direct-LiNGAM | **4** | 0.5 | **0** | 28 | 0.12 | 0.82 | 40 | 0.39 | 0.5 | 110 | 0.1 | 0.07 |
| NOTEARS | 12 | 0.13 | 0.83 | **22** | 0.16 | 0.64 | 41 | 0.17 | 0.38 | 123 | - | - |
| DAG-GNN | 7 | 0.25 | 0.5 | 24 | 0.24 | 0.7 | **39** | 0.196 | **0.31** | 123 | 0 | 1 |
| GraN-DAG | 7 | 0.13 | **0** | 24 | 0.04 | **0** | 44 | 0.044 | 0.75 | 122 | 0.008 | **0** |
| GOLEM | 11 | 0.25 | 0.75 | 49 | 0.2 | 0.88 | 60 | 0.26 | 0.71 | 157 | 0.05 | 0.89 |
| MCSL | 19 | 0.5 | 0.82 | 140 | **0.56** | 0.91 | 464 | 0.72 | 0.93 | 1743 | **0.45** | 0.97 |

From the results reported in Table 8, we see that for the ASIA dataset, both GES and Direct-LiNGAM approaches have the best (lowest) SHD. MCSL on the other hand has the worst (highest) SHD for ASIA. For the CHILD dataset, NOTEARS performs the best w.r.t. SHD, and once again MCSL has the worst SHD. A reason for the poor performance of MCSL could be due to its tendency to produce as many as possible edges without caring much about the false positives. DAG-GNN has the best (lowest) SHD for the ALARM dataset, and once again GES outperforms others with the lowest SHD in the case of the HEPAR2 dataset. In terms of TPR, GES, and MCSL both outperform others twice. That is GES has the best TPR for

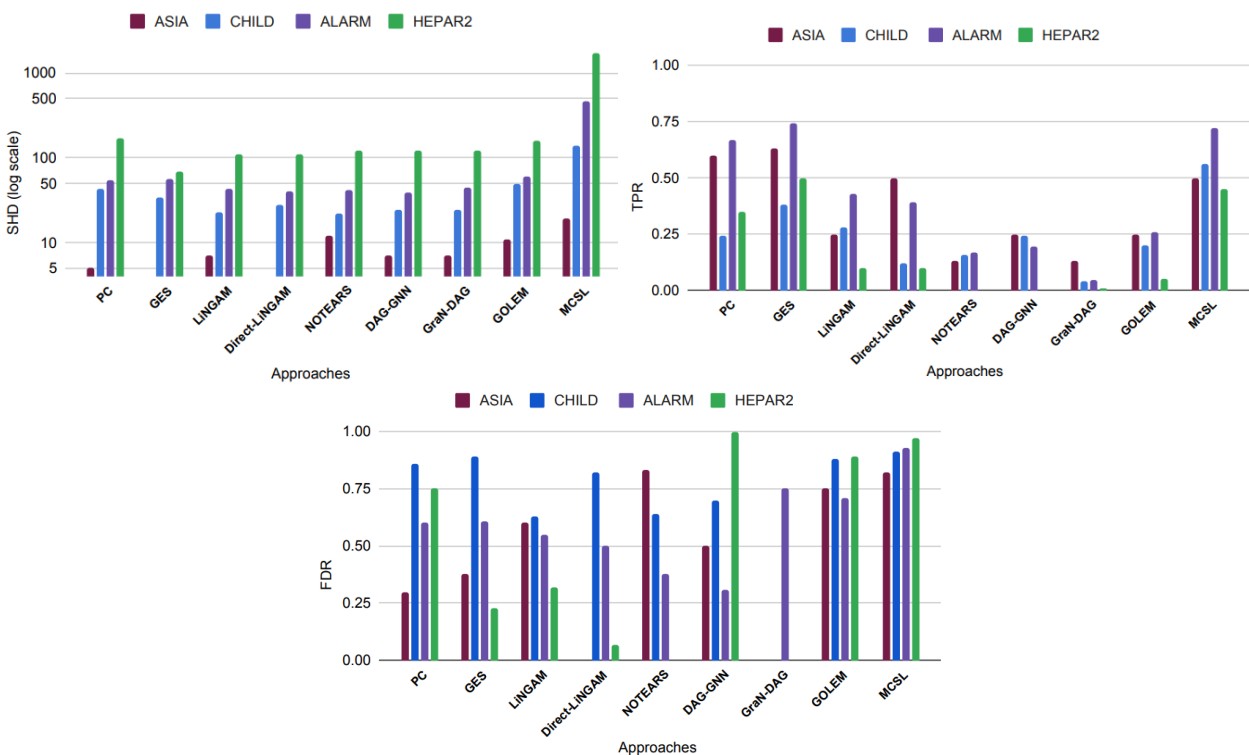

Figure 34: SHD, TPR, and FDR plots of the different benchmarked approaches on some I.I.D. datasets. Lower SHD and FDR are better, while a higher TPR signifies a better performance.

the ASIA and ALARM networks, and MCSL has the highest TPR for the CHILD and HEPAR2 networks. With respect to FDR, GraN-DAG outperforms the other algorithms with the lowest FDR in the case of all the datasets except the ALARM dataset. DAG-GNN has the best FDR in the case of ALARM. PC and GES seem to do well comparatively for the datasets with small graphs. Gradient-based methods such as NOTEARS, DAG-GNN, and GraN-DAG seem to be on par w.r.t. the SHD metric across all the datasets. However, the metrics of all the approaches in the case of the HEPAR2 dataset which has a large ground-truth network are quite poor. This signifies that most of the existing approaches are not fully sufficient to handle large or very large networks, and should focus on improving their scalability. The development of new approaches should consider the scalability factor of the algorithm so that they can handle real-world large networks having 100 to 1000 nodes.

## 7.2 Experiments on time series data

We compared the performance of some temporal causal discovery algorithms namely PCMCI, PCMCI+, VarLiNGAM, DyNOTEARS, and TCDF on two time series datasets. The first dataset is the *Syn-6* data which is a synthetic dataset with 6 variables and a lag period of 2. The details of data generation can be found in the study Ferdous et al. (2023b). The second dataset is *fMRI* data with 10 variables having a lag period of 1. Please refer to the study Nauta et al. (2019) for the details of this dataset. The ground-truth graphs of both datasets are also available in the listed studies. Some of the temporal algorithms only produce summary causal graphs that lack any information about the time lag between the cause and effect. Since both of our experimental datasets have ground-truth graphs with specified time lags, we tested and compared only those temporal algorithms that specify the time lags (i.e. produce full-time causal graphs). The implementation of the PCMCI and PCMCI+ have been adopted from the following repository: `https://github.com/jakobrunge/tigramite`, VarLiNGAM and DyNOTEARS from `https://github.com/ckassaad/causal_discovery_for_time_series`, and TCDF from `https://github.com/M-Nauta/TCDF`. The performance metrics from the conducted experiments are reported in Table 9.

Table 9: The benchmarking of some common causal discovery algorithms for time series data. The best results w.r.t each metric (SHD, TPR, and FDR) are boldfaced. A lower SHD and FDR are better. While a higher TPR signifies a better performance.

| Methods | Syn-6 data | | | fMRI data | | |
|---|---|---|---|---|---|---|
| | SHD | TPR | FDR | SHD | TPR | FDR |
| PCMCI | 14 | 0.63 | 0.69 | 61 | 0.52 | 0.82 |
| PCMCI+ | **11** | 0.50 | **0.63** | **26** | 0.43 | **0.61** |
| VarLiNGAM | 15 | 0.50 | 0.73 | 30 | 0.48 | 0.66 |
| DyNOTEARS | 28 | **0.83** | 0.83 | 95 | **0.81** | 0.84 |
| TCDF | 10 | 0 | 1 | 27 | 0.38 | 0.64 |

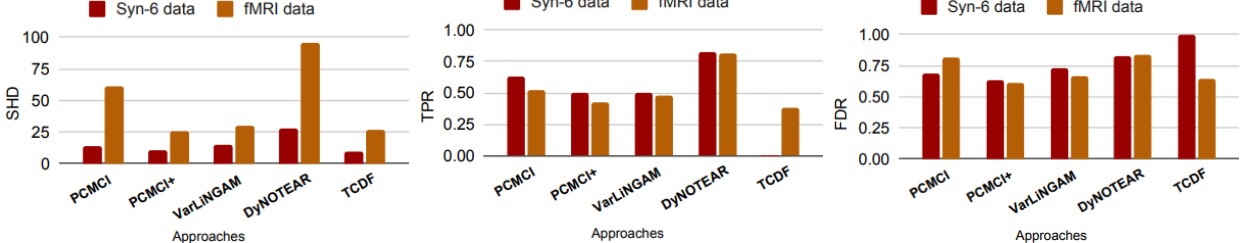

Figure 35: SHD, TPR, and FDR plots of the benchmarked approaches on the time series datasets. A lower SHD and FDR are better. While a higher TPR signifies a better performance.

The above results demonstrate that PCMCI+ performs the best in the case of both datasets w.r.t. the metrics SHD and FDR. This signifies that PCMCI+ produces lower false edges compared to the other approaches. DyNOTEARS has the best TPR in the case of both datasets. This signifies its ability to discover more true edges than others. However, it does not perform well in terms of SHD and FDR. TCDF performs very poorly in the case of the Syn-6 data. It couldn't discover a single true edge which causes its TPR to be 0 and a very high FDR. Although in the case of fMRI data, it has good SHD and FDR, it still has the lowest TPR signifying its tendency to produce a low amount of true edges.

Table 10: Performance results of two other temporal causal discovery algorithms. These are compared separately since the edges produced by these methods do not provide any time lag information.

| Methods | Syn-6 data | | | fMRI data | | |
|---|---|---|---|---|---|---|
| | SHD | TPR | FDR | SHD | TPR | FDR |
| oCSE | 9 | 0.75 | 0.54 | 15 | 0.48 | 0.29 |
| GrangerPW | 15 | 1 | 0.65 | 19 | 0.57 | 0.45 |

We further tested the ability of some temporal algorithms which only produce a summary time graph as their outcome. Their performance results are reported in Table 10. The graphs produced by these methods do not incorporate any time lag information. Hence, only partial information may be obtained from these summary graphs. From the empirical results, we see that in the case of both datasets, oCSE performs better than GrangerPW in terms of the SHD and FDR metrics. While GrangerPW outperforms oCSE in both cases w.r.t. TPR.

# 8 Tools for Causal Discovery

We briefly introduce the tools and software publicly available for users to perform causal discovery. These tools include the implementations of some benchmark causal discovery approaches as well as famous datasets, and commonly used evaluation metrics. Please refer to the table in the following page for the details of the tools or software packages.

Table 11: A brief overview of the tools/packages for Causal Discovery.

| | Repository | Language | Developer | Year | Available algorithms/ datasets/ APIs |
|---|---|---|---|---|---|
| 1 | bnlearn | R | Marco Scutari, Ph.D. | 2007 | • Implements a variety of score-based, constraint-based, hybrid algorithms. • Also, contains famous benchmark networks in different formats such as BIF, DSC, NET, RDA, and RDS. • Link: `https://www.bnlearn.com/`. |
| 2 | CDT | Python | FenTech solutions | 2018 | • Causal Discovery Toolbox has algorithms from the bnlearn package, pcalg packages, etc. based on observational data. • Link: `https://github.com/FenTechSolutions/CausalDiscoveryToolbox` |
| 3 | Tigramite | Python | Jakob Runge | 2017 | • Temporal CD algorithms such as PCMCI, PCMCI+ and LPCMCI. • Link: `https://github.com/jakobrunge/tigramite` • A GUI version of the package is also available for easy user control. |
| 4 | causal-learn | Python | CMU-CLeaR group | 2022 | • Implements state-of-the art CD approaches and relevant useful APIs. Also implements CI tests like the Fisher-z test, Chi-Square test, etc., and score-functions such as BIC, BDeu. • Contains several benchmark causal datasets. • Link: `https://github.com/cmu-phil/causal-learn` |
| 5 | gCastle | Python | Huawei Noah's Ark Lab | 2021 | • Includes some recently developed gradient-based causal discovery methods with optional GPU acceleration. • Contains functions for generating data from either simulator or real-world dataset. • Link: `https://github.com/huawei-noah/trustworthyAI` |
| 6 | CausalNEX | Python | QuantumBlack Labs | 2021 | • Implements the NOTEARS algorithm and uses NetworkX to visualize the causal edges. • Contains functions that allow users/experts to add the known causal edges or remove the false edges to model the relationships better. • Link: `https://github.com/quantumblacklabs/causalnex` |
| 7 | Tetrad | JAVA | cmu-phil | 2018 | • Allows graph creation by hand, using a random graph generator, or a search algorithm. • It has a user-friendly GUI which allows users to input data and background knowledge and it outputs the causal graph based on the search algorithm selected by the user. It also allows learning the generated graph's parameters. • Link: `https://github.com/cmu-phil/tetrad` |
| 8 | Causal MGM | R | Neha Abraham & Benos Lab | 2017 | • Implements Mixed Graphical Models (MGMs) to learn causal connections from directed or undirected graphs. Provides a framework for learning causal structure from mixed data (both continuous and discrete). • Link: `https://github.com/benoslab/causalMGM` |
| 9 | py-causal | Python | BD2K Center for Causal Discovery (UPitt-CMU) | 2016 | • Implements CD algorithms (FGES, GFCI, RFCI, FCI, etc. ) for continuous data, discrete data, and mixed data. • Link: `https://github.com/bd2kccd/py-causal` |
| 10 | do-why | Python | Microsoft | 2018 | • Focuses on causal assumption and their validation • Contains CausalDataFrame, an extension of pandas DataFrame • Provides general API for the four steps of causal inference (modeling, identification, estimation, and refutation) • Link: `https://github.com/py-why/dowhy` |

# 9 Challenges and Applications of Causal Discovery

## 9.1 Challenges

Despite the years of progress made in developing different approaches for causal discovery, there exist some concerns, and challenges that need to be addressed during the development of any causal discovery approach. One of the major concerns about the causal discovery algorithms is the strong *assumptions* they make to recover the underlying causal graph from data. These assumptions make the task really challenging when *any of these are violated*. One such assumption is the causal sufficiency which considers that *there are no unobserved/latent variables*. Several methods estimate the causal relationships assuming there are no unobserved confounders. However, this might not be the case in real-world data. When real-world data violates this assumption and has hidden confounders, the estimation results could be distorted, and lead to false conclusions. Often *real datasets have hidden confounders* that must be taken into account to obtain a true causal graph that represents the data generating process efficiently. Otherwise, this may lead to the *possibility of biases* in the analysis. Therefore, the collected observational data with latent confounders is insufficient to infer the true underlying causal structure (Squires et al. (2022)). Some studies such as Jabbari et al. (2017), Liu et al. (2021), etc. address the presence of latent variables in causal discovery. Another assumption which is the causal faithfulness condition also fails in multiple cases (e.g., if some variables are completely determined by others).

Most of the CD algorithms are based on the assumption that the data samples are *independent and identically distributed (I.I.D.)*. However, in many real-world scenarios, the data may have been generated in a different way, and thus, the iid assumption is violated (Lee & Honavar (2020)). In such cases, using CD algorithms that assume that the data is I.I.D. may produce spurious and misleading relationships. Apart from failures of the assumptions, some approaches may get stuck to a local optimum. Especially, greedy methods (e.g. GES (Chickering (2002)), SGES (Chickering & Meek (2015)), etc.) can get trapped in local optimum, even with large datasets. These methods may often produce sub-optimal graphs in the absence of infinite data. *Computational complexity* is another challenge for causal discovery algorithms. The *search space grows super-exponentially* due to the *combinatorial nature of the solution space*, which makes even simple methods computationally expensive (Chickering (1996)). In the case of the score-based approaches, the *large search space* over all possible DAGs is a major drawback. Hence, score-based methods seem to work well when there are a few or moderate number of nodes. However, these methods suffer when the space of equivalence classes tends to grow super exponentially for dense networks. *Lack of abundant observational data* is another major concern for many CD approaches. For constraint-based approaches such as PC (Spirtes et al. (2000b)), FCI (Spirtes et al. (2000a)), etc., accurate CI testing is possible only when an infinite amount of data is available. With a *finite amount of data, conditional independence (CI) tests become really challenging*. Another disadvantage of the constraint-based approaches is that with a *large sample size or high dimensionality*, the *number of CI tests grows exponentially*. Even, in some cases, the algorithm might take weeks to provide the output. That is, the run time of the algorithm becomes way too long.

Structure *identifiability* of the underlying causal model (Shimizu et al. (2006)) is another issue in causal discovery. A causal graph $G$ is typically not identifiable given observational data only, as a set of possible graphs could have generated the data. Also, the statistical issues stemming from high-dimensional datasets are of concern. Apart from these, a major challenge is the *lack of enough benchmark datasets* with ground truth to train and evaluate the developed causal models. The lack of a comprehensive public data repository consisting of ground-truth graphs hinders the proper evaluation of CD approaches. This problem is severe for areas such as climate science where there is almost never any exact ground truth available (Melkas et al. (2021)). Hence, the only way to analyze the produced graphs in such scenarios is to let domain experts inspect those and see if they actually make sense (Ebert-Uphoff & Deng (2017), Gani et al. (2023)).

In the case of causal discovery from *time series data*, along with the aforementioned challenges, there are some other challenges too which cause research in this area to be still growing. In many real-world applications, the observed data are obtained by applying subsampling or temporal aggregation to the original causal processes, which makes it tough to discover the underlying causal graph (Gong et al. (2017)). Also, it is difficult to infer the causal relations across samples with different underlying causal graphs (Löwe et al. (2022)). Some approaches also suffer from nonlinear relations in time-series data. Another important challenge is the

discovery of causal relations from large-scale observational time series datasets which is an active area of research.

## 9.2    Applications

Causal discovery is widely used in various fields, ranging from healthcare, economics, earth science, education, machine learning, natural language processing, and many more. The challenges faced with correlation-based machine learning have facilitated the development of several causal discovery techniques and increased their applications in many domains.

In **biomedical and healthcare domains,** the key research questions revolve around identifying the underlying causal mechanism to find the risk factors that can be changed to cure a disease. To serve this purpose, researchers have been using causal discovery techniques for a long time. Mani & Cooper (1999) used a modified local causal discovery technique to identify the factors contributing to infant mortality in the USA. Wang et al. (2006) used a stepwise causal discovery method to identify active components or combinations of the components in herbal medicine. The *Fast Causal Inference (FCI)* and *Fast Greedy Equivalence Search (FGS)* methods were used by Shen et al. (2020) to see how accurately these techniques can generate the 'gold standard' graph of Alzheimer's Disease. They evaluated the performance of the algorithms on the dataset collected from the Alzheimer's Disease Neuroimaging Initiative (ADNI) and found that the causal graphs generated by FCI and FGES are almost identical to the 'gold standard' graph created from the literature. They also suggested that using longitudinal data with as much prior knowledge as feasible will maximize the effectiveness of causal discovery algorithms. More recently, Shen et al. (2021) proposed a causal discovery technique that can be applied to large-scale Electronic Health Record (EHR) data and has been applied to identify the causal structure for type-2 diabetes mellitus. Before applying the algorithm, they utilized a data transformation method that converts longitudinal data to disease events. The algorithm uses a BIC score to find the causal graph which overlaps 81% with the graph validated by the professionals. Some studies (Bikak et al. (2020), Gani et al. (2023)) have combined the outcomes from several causal discovery algorithms with the opinions of healthcare experts to develop more reliable and plausible causal graphs. Gani et al. (2023) studies the effect of liberal versus conservative oxygen therapy on the mortality of ICU patients where they present an expert-augmented causal estimation framework. The framework systematically combines results from a set of causal discovery algorithms with expert opinions to produce the final causal graph (Figure 36) that is used to answer some important clinical causal queries.

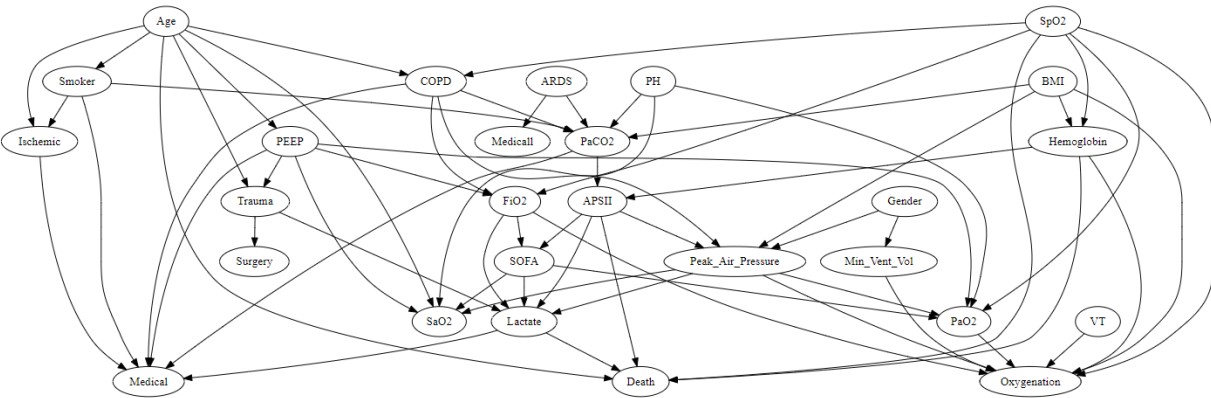

Figure 36: Causal factors determining the influence of oxygen therapy on the mortality of critical care patients (Gani et al. (2023)). This causal graph was determined by the majority voting of 7 causal discovery algorithms combined with opinions from the domain experts.

**Earth science and climate** related research is another domain where causality has been widely adopted. The well-known *PC algorithm* was applied to find the causal links between Eastern Pacific Oscillation (EPO), Western Pacific Oscillation (WPO), North Atlantic Oscillation (NAO), and Pacific-North America (PNA) patterns which are four important patterns of atmospheric low-frequency variability in boreal winter (Ebert-

Uphoff & Deng (2012)). The results, which support earlier research on dynamical processes, suggested that WPO and EPO are almost identical from a cause-and-effect standpoint due to their high contemporaneous coupling. The *PC* and *PC stable* algorithms were applied to daily geopotential height data at 500MB over the boreal winter (Ebert-Uphoff & Deng). The results showed that the atmospheric interactions become less strong on average over the whole Northern Hemisphere. Reduced interconnectedness across various geographic places is the result of this weakening, particularly in the tropics. Causal discovery methods were also applied to verify the results obtained from dynamic climate models. Hammerling et al. (2015) used the *PC algorithm* to learn the causal signatures from the output of the dynamic model. These causal signatures can provide an additional layer of error checking and identify whether the results of dynamic models are accurate or not. Ombadi et al. (2020) applied Granger causality, PC, convergence cross-mapping, and transfer entropy to hydrological models. The authors used these causal discovery methods to identify and investigate the causes of evaporation and transpiration in shrubland areas throughout the course of the summer and winter. Furthermore, the study Huang et al. (2021) investigated the causal relations between multiple atmospheric processes and sea ice variations using three different data-driven causal discovery algorithms, and based on their experiments, they found that it is very challenging to directly apply the state-of-the-art data-driven causal discovery approaches to the specific climate topic considered. Recently, Ali et al. (2023) studied the causal relation between Greenland blocking and sea ice melt using some deep learning-based causal analysis techniques.

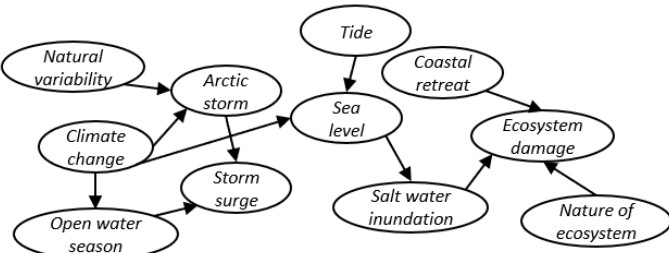

Figure 37: Causal influence of climate and environmental factors on the collapse of an Arctic ecosystem from a storm surge (Shepherd & Lloyd (2021)).

The **education sector** has leveraged causal discovery techniques for decades now. Druzdzel & Glymour (1995) performed an experiment based on the *Tetrad II* (Scheines et al. (1994)) causal discovery program on why the retention rate of U.S. universities is low compared to their reputation. The causal discovery model identified that the retention rate mostly depends on the quality of incoming students. Fancsali (2014) used the *PC* and *FCI* algorithms to answer questions based on their causal effect. They specifically considered the situation given that a student who plays computer games scored poorly in his exam, can the algorithms answer whether reducing gaming time will improve his results? Quintana (2020) employed the *PC* and *FGS* algorithms to find which social and economic factors are directly related to academic achievements. The algorithms found earlier accomplishment, executive functions such as thinking skills, sustained attention focusing, and ambition as the primary drivers of academic performance, which is in line with other studies.

Over the past few years, the intersection of causality with **Machine Learning (ML) and Artificial Intelligence (AI)** techniques is quite a topic of interest. Sun et al. (2015b) utilized *Granger causality* for selecting machine learning features in two-dimensional space. This approach outperformed the traditional feature selection techniques like Principal Component Analysis (PCA) (Abdi & Williams (2010)), Functional Connectome (FC) (Bishop et al. (1995)), and Recursive Feature Elimination (RFE) (Guyon et al. (2002)) due to the ability of *Granger* causality to identify the causal connection between the input variable and the chosen time series. Nogueira et al. (2021) published a survey paper that mainly focused on the applications of causal discovery in machine learning. They discussed how the constraint-based and score-based approaches, as well as causal neural networks and causal decision trees, were applied along with machine learning in various topics. Although previously researchers were not much interested in applying causal techniques in **Natural Language Processing (NLP)**, an important sub-field of AI, recently several causal discovery methods have been applied in this area. To get a deeper explanation, one can read the survey written by

Feder et al. (2021) that discusses the applications of different causal discovery techniques in NLP and how these techniques can help to improve this domain further.

In addition to the abovementioned domains, causal discovery techniques are also being used in **business, macroeconomics, manufacturing, and software engineering,** to name a few. Hu et al. (2013) used a causal Bayesian network with some specialized constraints to analyze the risks associated with software development projects. Luo et al. (2021) used causal discovery models to identify the relationship between flight delays and service nodes. Hall-Hoffarth (2022) employed causal discovery in macroeconomic dynamic stochastic general equilibrium (DSGE) models to learn the underlying causal structure. Vuković & Thalmann (2022) wrote a review paper identifying the applications of causal discovery in manufacturing where root cause analysis, causality in a facilitator role, fault detection, analysis, and management have been highlighted as important areas of application. In business, the understanding of causal relations plays a vital role in designing effective interventions such as launching a new advertising campaign or a promotion (Borboudakis & Tsamardinos (2016)).

Apart from these applications, to learn the causal structure from relational data, Lee & Honavar (2020) developed a method called RRCD (Robust Relational Causal Discovery) where they demonstrated how a CI test created for I.I.D. data can be successfully used to test for relational conditional independence (RCI) against relational data. Moreover, some methods model casual relationships by the incorporation of background knowledge obtained from several sources including experts' opinions, domain knowledge, prior evidence, relevant literature, etc. An importance of such a knowledge-based strategy is that additional causal relationships may become identifiable with the incorporation of background knowledge (Hasan & Gani (2022)). Even specifying one variable as the cause of another, can further refine the set of potential graphs, thereby increasing the number of identifiable causal relationships (Wang et al. (2020)). Some studies (Gani et al. (2023), Adib et al. (2022b), Adib et al. (2022a)) even highlight the importance of human-in-the-loop, and recommend taking into account domain experts' opinions to verify the graphs produced by different causal discovery algorithms. Other than these, causal discovery have been applied in representation learning as well. Yang et al. (2021) proposed a generative model named CausalVAE which learns disentangled and causally meaningful representations of the data by combining ideas of VAE with the SCM. They introduced a *causal layer* with a DAG structure to be learned inside the vanilla VAE model which converts independent exogenous factors into causal endogenous ones.

All in all, causal discovery approaches and techniques have been widely adopted in several areas for understanding the underlying causal relationships, and thereby deriving actionable insights. However, while applying causal discovery methods it is very important to consider the *corresponding assumptions*. If the assumptions made by the respective algorithm are violated by the data, then it may often lead to biased results. Thus, it is important to ensure that the assumptions hold for unbiased discovery of causal graphs.

## 10   Discussion

Traditional AI applications that solely rely upon predictions lack explainability and are often difficult to comprehend due to their black-box nature. *Causal analysis* can overcome the lack of explainability in the existing AI models by embedding casual knowledge into them. These models have greater transparency, and thereby, achieve greater reliability. A crucial part of the causal analysis is *causal discovery*. It is the recovery of the underlying causal structure represented in a graphical form. Such visualizations of causal relationships are easy to comprehend as well as more appealing to a user. In this survey, we introduce a wide variety of existing approaches to perform causal discovery. We also provide a brief overview of the common terminologies used in the area of causal discovery and summarize the different types of algorithms available for structure learning from both I.I.D. and time series data. Apart from discussing the approaches, we also discuss the commonly used datasets, metrics, and toolboxes for performing causal discovery efficiently. In order to select an appropriate approach for learning the causal structure, one can consider some of the following aspects. Selecting the approach whose assumptions are met by the data is very crucial. For example, some methods work well for either linear or non-linear data while some may need a very high amount of samples to operate efficiently. So, before choosing an algorithm, it is important to understand if the assumptions made are also supported by the data or not. When the assumptions are violated by the

data, there is a high chance of obtaining misleading results from the algorithm. Also, some algorithms have a faster run time while others are comparatively much slower. Particularly, some constraint-based methods tend to be much slower when the number of variables are high as they need to conduct a huge amount of CI tests. Therefore, one should also select the approach considering the time available to perform the experiments.

With the growing number of approaches for causal structure learning, an essential *future research direction* is to look deeper into the common challenges or limitations faced during the process. Towards the end of this paper, we discuss some of the common challenges as well as a wide variety of applications of causal discovery in multiple fields. *Future causality research* should focus on the nature of real-world datasets, and develop methods that take into account these practical constraints for better and more reliable structure recovery. It is often observed during experiments that different methods produce causal graphs that disagree with each other to a great extent. In fact, in the experiments (benchmarking) that we performed, we also observed a significant disagreement among the approaches w.r.t. their estimated causal graphs. Therefore, it is needed to accurately quantify the uncertainty of the inferred structures. This is particularly important for the areas such as the healthcare sector which is related to the well-being of humans. It is also important to consider any available background knowledge such as domain expertise, literature evidence, etc. during the causal discovery process which may help to overcome the existing challenges. Once the causal community becomes successful in addressing the existing challenges, we may hope to have better approaches with greater accuracy and reliability.

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
