# OpenReview forum: "A Survey on Causal Discovery Methods for I.I.D. and Time Series Data"
_TMLR — Accepted by TMLR_

### Review · Reviewer_xzg1 · 2023-05-27

**Summary Of Contributions:**

this paper is a survey on existing causal discovery method for i.i.d. and time series data. It discuss some of relevant methods in these areas, as well as metrics, datasets, and available software and toolboxes. In addition, some prior knowledge based approaches are discussed, which is also quite different from existing literature.

**Audience:**

Yes

**Claims And Evidence:**

No

**Requested Changes:**

- Title: temporal data contains irregular time series data (using authors's definition, non-consistent interval data) as well and quite a few causal discovery, which this paper does cover. Authors should just use time series or specify regular time series. Moreover, since authors discuss and used more contents to cover i.i.d. case, it does not make sense to put "non-temporal" after "temporal" data,

- page 2, paragraph 2: a lot of citation are wrongly attributed and should go back to original papers. For example, Chen et al 2021b and Hasan & Ganni 2022 cannot be the first one who mentioned RCT cost.

- by combining i.i.d. data and time series data survey into one, authors maybe could offer some insight across both regimes, which would justify such a combination.

- background knowledge: typical tabu list or other background knowledge is heavily related to Bayesian formulation (as prior). Authors could have explored this direction and related literature much further.

- I suggest authors make a separate section and add more related works on learning graphs with latent variables (there are way more works around MAG/ADMG/latent tree learning, from Richardson et al, and growing body of literature in recent AI conferences) or condense these methods just to briefly describe them, as I don't think they are the focus on this work.

- Organization and Figure 9: while some existing literature my use such taxonomy, I do not quite agree such a classification of different methods, as they are quite confusing and do not illustrate the differences among each other. For example, continuous optimization methods a lot of times uses functional causal model based models (the reason is they share similar objective functions), and dividing them into different categories does not make much sense. In addition, the continuous optimization or gradient-based approach in name is just an optimization approach, in contrast to traditional discrete optimization (such as search approaches, including integer programming formulation used in score-based methods). Their objectives are a scoring function as well (for example, least square or others are log-likelihood objective scores. They may not use Bayesian prior such as complexities in BIC, but they have L1 as complexity control. Overall, they are still a scoring function). Hence, they are also score-and-search methods (just different searching strategies). Prior knowledge based may just be additional constraints in the optimization process, and can be constraint or score-based. Overall, this organization is quite superficial.

- Some inclusion of algorithms and/or its placement in the section organization are quite arbitrary. For example, RRCD seems for relation data, but not pure i.i.d. or times series data. Some algorithms like ANM and PNL focuses on bivariate cases instead of learning the whole graphs, and authors fail to distinguish and highlight it for each method. I'm not sure if JCI with multiple datasets or domains count as prior knowledge or background knowledge based approaches (maybe a separate direction for causal algorithms to improve generalization. along with CD-NOD - which is not for time series data, makes more sense). Moreover, I don't think CausalVAE fit in causal discovery algorithms, as they never evaluate CD performances.

- Moreover, A few important works are not included. For example, a big part of score-based approaches on dynamic programming and integer programming causal discovery algorithms are not discussed (they are the state of the art methods in exact causal discovery learning). Popular CAM method (which include PNS) is not discussed. DAG-NoFear, which highlights the limitation of gradient-based methods, is not discussed.

- While it is nice to have discussion on each method, the overall clarity on the similarity and difference between each method is not very high. If there is a complete list of dimensions to organize and compare these methods, it would greatly improve the paper. Maybe a summarizing table or figure along these dimensions would be useful.

- Which metric is more insightful, and which is problematic? or what conditions does one should consider use them?

- Some datasets do not have link provided, while others do.

- Benchmark: how do you generate data, as some algorithms are traditionally use discrete data while others use continuous algorithms?


Other:

- "gold standard"
- "no cycle among edges" vague
- ". While every endogenous variables" fragment
- "existence of selection biases ..." I don't think these assumptions are common for causal discovery algorithms talked in the paper.
- Anytime FCI "takes into consider selection bias" but authors mention FCI works "in the presence of many arbitrary latent and selection variables".
- PNL " effectively separate cases .." more details needed on "effectively".

**Strengths And Weaknesses:**

Strengths:

- This review is quite long and covers both i.i.d. and times series data causal discovery methods. In term of pure contents, this list is quite extensive.
- list of toolboxes and implementation of each method is quite useful
- for each method, authors add some detailed discussed on their approaches.


Weaknesses:

- the benchmark experiment does not quite compare each method extensively, and only done with i.i.d. data. No conclusion or guidance is provided in this regard, so experiment section is lacking.
- There are a few presentation issues (discussed below). Mainly, I feel the organization of methods does not make too much sense in a lot of places, and some explanation or justification are needed.

---

> ### Author Response · Authors · 2023-06-12
> **Addressing the requested changes**
>
> We would like to express our gratitude for such valuable suggestions and comments. We highly appreciate these comments as we believe they are really essential to improve the overall quality of the survey. We have addressed the requested changes and comments below:
>
> #Requested Change: “Title: temporal data contains irregular time series data (using author's definition, non-consistent interval data) as well as quite a few causal discoveries, which this paper does cover. Authors should just use time series or specify regular time series. Moreover, since authors discuss and use more contents to cover i.i.d. case, it does not make sense to put "non-temporal" after "temporal" data.”
>
> +Reply: Thanks for pointing this out. We agree that i.i.d. should come first, and instead of temporal it should be time series. Hence, we have made the requested changes in the title. The new title is “A Survey on Causal Discovery Methods for I.I.D. and Time Series Data”.
>
> #Requested Change: “page 2, paragraph 2: a lot of citations are wrongly attributed and should go back to original papers. For example, Chen et al 2021b and Hasan & Ganni 2022 cannot be the first one who mentioned RCT cost.”
>
> +Reply: We have removed the previously cited papers for RCT cost and included the citation of the original papers in page 2, paragraph 2.
>
> #Requested Change: “by combining i.i.d. data and time series data survey into one, authors maybe could offer some insight across both regimes, which would justify such a combination.”
>
> +Reply: Thank you for the suggestion. To justify the combination of i.i.d. data and time series data in our survey, we have discussed some reasons for such a combination in the last paragraph of page 2. One reason for such a combination is that in many domains including healthcare and climate change, we often encounter both i.i.d. and time series data in different applications. We believe it is important for the readers to have a comprehensive overview of the causal discovery approaches available for both the domains.
>
> #Requested Change: “Which metric is more insightful, and which is problematic? or under what conditions should one consider using them?”
>
> +Reply: We really appreciate this comment. It is an important discussion to add. Hence, we have added a discussion on this (metrics) in the last paragraph of Section 5 in page 39.
>
> #Requested Change: “Some datasets do not have links provided, while others do”.
>
> +Reply: Thanks for pointing out this issue. We have now added the links for all the listed datasets in Section 6.1 and Section 6.2.
>
> #Requested Clarification: “Benchmark: how do you generate data, as some algorithms traditionally use discrete data while others use continuous algorithms?”
>
> +Reply: The datasets that we have used for comparing the algorithms are all benchmark datasets that are commonly used by different approaches for evaluation purposes. These benchmark datasets are available publicly online in different formats in repositories such as bnlearn, causal-learn, etc. For our experiments, we used the csv version of the datasets available in the causal-learn github repository. We mentioned this in the first paragraph of Section 7.1. Since these datasets are common benchmark i.i.d. datasets, we used them to evaluate all the algorithms to maintain a uniformity in the experimental settings. We also did some new experiments for the temporal causal discovery methods where we used different sets of benchmark time series datasets which are listed in Section 7.2.
>
> #Requested Change: While it is nice to have discussion on each method, the overall clarity on the similarity and difference between each method is not very high. If there is a complete list of dimensions to organize and compare these methods, it would greatly improve the paper. Maybe a summarizing table or figure along these dimensions would be useful.
>
> +Reply: We completely agree with the valuable suggestion that adding a summarizing table for the overall clarity on the similarity and difference between each method would greatly improve the paper. Hence, we have created a summarizing table for both i.i.d. and time series causal discovery algorithms on the basis of some important dimensions such as their assumptions, outcome graph, techniques used, advantages and disadvantages. Table 6 in page no. 28 and Table 7 in page no. 37 are the requested summary tables. We tried to include most of the algorithms particularly the ones that are well established, unique, and commonly used.

---

> > ### Author Response · Authors · 2023-06-12
> > **Addressing the requested changes**
> >
> > #Requested Change: “Mainly, I feel the organization of methods does not make too much sense in a lot of places, and some explanation or justification are needed.”
> >
> > +Reply: We have primarily categorized some methods as a gradient-based method as they use gradient descent for optimization. However, we completely agree that some of the gradient-based approaches compute data likelihood scores. Hence, for a better clarity to the readers, we have updated the Figure 9 by adding a diamond symbol besides those gradient-based methods that also use a score function. We have mentioned clearly in the figure caption that they can be also termed as a score-based method as they compute data likelihood scores on the way. However, we have put them under the gradient-based category based on their primary contribution. Even some of these methods such as GAE, GOLEM, MCSL, etc. clearly mentions in their paper that they are a gradient-based optimization approach. Also, we further agree that prior knowledge is just an additional constraint. Hence, we removed that category, put those algorithms into the right category based on the approach they employ, and also marked an asterisk (*) symbol beside these methods to clarify that they can also leverage some prior knowledge if available.
> >
> > #Requested Change: “Some inclusion of algorithms and/or its placement in the section organization are quite arbitrary. For example,  RRCD seems for relation data, but not pure i.i.d. or times series data. Some algorithms like ANM and PNL focus on bivariate cases instead of learning the whole graphs, and authors fail to distinguish and highlight it for each method. I'm not sure if JCI with multiple datasets or domains count as prior knowledge or background knowledge based approaches (maybe a separate direction for causal algorithms to improve generalization, along with CD-NOD - which is not for time series data, makes more sense). Moreover, I don't think CausalVAE fits in causal discovery algorithms, as they never evaluate CD performances.”
> >
> > +Reply: We agree that RRCD can not be categorized purely for i.i.d. or times series data.  Hence, to remove the ambiguity, we have removed RRCD from the Section 3 i.i.d. methods and discussed it in the section 9 “Application” along with other papers.
> >
> > For JCI, as it's a causal modeling framework rather than a particular algorithm, we have placed it in the miscellaneous approaches section, and the prior-knowledge based category is removed completely to avoid any ambiguity. We further added an asterisk (*) symbol to the JCI method in Figure 9 to indicate that it considers prior knowledge constraints.
> >
> > For CD-NOD, we believe it is for time series data since in the experimental results section of the paper,  they evaluated the CD-NOD on nonstationary time series data.
> >
> > For CausalVAE, we placed it in the causal discovery algorithms as the CausalVAE model involves a Causal Layer with a DAG structure to be learned. However, we agree that it does not properly fit into the causal discovery algorithms, and hence we have removed it from Section 3.
> >
> > We will highlight that ANM and PNL focuses on bivariate cases instead of learning the whole graphs.
> >
> > #Requested Change: “background knowledge: typical tabu list or other background knowledge is heavily related to Bayesian formulation (as prior). Authors could have explored this direction and related literature much further”.
> >
> > +Reply: We have discussed a typical tabu list (tabu search) algorithm for structural equation modeling (SEM) in Section 3.5.13 on page 27. We could not explore more due to the time limitation. However, we will try to discuss some more before the final submission.
> >
> > #Requested Change: “I suggest authors make a separate section and add more related works on learning graphs with latent variables (there are way more works around MAG/ADMG/latent tree learning, from Richardson et al, and growing body of literature in recent AI conferences) or condense these methods just to briefly describe them, as I don't think they are the focus on this work”.
> >
> > +Reply: We appreciate the suggestion. We would like to highlight that the work from Richardson et al (i.e. the RFCI algorithm) is already listed in the survey paper in Section 3.1.4. Also, we have listed a few more works on learning graphs with latent variables which include FCI (3.1.2), ENCO (3.4.8), FRITL (3.6.2), LFCM (3.7.4) and LPCMCI (4.1.4). We have highlighted the fact that these approaches are able to handle latent variables in the summarizing Table 6 as well as in the explanation body of these methods in italic.

---

> > > ### Author Response · Authors · 2023-06-12
> > > **Addressing the requested changes**
> > >
> > > #Requested Change: “Moreover, A few important works are not included. For example, a big part of score-based approaches on dynamic programming and integer programming causal discovery algorithms are not discussed (they are the state of the art methods in exact causal discovery learning). Popular CAM methods (which include PNS) are not discussed. DAG-NoFear, which highlights the limitation of gradient-based methods, is not discussed”.
> > >
> > > +Reply: Thanks for highlighting this. We have included some papers belonging to the exact score-based causal discovery learning. The newly added papers are A-star search (Section 3.2.5) method, Triplet A-star (Section 3.2.6) search method, and also, one method on integer programming (Section 3.2.7) for causal discovery. Also, we included the other suggested papers, popular CAM method (Section 3.3.8) and also, DAGs with No Fears paper (Section 3.4.9). Also, one of the previously included papers, the GAE method, which is a special case of CAM is highlighted in the discussion of the paper.
> > >
> > > #Requested Change:
> > > Other:
> > > "gold standard"
> > > "no cycle among edges" vague
> > > ". While every endogenous variables" fragment
> > > "existence of selection biases ..." I don't think these assumptions are common for causal discovery algorithms talked about in the paper.
> > > Anytime FCI "takes into consideration selection bias" but authors mention FCI works "in the presence of many arbitrary latent and selection variables".
> > > PNL " effectively separate cases .." more details needed on "effectively".
> > >
> > > +Reply: The “other” requested changes have been addressed and are available in the updated paper. Particularly,
> > > "gold standard" is replaced with “standard approach”; and "no cycle among edges" is replaced with “And, a directed graph in which the edges have directions (→) and has no cycle is called a directed acyclic graph (DAG)”.
> > >
> > > In “While every endogenous variables", the word “while” is removed.
> > >
> > > For the Anytime FCI, we mistakenly mentioned the selection bias issue which is redundant here. Now it is corrected.
> > >
> > > For PNL, we added some explanations/details on "effectively"
> > >
> > > Regarding the "existence of selection biases” and other subsequent listed assumptions, we mentioned that these assumptions are observed occasionally, and are not the common assumptions of the causal discovery algorithms discussed in the paper.
> > >
> > > #Mentioned Weaknesses: “The benchmark experiment does not quite compare each method extensively, and only done with i.i.d. data. No conclusion or guidance is provided in this regard, so the experiment section is lacking.”
> > >
> > > +Reply: Thanks for mentioning this. We have included some discussion regarding the performance comparison of the methods in Subsection 7.1. Furthermore, we agree that experiments were done only with i.i.d. data. Therefore, we have done some new experiments to compare the performance of different time series causal discovery algorithms on some benchmark time series datasets, and reported the corresponding evaluation metrics (Table 9) and a discussion of the results in Subsection 7.2.

---

### Review · Reviewer_GhUw · 2023-05-28

**Summary Of Contributions:**

This paper provides a comprehensive review of causal discovery methods, including well-known methods and some new ones. This survey can help beginners quickly become familiar with the methods of causal discovery.

**Audience:**

Yes

**Claims And Evidence:**

Yes

**Requested Changes:**

This survey lacks in-depth analysis and comparisons among the different causal discovery approaches. Therefore, it is recommended that the authors incorporate more comprehensive analyses of the advantages and disadvantages associated with each method, along with guidance on selecting appropriate approaches based on different conditions. Additionally, the inclusion of more empirical results would greatly enhance the overall quality and usefulness of the survey.




**Strengths And Weaknesses:**

Strengths:
The survey is easy to follow and the writing is clear. It covers most of the well-known methods in causal discovery and some new ones. This survey can help beginners quickly become familiar with the methods of causal discovery.

Weaknesses:
This survey is more like a list of causal discovery approaches, but it falls short in terms of in-depth analysis and comparisons among these methods. Furthermore, it may be also important to provide some comments about future research directions.

---

> ### Author Response · Authors · 2023-06-12
> **Addressing the requested changes**
>
> We are really thankful for such valuable suggestions and comments. We highly appreciate these comments as we believe they are really essential to improve the overall quality of the survey. We have addressed the requested changes and comments below:
>
> #Requested Change: This survey lacks in-depth analysis and comparisons among the different causal discovery approaches. Therefore, it is recommended that the authors incorporate more comprehensive analyses of the advantages and disadvantages associated with each method.
>
> +Reply: We really appreciate pointing out this important analysis. We agree with the comment that an in-depth analysis about the advantages and disadvantages associated with the methods is needed. Hence, we updated the paper by adding the comparison tables, Table 6 in page no. 28 and Table 7 in page no. 37 for I.I.D. and time series causal discovery algorithms respectively where we specify the advantages and disadvantages of the approaches as well as compare them on the basis of some important criteria. We would also like to mention that while we discussed each of the approaches, we included their advantages and disadvantages too in those discussions.
>
> #Requested Change: Incorporation of the guidance on selecting appropriate approaches based on different conditions.
>
> +Reply: We are very thankful for this suggestion. We have included a discussion on how to select the appropriate approach based on different scenarios at the end of the first paragraph of Section 10 (Discussion). We believe this guidance will help readers to select the appropriate method.
>
> #Requested Change: Additionally, the inclusion of more empirical results would greatly enhance the overall quality and usefulness of the survey.
>
> +Reply: We agree that inclusion of more empirical results would greatly enhance the overall quality of the survey. Therefore, we have done some new experiments to compare the performance of different temporal causal discovery algorithms on some benchmark datasets, and reported the corresponding evaluation metrics (Table 9) and a discussion of the results too in Section 7.2. .
>
> #Requested Change: Furthermore, it may be also important to provide some comments about future research directions.
>
> +Reply: Thanks for the valuable suggestion. We extended our discussion on the important future research directions in the second paragraph of Section 10 (Discussion) in page no. 50. We understand the importance of such discussion, and hence highlighted them well in italic for the ease of readability.

---

### Review · Reviewer_hQ5b · 2023-06-01

**Summary Of Contributions:**

This survey paper focuses on reviewing causal discovery methods for temporal and non-temporal data. Overall, this paper well presents the basic concepts, terminologies, and assumptions for causal discoveries, and provides a good collection of existing methods with details for both i.i.d. and time series data. The authors also list some benchmark datasets and did comparison studies for i.i.d data. However, this paper is trying to put everything which unfortunately loses the main focus and leads to a few notable weaknesses that the authors may consider addressing.

**Audience:**

Yes

**Broader Impact Concerns:**

The authors may alert audiences in Section 9.2 Applications, that applying causal discovery methods without validating assumption may lead to biased results.

**Claims And Evidence:**

No

**Requested Changes:**

Please consider addressing *Weaknesses* above.

**Strengths And Weaknesses:**

*Strengths*

1. this paper well presents the basic concepts, terminologies, and assumptions for causal discoveries;
2. the authors provide a good collection of existing methods with details for both i.i.d. and time series data;
3. the authors also list a few notable benchmark datasets and did some comparison studies for i.i.d data.

*Weaknesses*

1. As a survey paper, the authors use 7 pages to talk about assumptions and methods for temporal causal discovery, which is very unbalanced compared to the overall (a total of 45 pages). I highly recommend the authors either focus on causal discovery for i.i.d. data, or for temporal data, but definitely not both.

2. As for the part of "causal discovery for i.i.d. data":

 a. What is the difference between this survey and existing works? To name a few:
  - Glymour, Clark, Kun Zhang, and Peter Spirtes. "Review of causal discovery methods based on graphical models." Frontiers in genetics 10 (2019): 524.
  - Nogueira, Ana Rita, et al. "Methods and tools for causal discovery and causal inference." Wiley interdisciplinary reviews: data mining and knowledge discovery 12.2 (2022): e1449.
  - Spirtes, Peter, and Kun Zhang. "Causal discovery and inference: concepts and recent methodological advances." Applied informatics. Vol. 3. No. 1. SpringerOpen, 2016.

 b. Why did the authors separate the score-based and the optimization-based methods? They should all belong to the score-based as the optimization-based methods are also constructing a score.

 c. Please be careful to use "causal discovery" v.s. "learning DAG". These two are different concepts and as a survey, it is important to clarify them.

3. As for the part of "causal discovery for temporal data", many things remain unclear:

 a. What are rigorous definitions of properties on page 28?

 b. What are evaluation metrics here?

 c. Why there is no comparison study for causal discovery methods using temporal data?

 d. What are the challenges here?

Overall, this paper is trying to put everything which unfortunately loses the main focus and leads to a few notable weaknesses that the authors may consider addressing.

---

> ### Author Response · Authors · 2023-06-12
> **Addressing the requested changes**
>
> We are highly thankful for such valuable suggestions and comments. We really appreciate these comments as we believe they are really essential to improve the overall quality of the survey. We have addressed the requested changes and comments below:
>
> #Requested Change: “As a survey paper, the authors use 7 pages to talk about assumptions and methods for temporal causal discovery, which is very unbalanced compared to the overall (a total of 45 pages). I highly recommend the authors either focus on causal discovery for i.i.d. data, or for temporal data, but definitely not both.”
>
> +Reply: We really appreciate your recommendation, and understand your concern. However, the imbalance that seems for the temporal causal discovery methods is due to the reason that compared to the existing algorithms for the I.I.D. data, the amount of algorithms available for time series causal discovery are very few. In fact, the field of temporal causal discovery is a growing area of research compared to the well established area of I.I.D. causal discovery. We have tried to discuss a comprehensive list of the available time series causal discovery methods. Our purpose for this survey is to expose readers to the existing research in causal discovery as well as providing an overview of the entire field covering approaches for both I.I.D. and time series data. Often in daily settings we come across some real-world data that are time series. Hence, we feel it’s essential to present the temporal approaches too for a greater usability of the survey. Moreover, the terminologies presented in Section 2 of our survey are some common terminologies related to causal discovery for both I.I.D. and time series data. We further added benchmark analyses for time series causal discovery methods. We have also discussed future research directions, challenges, applications, metrics, tools, etc. that are common to both I.I.D. and time series methods.
>
> #Requested Clarifications/Changes:
> “As for the part of "causal discovery" for i.i.d. data":
> a. What is the difference between this survey and existing works? To name a few:
> Glymour, Clark, Kun Zhang, and Peter Spirtes. "Review of causal discovery methods based on graphical models." Frontiers in genetics 10 (2019): 524.
> Nogueira, Ana Rita, et al. "Methods and tools for causal discovery and causal inference." Wiley interdisciplinary reviews: data mining and knowledge discovery 12.2 (2022): e1449.
> Sprites, Peter, and Kun Zhang. "Causal discovery and inference: concepts and recent methodological advances." Applied informatics. Vol. 3. No. 1. SpringerOpen, 2016.
>
> +Reply:
>
> (a) The difference between this survey and the existing works are as follows:
>
> The survey by Glymour et.al. only talks about traditional constraint-based, score-based and FCM-based approaches and does not mention the recent advances in the field of causal discovery methods. It does not cover a lot of the recently developed methods as well as it does not explicitly talk about approaches that employ gradient optimization or prior knowledge. Our survey on the other hand tries to discuss as many as possible latest studies in the field, and also we point out about the growing body of literature that handles prior knowledge, employs gradient optimization, tackles latent variables, etc. Also, we have tried to benchmark and compare the performance of several approaches that they have not done.
>
> In the survey by Nogueira, Ana Rita, et al., they have not mentioned in detail about the way each method works as well as listed a few papers in the field of causal discovery. Also, they have not discussed the tools that are available to perform causal discovery easily. We believe such an analysis should be a part of any causal discovery survey to provide the readers with almost everything they need to start any causal analysis.
>
> The study by Sprites, Peter, and Kun Zhang differs from us in that they have not discussed and compared the methods well, and also did not perform an extensive experimental comparison study among the approaches.
>
> Overall, we differ from the existing surveys on multiple grounds as mentioned above, among which discussing most of the significant existing works as well as the latest methods in case of both i.i.d. and time series data, highlighting approaches that leverage prior knowledge, benchmarking of the approaches, discussion of the challenges, applications, tools, etc. are some of the important ones.

---

> > ### Author Response · Authors · 2023-06-12
> > **Addressing the requested changes**
> >
> > #Requested Clarifications/Changes: “As for the part of "causal discovery" for i.i.d. data":
> >
> > b. Why did the authors separate the score-based and the optimization-based methods? They should all belong to the score-based as the optimization-based methods are also constructing a score.
> >
> > c. Please be careful to use "causal discovery" v.s. "Learning DAG". These two are different concepts and as a survey, it is important to clarify them.”
> >
> > +Reply:
> >
> > (b) We understand your concern regarding such separation. We have primarily categorized some methods as a gradient-based optimization method as they use gradient descent for optimization. However, we agree that some of the gradient-based optimization approaches compute data likelihood scores so they should belong to the score-based category. Hence, for a better clarity to the readers, we have updated Figure 9 by adding a diamond symbol beside those gradient-based methods that also use a score function. We have mentioned clearly in the figure caption that they can be also termed as a score-based method as they compute data likelihood scores on the way. However, we have put them under the gradient-based category based on their primary contribution. Even some of these methods such as GAE, GOLEM, MCSL, etc. clearly mentions in their paper that they are a gradient-based optimization approach.
> >
> > (c) Yes, we also agree with this. For that reason, we have specified in Section 2.2 that apart from a DAG, the outcome of some causal discovery methods can be a PDAG, CPDAG, ancestral graphs, etc. and provided a list of the outcome graphs of some common algorithms in Table 3 (Section 2.2) so that readers can have an idea of the expected types of output from a causal discovery task.
> >
> > #Requested Clarifications/Changes: As for the part of "causal discovery for temporal data", many things remain unclear:
> >
> > a. What are rigorous definitions of properties on page 28?
> >
> > b. What are evaluation metrics here?
> >
> > c. Why is there no comparison study for causal discovery methods using temporal data?
> >
> > d. What are the challenges here?
> >
> > +Reply:
> >
> > (a) These definitions have been included to provide any reader with some common background and terminologies that are relevant to temporal causal discovery. These can be helpful for anyone who is not familiar with the temporal causal discovery field, and wants to know some basic and important terms before getting started.
> >
> > (b) The evaluation metrics for causal discovery from temporal data are generally the same as the evaluation metrics for causal discovery from i.i.d. data. Hence we have listed and explained all the evaluation metrics in Section 5 which can be used for both temporal and non-temporal causal discovery algorithms. We updated Section 5 to highlight this.
> >
> > (c) Thanks for mentioning this. We have done some new experiments to compare the performance of different temporal causal discovery algorithms on some benchmark time series datasets, and reported the corresponding evaluation metrics (Table 9), and a discussion of the results too in Subsection 7.2.
> >
> > (d) Thanks for pointing this out. The challenges of causal discovery from iid data are also applicable for causal discovery from temporal data. However, there are some other crucial challenges that are particular to the temporal causal discovery task. We have included a paragraph at the end of Subsection 9.1 mentioning some important challenges faced in temporal causal discovery.
> >
> > #Requested Change: “The authors may alert audiences in Section 9.2 Applications, that applying causal discovery methods without validating assumptions may lead to biased results.”
> >
> > +Reply: Thanks for this valuable suggestion. We have mentioned about the consequences of applying causal discovery methods without validating the respective assumptions in the last paragraph of Section 9.2 “Applications” in page 49.

---

### Author Response · Authors · 2023-06-12
**Thanking the reviewers and the action editor.**

We would like to thank the reviewers and the action editor for their time and valuable feedback. We highly appreciate the comments and suggestions as we believe they are essential to improve the overall quality of the survey. We tried to address all the suggested changes, and also tried to clarify the questions raised.

---

### Decision · Action_Editors · 2023-08-01

**Recommendation:** Accept as is

**Comment:**

The balance of the reviewer sentiment is more towards accept than reject. There were initial and lingering concerns that the material would be better presented as two separate papers: one survey on causal discovery for IID data and one survey on causal discovery for time series data. This may be more of an editorial choice on the part of the authors and did not seem to diminish the quality of the claims or support of the evidence. Indeed, by combining them into one paper the audience may be able to better understand the similarities and differences between methods for each type of data. Overall, the survey is a timely contribution to the causal discovery literature and meets the acceptance criteria for the venue.

**Audience:**

The audience for this paper include machine learning researchers and engineers as well as statisticians looking to bring their understanding of causal discovery methods up to date.

**Claims And Evidence:**

The paper is a survey of existing methods and literature in the area of causal discovery. As such, there are no new claims. However, there are tables and some experiments comparing methods and the reviewers seem to find that the comparisons are accurate after revisions.